# Temporally organized representations of reward and risk in the human brain

Vincent Man [1] ✉, Jeffrey Cockburn [1], Oliver Flouty[2], Phillip E. Gander [3,4,5], Masahiro Sawada[3], Christopher K. Kovach [3,6], Hiroto Kawasaki[3], Hiroyuki Oya [3,5], Matthew A. Howard III[3,5] & John P. O'Doherty [1,7]

The value and uncertainty associated with choice alternatives constitute critical features relevant for decisions. However, the manner in which reward and risk representations are temporally organized in the brain remains elusive. Here we leverage the spatiotemporal precision of intracranial electroencephalography, along with a simple card game designed to elicit the unfolding computation of a set of reward and risk variables, to uncover this temporal organization. Reward outcome representations across wide-spread regions follow a sequential order along the anteroposterior axis of the brain. In contrast, expected value can be decoded from multiple regions at the same time, and error signals in both reward and risk domains reflect a mixture of sequential and parallel encoding. We further highlight the role of the anterior insula in generalizing between reward prediction error and risk prediction error codes. Together our results emphasize the importance of neural dynamics for understanding value-based decisions under uncertainty.

Adaptive behavior is predicated on an ability to evaluate relevant features of the environment. A substantial collection of evidence has demonstrated that humans and animals alike direct their actions according to reward expectations[1]. However, developing robust estimates of future reward is complicated by the fact that real-world settings are often inherently unpredictable. To act well in face of this unpredictability, the brain tracks and employs higher-order variables such as expected uncertainty[2–6]. Nevertheless, the temporal dynamics by which these variables are computed in the brain remain poorly described, largely due to the field's reliance on non-invasive neuroimaging techniques that are spatially or temporally coarse. To address this limitation, we leveraged intracranial electroencephalography (iEEG) to probe the neural evolution of reward and risk computations at high spatial and temporal resolution.

Reward prediction error (RePE) signals that quantify the discrepancy between observed and expected reward[7,8] not only offer a means of iteratively improving reward estimates[9], but importantly also provide a quantity to describe outcome variability (i.e., risk)[10]. Analogous to RePE, it has been suggested that risk prediction error (RiPE) signals are computed by the brain when predictive associations between choices and outcomes change[5,11]. To explicitly investigate relationships between RePE and RiPE, we employed a normative model following from[4,12] that describes RiPE as a second-order uncertainty term which is derived as a function of RePE, therefore making an explicit temporal prediction that neural signals reflecting reward error should precede and predict those supporting risk error computations.

However, little is known about the temporal characteristics of reward and risk representations at the wide spatial scale for which correlates of these decision variables have been reported in the human brain[13,14]. Studies using scalp EEG from us[15] and others[16,17] have begun to probe the neural correlates of reward and risk at finer timescales, with evidence suggesting a mixture of both parallel and sequential computations across the brain[15]. In support of this, an emerging line of work has begun to describe possible modes of temporal configuration

[1]Division of the Humanities and Social Sciences, California Institute of Technology, Pasadena, CA 91125, USA. [2]Department of Neurosurgery and Brain Repair, University of South Florida, Tampa, FL 33606, USA. [3]Department of Neurosurgery, University of Iowa Hospitals and Clinics, Iowa City, IA 52242, USA. [4]Department of Radiology, University of Iowa Hospitals and Clinics, Iowa City, IA 52242, USA. [5]Iowa Neuroscience Institute, University of Iowa Carver College of Medicine, Iowa City, IA 52242, USA. [6]Department of Neurosurgery, University of Nebraska Medical Center, Omaha, NE 68198, USA. [7]Computation and Neural Systems, California Institute of Technology, Pasadena, CA 91125, USA. ✉e-mail: vman@caltech.edu

by which cognitive processes are carried out in the brain. For example, evidence suggests that both parallel and sequential temporal arrangements support the encoding of visual features and their subsequent maintenance, respectively[18], and similar interactions between parallel and sequential processing modes have been described in the context of multi-tasking[19,20]. Evidence of sequential processing in the reward domain has also highlighted a posterior to anterior flow of information[15,21]. However, studies using scalp EEG provide limited spatial resolution and are unable to probe deeper subcortical regions[22], thereby painting an incomplete spatio-temporal mapping between theoretically grounded variables and neural function.

We aimed to expand our understanding of the temporal dynamics of reward and risk processing in the brain by directly testing for interactions in the neural representations of pertinent variables. We investigated two theoretically motivated hypotheses: that reward prediction error should precede risk prediction error, and that reward prediction error representations should directly relate to risk error computations given that RePE serves as input to the computational function that defines RiPE (see "Computational analysis" for details). Importantly, evidence of an explicit correspondence between neural representations of different variables in a given brain region would lend support for that region's role in the computational transformations mid-stream. This hypothesis stands in contrast to the alternative possibility of distinct representational codes for different computational variables in the same neural population (e.g. ref. 23).

We also held strong prior hypotheses about candidate regions that would represent reward and risk variables. Within the prefrontal cortex, anterior (frontal pole) and ventromedial PFC (vmPFC) have differential contributions to risk[24,25] and reward[26,27], respectively[28]. We further hypothesized that the amygdala would be important given its embedding in a reward circuit along with vmPFC[29] and OFC[30], and by our own previous work in human electrophysiology highlighting its role in reward representations[31,32]. We further predicted that OFC and anterior insula in particular would be candidate substrates for holding representations across reward and risk domains. However, the temporal ordering of computational representations in OFC remains to be clarified given evidence of both simultaneous[33] and sequential[34] encoding of expected value and risk. Similarly, despite evidence of RePE[35,36] and RiPE[4,17,37] signals in anterior insula, to date no studies have directly compared the relative timings nor tested for interactions between these variables in anterior insula.

We recorded human iEEG in a sample of patients undergoing chronic evaluation for epilepsy. Patients performed a task designed to decouple the rapid unfolding of both reward and risk variables first described by[4,12], allowing us to investigate the neural mechanisms by which reward-related representations contribute to building risk computations. iEEG is well situated to overcome many of the spatio-temporal limitations of noninvasive neuroimaging[38] and has been used to elucidate functional contributions of neural activity both within specific regions (e.g. refs. 33,39) and across wide-spread areas of the brain[40,41]. We leveraged this latter advantage of iEEG by simultaneously recording from frontal, subcortical and parietal areas to investigate the temporal organization of reward and risk encoding across widespread brain networks. Among the locations from which we were able to record the intracranial potential, we focused a priori on a smaller set of hypothesized ROIs while also exploring a wider span of regional targets. Our objectives were to address outstanding questions about the temporal signature of reward and risk encoding at scales unattainable with non-invasive human neuroimaging, and to test directly for neural interactions between computations.

We report wide-spread outcome processing that unfolds along the anteroposterior axis of the brain. In contrast, expected value can be decoded from multiple regions simultaneously, whereas error signals exhibit a mixture of sequential and parallel processing. We highlight the computational contribution of anterior insula in generalizing

between RePE and RiPE. Together our results emphasize the utility of uncovering temporal dynamics in the human brain for understanding how reward and risk computations unfold.

## Results

To understand how the brain temporally organizes reward and risk representations, we combined iEEG recordings in 10 patients across 16 recording sessions, all of whom completed a task designed to elicit an array of reward and risk computations[12]. All patients were recruited on the basis of clinical considerations only and were implanted with multi-contact subdural grid and/or depth electrodes to enable chronic evaluation for the treatment of refractory epilepsy. Electrode placement was determined by the clinical requirements specific to each patient; as such the number of contacts varied across individuals. We thus pooled all recording contacts across patients using a pseudo-population approach[42] to sample from our regions of interest (see individual subject coverage in Fig. S1). In total 428 contacts were included in the analyses across all ROIs. The number of contacts within each ROI is shown in Supplementary Table 1.

On a trial, participants guessed whether the second of two sequentially-drawn playing cards would have a higher or lower numerical value than the first. After the second card was presented, participants had all the necessary information to determine the accuracy of their initial guess. Correct guesses earned game points and incorrect guesses resulted in point deductions. After each trial, participants reported whether they had won (Fig. 1A). The deck was shuffled and cards were replaced between trials. The task was designed in conjunction with a normative model of reward and risk[4,12], allowing for the calculation of expected value (EV), outcome (OUT), expected risk (E.Risk), reward prediction error (RePE), and risk prediction error (RiPE) (Fig. 1B). These variables are computed sequentially over the course of a single trial, based on information provided by the cards drawn. After the second card is drawn, participants can compute RePE by comparing the actual outcome with EV and compute RiPE in terms of variance around E.Risk, thereby taking as input the RePE (see "Computational analysis"). Critically, each trial provides an independent sample of these variables, allowing us to employ single-trial decoding to unpack locally distributed multivariate representations.

### Behavior reflects knowledge of task structure and control for learning confounds

Our objective was to characterize the temporal properties of the neural representations underlying reward and risk-related computations. We aimed to study how multiple brain regions encoded these variables free from the potentially confounding effects of learning dynamics on neural activity. As such, we used a task that constrained the computational process within a single trial and did not elicit learning dynamics between trials. The task was designed such that the only optimal behavior was to report correctly at the end of each trial, and there was no optimal guessing strategy given that the deck was reset on each trial.

We found evidence that participants indeed attended to the information provided by the presented cards, as demonstrated by strong reporting accuracy significantly above chance in our sample (mean = 91.50%, s.e.m. = 2.84%, $\beta = 0.415$, 95 CI [0.358, 0.471], $t(9) = 14.398$, $p = 1.184e{-}7$; Fig. 2A). There was no effect of trial count on reporting accuracy ($\beta_t = -1.272$, 95 CI [−2.672 0.127], $z = -1.782$, $p = 0.075$) and the accuracy remained high even in the late trials (mean = 88.4%, s.e.m. = 3.96%). In all subsequent neural analyses we controlled for trials with inaccurate reports across the entire span of the task. Further, participants did not show any evidence that they adopted behavioral strategies commonly seen in learning tasks such as win-stay/lose-shift ($\beta_{OUT} = -0.103$, 95 CI [−0.220 0.014], $z = -1.719$, $p = 0.086$; Fig. 2B), nor did we find evidence that participants adopted other behavioral heuristics such as sticking to the same guess ($\beta_{t-1} =$

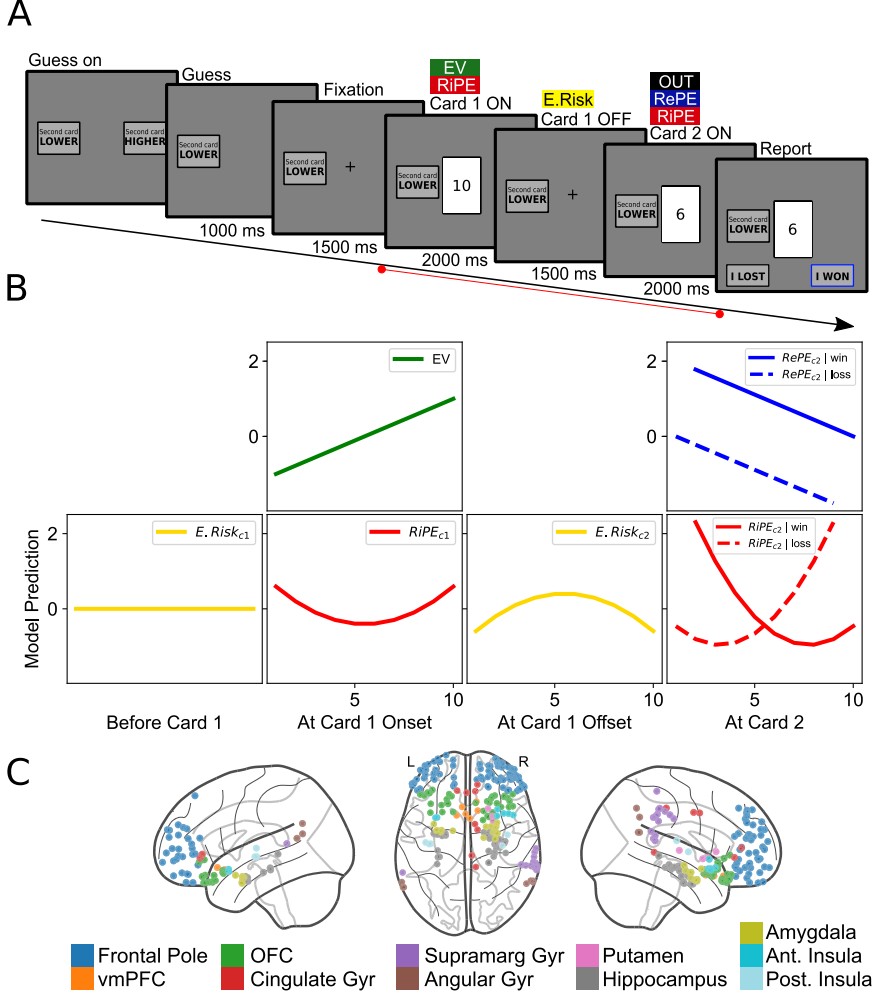

**Fig. 1 | Experimental paradigm and computational model. A** Overview of a single trial in the task completed by participants. Distinct trial events are labeled above, and each event duration is given below, the panels. The within-trial period used in subsequent neural analyses is denoted by the red line at the bottom, and a priori windows for each computational variable are overlaid in the colored boxes. Neural analyses were conducted in epochs from −200 to 500 ms around respective trial event onsets. In the task, playing cards are shown; each card is represented in this figure by their corresponding number. **B** Variation in computational variables as a function of the drawn number of card 1, conditioned on an initial guess that card 2 will be lower as illustrated in the example trial depicted in (**A**). Each column anchors at a trial event for the events analyzed in the study (the period before Card 1 is not included in the neural analysis but depicted here for completeness). Model

predictions of the computational variables are shown at each respective trial event. Reward-related variables including expected value (EV; green) and reward prediction error (RePE; blue) are shown in the top row. Risk-related variables including expected risk (E.Risk; yellow) and risk prediction error (RiPE; red) are in the bottom row. Variables after the onset of card 2 are conditioned on the experienced outcome (win or loss). **C** Contact locations across the pseudo-population. Each dot represents a bipolar sensor located at a projected coordinate between the paired contacts' locations. Dots are colored by their assignment to each atlas-defined region of interest (ROI). Copyright (c) 2007–2023 The nilearn developers. All rights reserved. https://github.com/nilearn/nilearn. Source data are provided as a Source Data file.

−0.092, 95 CI [−0.346 0.162], $z = 0.709$, $p = 0.478$; $\beta_{t-2} = 0.005$, 95 CI [−0.235 0.245], $z = 0.039$, $p = 0.969$) or left/right response key ($\beta_{t-1} = 0.181$, 95 CI [−0.488 0.851], $z = 0.531$, $p = 0.595$; $\beta_{t-2} = 0.166$, 95 CI [−0.270 0.602], $z = 0.746$, $p = 0.456$; Fig. 2C).

Finally, given our interest in the underlying computations that emerge over the span of a trial, we tested whether computations on previous trials might have any influence on participants' guesses on the following trial. We expected there to be no effect of computational variables on the guess choice of the next trial if participants truly treated each trial independently. Critically, this analysis allowed us to carefully test the contribution of reward and risk error on the next trial's choice, where a lack of significant prediction would suggest that we succeeded in isolating the task from learning dynamics. Participants' behavior was analyzed using logistic regression to assess whether guesses were predicted by reward and risk features of the previous trial, as index by the level of each of our variables of interest. We found that participants' guesses were independent of their past experience both in terms of

reward ($\beta_{EV} = -0.104$, 95 CI [−0.311 0.104], $z = -0.980$, $p = 0.327$; $\beta_{RePE} = -0.010$, 95 CI [−0.173 0.153], $z = -0.122$, $p = 0.903$) and risk ($\beta_{ERisk} = 0.138$, 95 CI [−0.184 0.461], $z = 0.842$, $p = 0.400$; $\beta_{RiPE} = 0.159$, 95 CI [−0.096 0.415], $z = 1.225$, $p = 0.220$; Fig. 2D; see "Computational analysis" for their definition). Again, no behavioral strategy was necessary to perform optimally given the task design; these results along with their high reporting accuracy provide evidence that participants understood the nature of the task well, and paid attention to both their guesses and the information presented through the span of a trial to report their outcome accurately. Importantly, this allowed us to proceed with confidence to the neural analyses which relied on the derived computational variables.

### Decoding computational variables with multivariate neural signals
We aimed to investigate the coordination of reward and risk representations across multiple brain regions. Using a multivariate

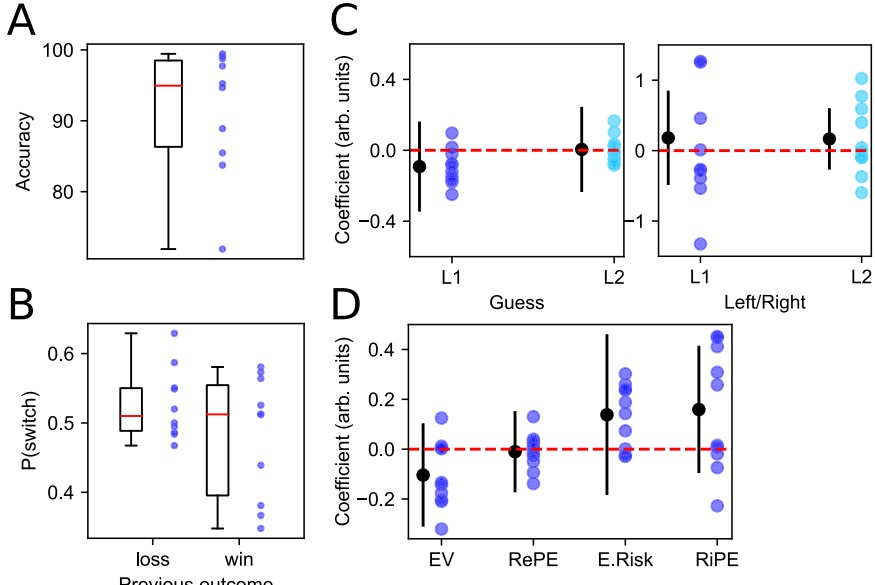

**Fig. 2 | Behavior reflects understanding of task structure. A** Overall average report accuracy across the task. Boxes depict the interquartile range (25th to 75th percentile) and median (red line). Whiskers extend from the minimum to maximum report accuracy across participants (blue dots). **B** Proportion of trials in which participants switched their guess as a function of the outcome on the previous trial. No effects are significant, suggesting that participants ($n = 10$) understood the task and did not adapt any heuristic strategy (e.g., win-stay-lose-shift) in their guess. Boxes depict the interquartile range (25th to 75th percentile) and median (red line). Whiskers extend from the minimum to maximum proportion of guess switches across participants (blue dots). **C** Guess choice (left) and left vs. right response (right) auto-regressive model coefficients. L1 and L2 denote a lag of 1 and 2 trials back, respectively. Black dots and bars denote the mean fixed-effect coefficients and 95 CI around coefficients, respectively, and colored dots are per-subject random effects. **D** Regression coefficients for the logistic model of predictors of guess choice. Black dots are mean fixed-effect coefficients and error bars depict 95 CI around coefficients. Colored dots depict per-subject random effects. Along with (**C**), no predictors are significant indicating that participants ($n = 10$) understood the structure of the task and were treating each trial independently. Source data are provided as a Source Data file.

decoding approach, we examined the timing of each computational variable's representation within a region of interest (ROI). We took a multivariate analytic approach given our objective of describing neural representations of complex computational variables. We reasoned that the information pertaining to these reward and risk computations were likely to involve multiple dimensions of neural representation given the multifaceted nature of the computational information[43], as such, a multivariate approach examining how information is coded across spatially distributed sites within a ROI would provide greater sensitivity[44,45] to uncover neural representations and their temporal properties.

Our design matrix for single-trial decoding allowed us to assess the decodability of each computational variable at its respective point of occurrence within a trial. While this allowed us to leverage the sequential structure of the task to temporally separate variables, it constrained our investigation of temporal organization to within the specified time windows respective to each variable. We statistically orthogonalized variables occurring at the same point in a trial. Note that while the computational model describes an expected risk term twice (before and after card 1 is shown; see "Computational analysis"), because expected risk before card 1 did not vary across trials it was not included in the decoding analysis. Consequently all references to E.Risk in the results describe E.Risk after the onset of card 1 and before the onset of card 2 (i.e., $\text{E.Risk}_{\text{card1Off}}$). Similarly risk prediction error (RiPE) in our decoding results refer to the RiPE computed after the onset of card 2 ($\text{RiPE}_{\text{card2}}$) due to our process of ensuring computational specificity by orthogonalizing variables occurring at the same trial event (see "Feature preprocessing"). To ensure unique decoding of each variable of interest, we regressed out all other variables and covariates (e.g., per-trial report accuracy) from each neural feature. We then extended our decoding approach across ROIs spanning cortical and subcortical regions, focusing primarily in our key regions of interest

but also conducting an exploratory investigation into a larger set of potentially relevant ROIs. Because our analytic approach compares across ROIs in terms of respective timings, we took care to ensure our neural signals were spatially precise and reflected unique sources of information. To do so, we used a bipolar referencing scheme in which pairs of adjacent contacts were subtracted to reflect a single source (see Fig. S1) alongside an ICA-based denoising method which removed distal noise. These processing approaches also increased the signal to noise ratio of our neural features, and together facilitated the comparison of decoding results across regions in order to identify temporal patterns of representation at a wide scale across the brain.

### Distinct modes of temporal organization between value-based computations

We hypothesized that the timing of decoding need not respect reward and risk domain boundaries, but could instead be organized in a manner consistent with the nature of the computation. Specifically, EV requires integrating over each remaining card's reward likelihood after card 1 whereas OUT representations rely on exogenous information given when card 2 is shown, which in conjunction with the initial guess and card 1 determines trial success. Thus one possibility is that of sequential processing for outcome computations as the external information provided by card 2 is used in multiple ways to realize the trial's outcome.

Consistent with a perspective highlighting differences in what the brain needs to do to compute each respective variable, we found different temporal configurations in the decoding patterns for EV and outcome across regions. We were able to decode EV in our hypothesized regions of vmPFC ($C_{\text{AUC}} = 0.725$, $C_{\text{thresh}} = 0.543$, $p = 0.032$, $\text{AUC}_{0.096} = 0.576$, 95 CI [0.517 0.605]) and amygdala ($C_{\text{AUC}} = 0.471$, $C_{\text{thresh}} = 0.234$, $p = 0.022$, $\text{AUC}_{0.070} = 0.564$, 95 CI [0.525 0.592]) (Fig. 3A) in temporally overlapping windows (vmPFC [0.040–0.142 s];

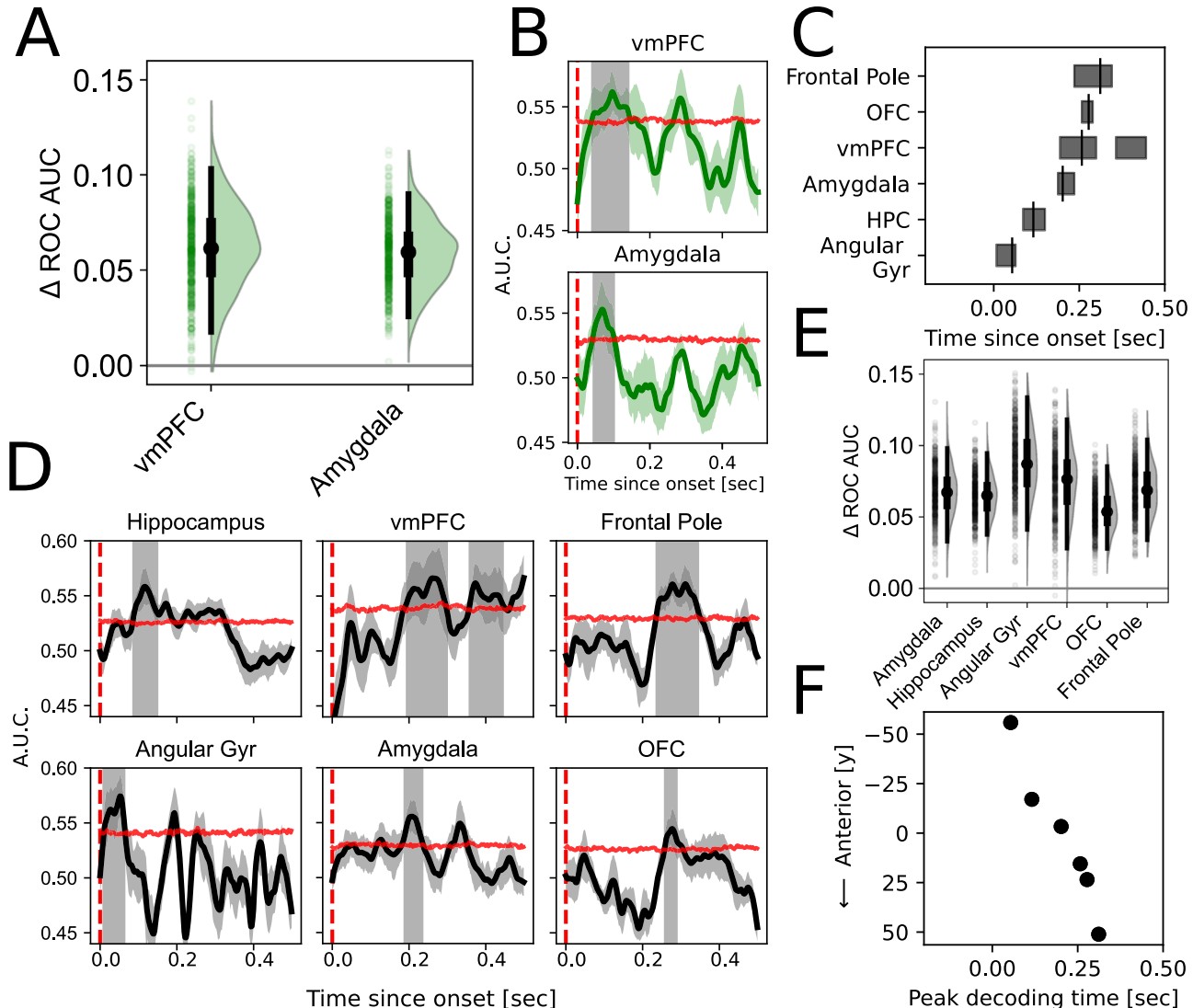

**Fig. 3 | Parallel versus sequential temporal organization for value-based representations. A** Uncertainty estimates around peak decoding accuracies. To compare across ROIs, differences between decoding accuracy and the median (50th percentile) of the permutation-based null distribution, respective to the decoding analysis of each ROI, is depicted. Points and the violin plots show the distribution of differential bootstrapped test accuracies against chance. Box plots depict the median (black point), interquartile range (thick bar), and 95 CI (thin bar) of the bootstrapped distribution ($n = 500$ bootstraps). **B** Decoding of expected value across contacts in each ROI, after the onset of card 1. Lines depict mean cross-validated receiver operating characteristic (ROC) area under the curve (AUC), and error bands depict SEM across folds. Horizontal red lines depict the 95th percentile of the permuted null distribution at each time point, and periods of statistical significance are shown in the shaded gray region (cluster corrected FWE < 0.05).

**C** The timing of significant outcome decoding within each ROI follows a temporally cascading pattern. Horizontal boxes depict periods of significant decoding and vertical bars indicate the timepoint of maximum significant ROC AUC. **D** Outcome decoding follows a sequential temporal organization across regions. Same as (**B**) but for outcome decoding in the period after card 2 is presented, when the participant has all information to know if their guess was correct and consequently if they won on that trial. Lines depict mean cross-validated ROC AUC, and error bands depict SEM across folds. **E** Same as (**A**) but for outcome decoding confidence ($n = 500$ bootstraps). **F** Outcome decoding exhibits both temporal and spatial structure. Peak decoding times across ROIs follows a trend along a posterior to anterior axis as indexed by the y-plane centroid of contacts respective to each ROI. Source data are provided as a Source Data file.

amygdala [0.044–0.102 s]) after the onset of card 1 (Fig. 3B). In an exploratory analysis, we were also able to decode EV in the hippocampus with two temporal clusters (early: $C_{AUC} = 0.382$, $C_{thresh} = 0.239$, $p = 0.025$; $q = 0.117$, $AUC_{0.100} = 0.559$, 95 CI [0.528 0.592]; late: $C_{AUC} = 0.428$, $C_{thresh} = 0.239$, $p = 0.023$; $q = 0.117$; Fig. S2A), with an earlier cluster overlapping in decoding timing [0.082–0.138 s] when compared to vmPFC and amygdala.

Whereas EV decoding was significant in temporally overlapping extents across relevant ROIs, we observed a temporal cascade in periods of significant outcome decoding across regions of the brain (Fig. 3C). We were able to decode outcome in a widespread set of regions across the brain, both in regions included in our a priori

targeted ROIs, and those we surveyed in an exploratory analysis (Fig. 3D). Ordered from latest to earliest decoding time relative to the onset of card 2, we found significant outcome decoding in all prefrontal cortex ROIs including the frontal pole ($C_{AUC} = 1.283$, $C_{thresh} = 0.216$, $p < 0.001$, $AUC_{0.311} = 0.572$, 95 CI [0.532 0.606]), OFC ($C_{AUC} = 0.289$, $C_{thresh} = 0.203$, $p = 0.020$, $AUC_{0.277} = 0.555$, 95 CI [0.526 0.586]), and in two distinct temporal clusters in vmPFC (early: $C_{AUC} = 0.887$, $C_{thresh} = 0.457$, $p = 0.011$, $AUC_{0.257} = 0.573$, 95 CI [0.526 0.619]; late: $C_{AUC} = 0.607$, $C_{thresh} = 0.457$, $p = 0.031$). Consistent with our expectations based on prior work[31,32] we were also able to decode outcome in the amygdala ($C_{AUC} = 0.504$, $C_{thresh} = 0.296$, $p = 0.020$, $AUC_{0.202} = 0.564$, 95 CI [0.531 0.599]). In our set of exploratory ROIs we

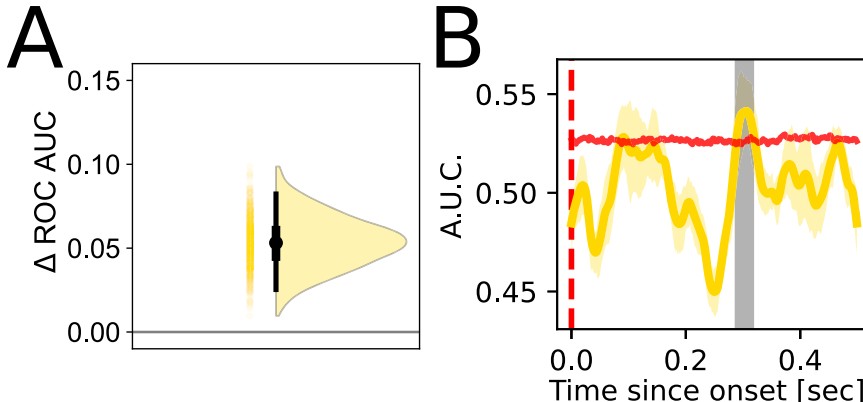

**Fig. 4 | Expected risk can be decoded from the distributed contacts in orbito-frontal cortex. A** Uncertainty estimates around peak decoding accuracies, same as in Fig. 3A, but for E.Risk. Points and the violin plots show the distribution of differential bootstrapped test accuracies against chance. Box plots depict the median (black point), interquartile range (thick bar), and 95 CI (thin bar) of the boot-strapped distribution ($n = 500$ bootstraps). **B** Timecourse of expected risk

decoding in OFC. Lines depict mean cross-validated ROC AUC, and error bands depict SEM across folds. Horizontal red lines depict the 95th percentile of the permuted null distribution at each time point, and periods of statistical significance are shown in the shaded gray region (cluster corrected FWE < 0.05). Source data are provided as a Source Data file.

could decode outcome in both hippocampus ($C_{AUC} = 0.771$, $C_{thresh} = 0.271$, $p = 0.002$, $q = 0.007$, $AUC_{0.116} = 0.567$, 95 CI [0.536 0.596]) and angular gyrus ($C_{AUC} = 0.704$, $C_{thresh} = 0.271$, $p = 0.002$, $q = 0.007$, $AUC_{0.054} = 0.597$, 95 CI [0.540 0.636]) (Fig. 3E). We found evidence that outcome encoding was spatially distributed in Frontal Pole (see Supplementary Materials and Fig. S9).

Strikingly, we observed a spatiotemporal pattern in which the most posterior regions were decoded earliest, and this progressed sequentially to the most anterior region (Fig. 3F). Here we characterized the anteroposterior location of each ROI by computing a projected centroid between the contacts respective to a given ROI, along the coronal ($y$) plane. We further tested whether there was a significant difference in the timing of decoding between pairs of ROIs, ordered by the relative latency of the peak of their respective decoding accuracy curves. For regions that exhibited two significant temporal clusters, we tested for differences in decoding latencies by using the most temporally proximal cluster with respect to pairwise ROIs. We found significant differences in timing between all pairs of temporally (and thus spatially) proximal ROIs, with outcome decoding at staggered latencies across regions: angular gyrus [0.008–0.064 s; $y = -56$], hippocampus [0.086–0.150 s; $y = -17$] ($U = 957$, $p = 2.44e-18$), amygdala [0.188–0.236 s; $y = -3$] ($U = 825$, $p = 5.73e-17$), and vmPFC [0.194–0.299 s; $y = 15$] ($U = 1108$, $p = 8.33e-7$). While the period of outcome decoding in OFC [0.257–0.289 s; $y = 23$] was significantly shifted later relative to vmPFC ($U = 688.5$, $p = 8.14e-4$), and earlier than frontal pole [0.236–0.345 s, $y = 51$] ($U = 620.5$, $p = 0.030$), its temporal span fully overlapped with both neighbors despite their peak decoding latencies respecting this anteroposterior gradient. In further analyses, we clarify that the late-epoch outcome decoding in Frontal Pole is driven by sub-region 9m, a relatively dorsorostral aspect of the ROI (Figs. S8B and S9C).

**Decoding of expected risk in OFC**

We particularly hypothesized that we would be able to decode E.Risk in the anterior insula and prefrontal regions such as OFC given prior work implicating the role of these regions in uncertainty computations[4,33,34]. We also had strong temporal hypotheses of when in the trial E.Risk representations would be held: specifically in the period after the offset of card 1 (but before the onset of card 2) given that the computation of E.Risk takes as input the information from EV, which itself is computed once card 1 is shown. While we were able to decode E.Risk in OFC ($C_{AUC} = 0.270$, $C_{thresh} = 0.206$, $p = 0.028$, $AUC_{0.293} = 0.551$, 95 CI [0.525 0.585]; Fig. 4A) in the predicted period after card 1 is offset

[0.287–0.317 s] (Fig. 4B), we did not find significant decoding of E.Risk in anterior insula ($C_{AUC} = 0.043$, $C_{thresh} = 0.296$, $p = 0.397$). In our exploratory analysis we were able to decode expected risk in posterior insula ([0.102–0.154 s], $C_{AUC} = 0.456$, $C_{thresh} = 0.302$, $p = 0.022$; $q = 0.103$, $AUC_{0.122} = 0.574$, 95 CI [0.526 0.611]) and in the supramarginal gyrus of the intraparietal lobule ([0.449–0.499 s], $C_{AUC} = 0.362$, $C_{thresh} = 0.234$, $p = 0.016$, $q = 0.103$, $AUC_{0.457} = 0.564$, 95 CI [0.526 0.602]; see Fig. S2B).

**Error computations across domains share a mixture of temporal configurations**

We observed direct evidence of a mixture of parallel and sequential temporal ordering when we examined reward prediction error (RePE) and risk prediction error (RiPE) representations. We found significant decoding of RePE in OFC ($C_{AUC} = 0.353$, $C_{thresh} = 0.190$, $p = 0.012$, $AUC_{0.287} = 0.554$, 95 CI [0.519 0.582]) and anterior insula (early: $C_{AUC} = 0.382$, $C_{thresh} = 0.314$, $p = 0.049$, $AUC_{0.293} = 0.562$, 95 CI [0.526 0.598]; late: $C_{AUC} = 0.379$, $C_{thresh} = 0.314$, $p = 0.033$, $AUC_{0.397} = 0.560$, 95 CI [0.519 0.601]; Fig. 5A), consistent with our hypotheses implicating both regions in representing this computational term. We observed that RePE representations extended to the posterior insula [0.269–0.299 s] (Fig. S3A), which we did not a priori predict to be implicated in this computation. Though the cluster in posterior insula did not survive correction ($C_{AUC} = 0.425$, $C_{thresh} = 0.309$, $p = 0.025$, $q = 0.091$, $AUC_{0.291} = 0.573$, 95 CI [0.529 0.615]), it overlapped in timing with the period of significant decoding in OFC [0.273–0.327 s] and in the anterior insula with an early partially overlapping cluster [0.277–0.307 s] (relative to OFC: $U = 160$, $p = 0.941$; Fig. 5B), which was followed by a later non-overlapping temporal cluster [0.339–0.405 s] (relative to OFC: $U = 952$, $p = 2.86e-18$; Fig. 5C). Interestingly, in an exploratory analysis we found significant early decoding of RePE in cingulate gyrus [0.002–0.078 s] ($C_{AUC} = 1.162$, $C_{thresh} = 0.288$, $p < 0.001$, $q = 0.011$, $AUC_{0.030} = 0.589$, 95 CI [0.555 0.626]; Fig. S3A) which taken together with the decoding profiles in OFC, anterior insula, and posterior insula exhibit a mixed pattern of early decoding in one region followed by a multi-regional set of simultaneous RePE decoding (cingulate relative to OFC: $U = 1092$, $p = 1.71e-19$).

We held strong hypotheses that we would be able to decode risk error in the anterior insula, given prior work using the same experimental paradigm[4,17]. Consistent with this hypothesis, we were able to decode RiPE in anterior insula ($C_{AUC} = 0.378$, $C_{thresh} = 0.239$, $p = 0.016$, $AUC_{0.493} = 0.540$, 95 CI [0.511 0.564]) and OFC ($C_{AUC} = 0.436$, $C_{thresh} = 0.156$, $p < 0.001$, $AUC_{0.222} = 0.543$, 95 CI [0.522 0.566]; Fig. 5D),

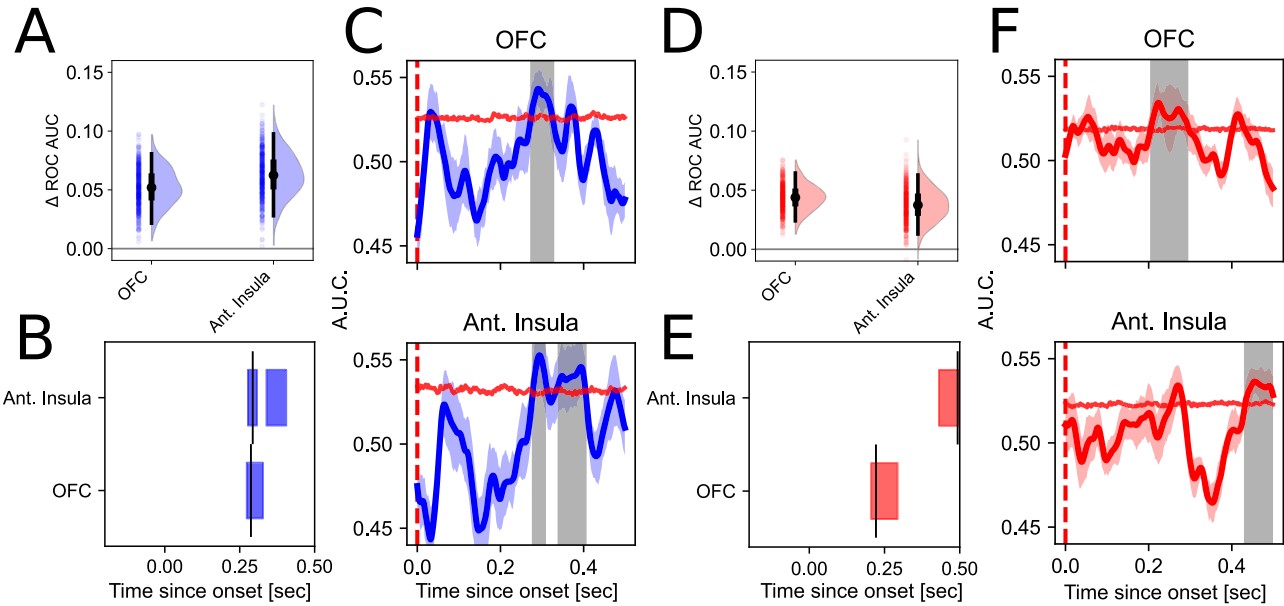

**Fig. 5 | Reward error and risk error decoding share temporal structure.** Decoding curves and periods of significance for reward prediction error (**A**–**C**; RePE) and risk prediction error (**D**–**F**; RiPE). In (**C**) and (**F**) lines depict mean cross-validated ROC AUC, and error bands depict SEM across folds. In both RePE and RiPE we see a mixture of cascading and parallel decoding (see also Fig. S3). Timing of

RePE (**B**) and RiPE (**E**) decoding periods across ROIs. RiPE decoding is shifted later with respect to RePE in anterior insula. Panel descriptions follow that described in Fig. 3, with (**A**, **D**) depicting n = 500 bootstraps. Source data are provided as a Source Data file.

the same pair of regions for which we could decode RePE, as well as the putamen in an exploratory analysis ($C_{AUC}$ = 0.438, $C_{thresh}$ = 0.316, p = 0.024, $AUC_{0.449}$ = 0.549, 95 CI [0.517 0.584]; Fig. S3B). Moreover, we hypothesized that the timing of decoding for risk error should occur later than that for RePE, given that in the computational formulation the risk error (RiPE) term takes as input the RePE (see "Computational analysis"). Consistent with this prediction, we were able to decode RePE earlier in the epoch after card 2 compared to RiPE using activity from the anterior insula. Further, the temporal pattern of risk prediction error decoding across ROIs followed a similar configuration characterized by early decoding in one region, here for RiPE in OFC ([0.206–0.293 s]; followed by late representation across multiple regions including anterior insula ([0.431–0.499 s]; timing relative to OFC: U = 1620, p = 7.68e−24; Fig. 5E, F), and the putamen [0.429–0.499 s], with no differences in timing between the latter two regions (U = 647.5, p = 0.425).

Our analyses of error computations across reward and risk domains revealed a further temporal configuration, in which we were able to decode multiple computational variables in overlapping periods within OFC. In this region we were able to decode all computational variables designed experimentally to arise in the period after the onset of card 2: outcome (Fig. 3D), RePE (Fig. 5C), and RiPE (Fig. 5F), with overlapping temporal extents [0.273–0.289 s]. However, in following analyses we found across variables that distinct sub-regions of OFC drove the reported decoding effects (see Supplementary Materials and Fig. S9), suggestive of intra-regional parallel processing of multiple computational variables during the same period, across sub-regions.

**Anterior insula representations of reward error can predict risk error**

Among regions of theoretical interest, we predicted that the OFC and anterior insula would stand out to be of particular relevance for representing multiple computations given their established involvement in both reward and risk processes (e.g. refs. 4,33,46). Consistent with this prediction, we were able to decode both reward error (RePE) and risk error (RiPE) in both anterior insula and OFC, despite the two regions being distinguished by the sequential versus simultaneous decoding of

these variables, respectively. Highlighting again the conceptual importance of characterizing the temporal organization of neural representation, here we directly tested for evidence that neural activity in OFC and anterior insula could hold a representation of one input variable for the purpose of a second downstream computation; specifically, whether the representation of RePE might be relevant for decoding RiPE.

We ran a generalized decoding analysis to directly test for the possibility that the neural representation of one input variable (e.g., RePE) can be leveraged for the purpose of computing another downstream variable (e.g., RiPE). We first focused specifically on signals in the anterior insula given prior work emphasizing its role in risk error computations[4,17,37], and the evidence that we were able to decode both variables individually in the anterior insula, in which the timing of RePE decoding preceded RiPE (Fig. 5). Consistent with the idea that the brain needs to first compute the reward prediction error before using this information to compute the risk error, we hypothesized that activity in the time window of RePE representation would be able to predict activity related to RiPE at a relatively later period in the same epoch. We constrained our decoding analysis here to the period in the trial relevant to RePE and RiPE representations (0.200–0.500 s after the onset of card 2) and tested for generalization both across variables (i.e., trained on reward error and tested on risk error) and across time. Indeed, we found evidence of significant cross-validated generalized decoding in anterior insula by which a model trained on RePE in a relatively earlier period [0.200–0.391 s] was able to predict RiPE later in the epoch [0.383–0.499 s] ($C_{AUC}$ = 62.504, $C_{thresh}$ = 28.641, p = 0.014; red cluster in Fig. 6A). Indeed, the temporal extent of significant cross-variable training and testing overlapped with the decoding periods for RePE and RiPE, respectively, as described in the section "Error computations across domains share a mixture of temporal configurations". We continued to find evidence of significant generalization when tested against a more stringent statistical threshold (trained on RePE: [0.200–0.283 s]; tested on RiPE: [0.439–0.489 s]; $C_{AUC}$ = 6.554, $C_{thresh}$ = 0.460, p < 0.001, orange cluster in Fig. 6A), and when we did not constrain the time window for analysis (see Fig. S4).

Our predictions regarding the contribution of simultaneous decoding of multiple variables in OFC were less concrete. On one hand

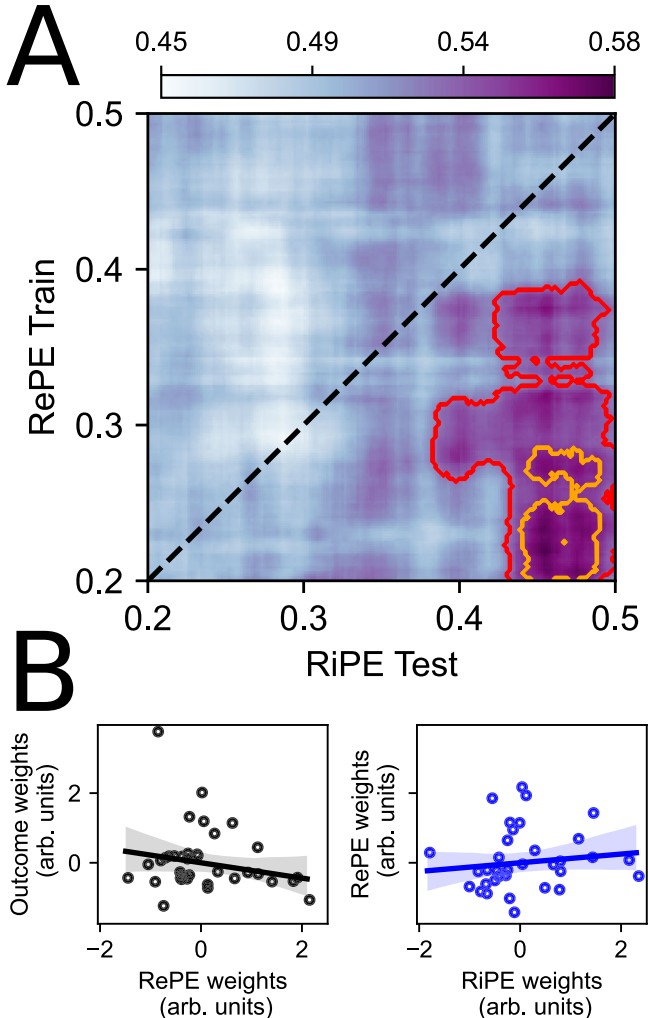

**Fig. 6 | Generalizing and independent representational roles in anterior insula and OFC. A** A model trained on multivariate signal in anterior insula encoding RePE early in the epoch after the presentation of card 2 can generalize across time and computations to decode RiPE later in the same epoch. Significant decoding areas are highlighted by the red ($p = 0.014$; one-sided non-parametric test) and orange ($p = 0.002$; one-sided non-parametric test) contours (cluster-corrected for multiple comparisons; FWE < 0.05). **B** OFC feature weights were not correlated across variables, both those respective to the decoding of outcome and RePE ($r = -0.206$, $p = 0.214$; two-sided non-parametric test) and to the decoding of RePE and RiPE ($r = 0.126$, $p = 0.455$; two-sided non-parametric test). Axes denote the normalized feature weights averaged over the same period for all variables [0.273–0.289 s after the onset of card 2]. Error depicts 95 CI. Source data are provided as a Source Data file.

this parallel temporal configuration could reflect an interaction between RePE and RiPE within the same distributed population, in which case the neural representation for one variable would contribute to decoding another variable. Alternatively, our finding of simultaneous decoding across variables could reflect their separable representations in OFC. In support for this latter alternative, when we conducted the same generalized decoding analysis we did not find evidence of cross-variable decoding at any time point in OFC, both when we trained on outcome and tested on RePE (Fig. S5A) and when we trained on RePE and tested on RiPE (Fig. S5B). We proceeded to directly test the possibility that independently distributed codes within the OFC were implicated in the representation of each computational variable. For a given variable, we characterized the contribution of each OFC contact source to the significant cross-validated decoding scores reported above using a leave-one-feature-out

approach for measuring feature weights, averaging over the period of overlap between outcome, RePE, and RiPE [0.273–0.289 s]. We then tested whether the representational vectors capturing the distributed code in OFC correlated across variables. We found no significant correlation between features weights that encoded outcome and RePE ($r = -0.206$, $p = 0.214$), nor between RePE and RiPE ($r = 0.126$, $p = 0.455$), further supporting the notion of independent representations of outcome, RePE, and RiPE in OFC (Fig. 6B). Indeed, when we investigated sub-regional contributions to computational representations in OFC (see Supplementary Materials), we found that distinct sub-regions of 47o, Area 11, and Area 13 (Fig. S8A) contributed to the parallel decoding of outcome, RePE, and RiPE, respectively (Fig. S9A).

## Discussion

We set out to investigate how the human brain represents a host of computational variables relevant for decision-making in both value and uncertainty domains. Specifically, we aimed to elucidate the temporal properties of neural representations supporting computations of reward and risk expectations and errors. We were able to decode all our computational variables of interest in wide-spread regions. Critically, we distinguished two modes of temporal organization by which computations are represented in the brain. In the reward domain, expected value and reward outcome differed in their temporal profiles with expected value encoded in parallel within a similar time frame across regions, while outcome representations followed a sequential spatiotemporal cascade from posterior to anterior brain regions. Thus, while expected value and outcome are encoded in overlapping distributed value-based networks[13,47,48], these two distinct value-related variables exhibit very different spatiotemporal properties. Moreover, error computations for both reward and risk shared a similar temporal profile reflecting a mixture of sequential and parallel encoding. We demonstrate the importance of elucidating the timing of different types of computational representations by presenting evidence that the distributed code in anterior insula relevant for reward prediction error precedes and directly contributes to decoding risk prediction error. In contrast, representations related to outcome, reward prediction error, and risk prediction error are spatially distinct, but temporally parallel, across subregions of OFC. Together, our findings highlight distinctive patterns of timing by which the brain represents reward and risk variables, as well as potential functional roles of neural representation in regions such as the anterior insula and OFC.

We recorded human neural activity across the brain using intracranial electroencephalography, which is well situated to describe an intermediate level of analysis with broad spatial coverage across the brain[41,49] and relatively high specificity within a region[38]. We took a multivariate decoding approach in which we examined whether the distributed activity across contacts within a specific ROI was able to decode our computational variables of interest, and we applied this approach across multiple regions. We were motivated theoretically by existing work on reward and risk encoding in human iEEG[33,34,39], which had taken an encoding approach and focused predominantly on a single or a small subset of regions. We extend this work not only by exploring a larger set of regions, but importantly by elucidating the temporal organization of reward and risk representation across the brain.

Critically, we leveraged an experimental design in conjunction with a normative model of reward and risk which separated, in time over the span of multiple sequential trial events, when the information relevant for each computational process was available. We could therefore examine specific periods of neural data with respect to each computational variable in the span of a single trial. It is worth noting, however, that our approach of examining specific, hypothesized periods for a given variable constrains our conclusions about the temporal organization of that variable across regions to within our specified

time windows, and does not preclude the possibility of a variable being represented at other periods in a trial. Nevertheless, this approach allowed us to look at the unique decodability of a set of computational variables which are not often decoupled in other experimental contexts. By regressing out variance attributable to other computational variables in each decoding scheme, we further ensured that the neural data going into each decoding model did not carry residual information correlating with variables other than the variable of interest. Moreover, our task structure allowed us to investigate the degree to which neural activity could decode our computational variables at the level of a single trial[50] with high temporal specificity, and allowed us to control for the effect of learning dynamics on neural representations. We could therefore examine isolated, learning-independent computational features.

Our findings of both parallel and sequential modes of temporal organization are consistent with experimental work in other domains of cognitive neuroscience generally[18,20] and in the domain of value-based decision-making specifically[15]. We found a unique temporal cascade in the regions that represented the outcome of each trial in a manner that respected an anteroposterior gradient. Signal in the angular gyrus ROI in the posterior parietal cortex could be used to decode outcome first; this then proceeded sequentially across ROIs to the frontal pole over the period in which all information for knowing the outcome was available. While previous work has reported outcome correlates over widespread regions of the brain[47,51–53], here we exploited the high temporal resolution of human iEEG to show that outcome representations across the brain have a distinct spatiotemporal profile. The gradient-like organization from posterior to anterior regions of the brain is consistent with similar architectures reported across multiple regions for diverse cognitive processes[54–56]; here we show that this topographical arrangement is widespread (from parietal to prefrontal regions) and temporally ordered in the representation of reward outcome. In the context of this task, the outcome (i.e., the accuracy of the initial guess) is unique in that explicit feedback is not presented to the participants. Instead representing the outcome entails processing information conditioned on the guess, card 1, and finally card 2. Exploiting multiple pieces of observable information to derive the resulting outcome could potentially drive this widespread cascade in representation.

Indeed our finding that computations are structured in time across areas raised the question of whether the functional connectivity between different regions encoded computational information. Across all variables, we did not find significant differences in the cross-correlation between regions and high versus low levels of that variable (see Supplementary Materials). However, our approach is only one of several ways to address the question of directed functional connectivity relevant to processing these computations. Future work in human electrophysiology is well-positioned to elucidate the mechanisms driving directed connectivity. For example, cortico-cortical evoked potential manipulations[57,58] directly perturb neural activity and therefore might provide a casual picture of how information propagates across the brain.

Regions in which we were able to decode expected value (EV) showed overlapping temporal spans of significant decoding, after the onset of the first card when enough information was available for participants to form a conditional expectation of the probability of guessing correctly on that trial. Spatially, our findings of EV decoding were consistent with previous work that reported representations of EV in vmPFC[39], amygdala[29,32,59], and hippocampus[60,61]. Further, our findings are convergent with recent work that similarly decoded value-based variables from intracranial electrophysiology data despite different experimental contexts. For example, other work from our group has shown that EV can be significantly decoded in the vmPFC and amygdala in a Pavlovian conditioning task, with similar effect sizes (as determined by relative decoding accuracy[31]). Work from independent

groups have also reported similar significant decoding accuracy levels when decoding value (e.g., likeability) ratings from vmPFC and OFC[62]. Here we extend this previous work by showing that EV is coded in the distributed, multivariate activity within each relevant region, and showing that the code is held in a parallel temporal configuration. The EV term as defined in this context requires holding multiple pieces of information in the form of the reward probabilities associated with each possible second card, and the internal process of integrating over this information. One possible interpretation of simultaneous EV representation across regions is that this reflects the top-down nature of EV computation in which the relevant information (i.e., each possible draw for the second card and their respective likelihood of reward) does not require further external input but relies instead on held knowledge of the instructed task set. Interestingly, this temporal configuration has been discussed as a potentially important neural organization across cognitive domains[63] and scales of analysis[64,65].

We also sought to investigate the potential functional advantages conferred by temporally organized computational representations in the brain. One possibility is that multiple regions might all hold a similar piece of computational information (e.g., reward prediction error), but that different regions use this same information for different purposes. For example, our ability to decode RePE from activity in cingulate cortex rapidly after the onset of card 2 is consistent with the literature on error signals in dorsal anterior cingulate cortex (dACC), both measured with intracranial[66] and scalp[67] EEG. This fast temporal signature, in conjunction with prior work on cingulate encoding of error signals in other domains[66], suggests that the fast decoding of RePE we find in cingulate cortex might reflect a more general error.

In contrast, we show that the encoding of RePE in anterior insula serves a specific functional role that directly relates to later RiPE computations in the same region. We show a relationship between the neural code supporting RePE and RiPE, both indirectly via the shifted temporal profile between RePE and RiPE decoding in anterior insula over the same period, and directly with our generalized decoding analysis in which the neural representation of reward error at an earlier period can significantly decode risk error later. This finding speaks to the notion of neural representations progressively "building" computations, consistent with a line of evidence emphasizing the role of specific neural activity in composing computations from their components[51,52,68]. Here we extend beyond established associations between anterior insula and risk error[4,37] and show that the distributed activity in anterior insula plays a further crucial role in leveraging input computations (i.e., RePE) to decode risk error. In other words, we adopt the notion of neural representations of computations being compositional, and present evidence of this across the value and uncertainty domains. This nonetheless does not preclude the possibility of independent representations of multiple types of computations in non-overlapping distributed activity within a region, as we show in separable representations of outcome, RePE, and RiPE in OFC.

Further highlighting the importance of resolving computational timing at a fast timescale, we were able to decode expected risk later in the trial prior to the onset of the second card consistent with the description of expected reward and risk in our normative model. Our results were also congruent with previous human[34] and animal[68–70] electrophysiology studies on the relative timings between expected reward and risk encoding, in which expected reward is encoded shortly after cue presentation whereas expected risk encoding is evident in activity starting before the period in which the outcome is known. However, the temporal ordering of how multiple regions represent expected risk remains equivocal from our results, due to the lack of representation across multiple ROIs. While we found robust decoding of expected risk in OFC, contrary to our hypotheses we did not find that signal from the anterior insula could decode this variable. This stands in contrast to a previous fMRI study using the same task[4] which reported spatial distinctions between expected risk and risk

error correlates all within the anterior insula. One possible explanation for this discrepancy is that our recording contacts lie along the inferior plane of the anterior insula, more consistent with the reported BOLD correlates of risk prediction error than to that of expected risk. Another possibility speaks to potential discrepancies in the computational information encoded in human electrophysiology versus fMRI, an avenue of research to be explore more thoroughly in future studies.

While invasive electrophysiology confers the advantage of precise timing and high spatial specificity, critical here for isolating temporal effects across wide-spread regions of interest, the limitations of this neural modality need to be carefully considered. Like other work involving populations from which data collection is challenging, our current study is limited by a relatively small sample size and accompanying concerns about generalization. We carefully considered the design of the study to ensure that the task was simple and straightforward enough for patients to understand and complete. For example, our task was paced so that each trial event could be read and understood by the patients, and we collected and statistically controlled behavioral measures of task attention on each trial. We also took efforts during data collection to ensure patients were fully engaged with the task and only administered the task in periods distant from seizures. Finally, our behavioral analyses validated that patients understood the structure of the task (e.g., that trials were independent, which was important for our decoding scheme), and behavioral metrics of task performance (e.g., report accuracy) were similar to that reported in previous literature on healthy controls[12]. Further supporting the inference that the neural circuits implicated in our results generalize to the broader population, it is also worth noting that our findings corroborate reported findings in the literature as discussed above.

The approach of pooling across our patient sample to create a pseudo-population of recording sites follows from other work in invasive electrophysiology[42,62] and allowed us to sample from diverse regions across the brain. As with other studies employing this pseudo-population approach, an assumption underlying the interpretation of our findings is that neural signals actually recorded from separate patients can be treated as if they were recorded from a representative "pseudo-population" brain. Further, it results in different ROIs containing different numbers of measurements and thus differing amounts of explanatory power between decoding models across regions (see also ref. [62]). Specifically, regions that are under-sampled given a fewer number of recording sites may yield non-significant decoding accuracy due to decreased statistical power, especially if the true underlying neural representation is spatially widespread but weakly informative when only surveying each recording site individually[71]. Indeed we chose to aggregate bilateral ROIs with this consideration of increasing our sampling of each ROI, though we conducted additional analyses probing laterality differences in our reported decoding results across ROIs (see Supplementary Materials). Of course, since electrode placement is guided only by clinical considerations, failure to observe an effect can potentially arise in our study simply due to not having electrodes in a region relevant for a particular computation, an inherent limitation to invasive electrophysiology. Therefore, non-invasive imaging techniques that provide greater spatial coverage such as fMRI are an important complement to intracranial electrophysiology. Our findings should be contextualized in light of these considerations, and future work endeavoring to pool across larger samples, for example by acquiring data on the same task across multiple sites, will be important to mitigate these limitations. Larger samples will also be conducive to expanding our findings on temporal organization, for example by validating that a particular representation is specific to the time window we specified for that variable in our analyses, revealing whether the encoding of a variable can extend over the course of multiple periods within a trial, and elucidating phasic versus tonic properties of distinct representations.

Here we provide one picture by which the brain encodes multiple features relevant for value-based decisions, in which neural representations of computational variables are temporally organized and spatially distributed. Our findings also contribute to an emerging literature describing how neural implementation of different reward and risk computations unfold in time interactively. Importantly, while our normative model afforded us the ability to look at computational representations isolated from learning dynamics, further work extending our approach to subjectively computed computational variables, for example through learning or belief updating mechanisms, will enrich our understanding of how the human brain represents the many facets of information important for complex decisions in the world.

## Methods

### Participants

Our sample included patient participants ($n = 10$, 7 female, age range 22–56, mean = 37.70, s.d. = 9.93) who underwent neurosurgical implantation of intracranial depth and surface electrodes to enable chronic evaluation and localization of seizure foci for the treatment of refractory epilepsy. Within our full sample, 6 subjects were implanted only on the right side, 3 were implanted on the left, and 1 was implanted bilaterally. All data were collected in periods free from seizure activity at least 1 h before and after the experiment and data were visually inspected for interictal activity characterized by stereotyped transients in the raw recording. All research protocols were approved by the University of Iowa Institutional Review Board. Subjects provided written consent prior to participation in research and could rescind consent at any point without consequence to their medical evaluation.

### Experimental paradigm

Participants were presented with two cards drawn sequentially and without replacement on each trial. The cards were shuffled from a deck of cards comprising 10 cards (ace to ten) and excluding face cards. Participants were instructed to treat an ace card as denoting "1". The cards were reshuffled after every trial. Prior to the drawing of the first card, participants were instructed to predict whether the second card would have a higher or lower numerical value than the first.

Participants were given up to 7 s to respond. Their guess was displayed on the screen for a duration of 1000 ms, and a pre-card fixation cross was shown for 1500 ms. We then drew the first card which was displayed for 2000 ms before a 1500 ms inter-stimulus period in which only their initial guess remained on the screen. The second card was then shown for another 2000 ms. Upon the presentation of the second card, all information necessary to determine the accuracy of the initial guess was available to the participant. Finally, participants were prompted to report whether they had won (guessed correctly) or lost (guessed incorrectly). Note that the initial guess of the participant remained on the screen throughout the presentation of both cards as well as the report period to reduce memory demands. A schematic of the trial structure and event timings are presented in Fig. 1A. To incentivize participants to respond both to the initial guess and the final report, participants were instructed that they would receive a reward outcome in the form of 10 game points if their guess was correct and a deduction of 10 points from their cumulative total if their guess was incorrect. Correct or incorrect reporting of their guess accuracy at the end of the trial further resulted in gain or loss of 5 points, respectively. While these outcomes were displayed during the initial instructions and training block, during the task period in which data were analyzed these outcomes were not presented to the participant.

### Procedure

Participants remained in the epilepsy monitoring unit of the Epilepsy Center at the University of Iowa Hospitals and Clinics for 2–4 weeks after implantation, under clinical direction. The experiment reported

here was delivered during this monitoring period. The epilepsy monitoring unit included a recording facility in an electromagnetically shielded room. Participants were awake and in a hospital bed or arm chair for the duration of the experiment, including during the delivery of task instructions. Participants were simultaneously provided verbal instructions and written instructions programmed into the task code which comprised static messages describing what the participants needed to do during the task (see section "Experimental paradigm") as well as interactive exercises to reinforce key task details, familiarize participants with the behavioral demands of the task (i.e., guesses and accuracy reports), and elucidate the incentive structure of a trial. Participants responded using left/right keyboard presses for both the guess and report; left/right mappings were randomized on every trial. To motivate participants to attain the most points they could, participants were instructed that they competed against previous players to rank on a leaderboard based on the total accumulated points at the end of the task. After participants were instructed and verbally confirmed they understood the task, they proceeded to complete 1 ($n = 4$) or 2 ($n = 6$) sessions comprising 90 trials per session; the number of completed sessions was based on participant willingness. We thus recorded a total of 16 sessions and 1440 trials across the sample. A subset of the participants ($n = 2$) had previous exposure to instructions and experience with the task as they completed pre-implantation functional MRI scans. The experiment was delivered using the Psychtoolbox-3 toolbox (v3.0.14).

## Recording

We used a combination of depth electrodes comprising low impedance clinical contacts (2.2 mm–10 mm intervals), acquired using the Neuralynx (Bozeman, MT) Atlas system, and simultaneous multi-contact subdural grid electrodes embedded in a silicon membrane (Ad-Tech Medical Instrument, Racine, WI). The electrophysiological data were recorded at a sampling rate of 2 kHz with 24-bit resolution and bandpass filtered during acquisition between 0.1 to 500 Hz.

## Imaging and contact localization

Before and after electrode implantation, we collected high-resolution structural MRI data of the brain for each participant. For the first subset of participants ($n = 5$), pre-implantation structural images were acquired from a 3T GE Discovery MR750w scanner with a 32-channel head coil (T1: FSPGR BRAVO 1.0 mm², slice thickness = 0.8 mm, FOV = 256 mm², TR = 8504 ms, TE = 3.288 ms, FA = 12°, TI = 450). The second subset ($n = 4$) completed pre-implantation structural scans on a 3T GE SIGNA Premier scanner with a 32-channel head coil (T1: FSPGR BRAVO 0.8 mm², slice thickness = 0.8 mm, FOV = 256 mm², TR = 8576 ms, TE = 3.364 ms, FA = 12°, TI = 900). One participant completed their pre-implantation scan on a 1.5T Siemens Avanto with a 32-channel head coil (T1: MPRAGE 1.0 mm², slice thickness = 1.5 mm, FOV = 256 mm², TR = 2200 ms, TE = 2.96 ms, FA = 8°, TI = 450). Patients completed a post-implantation T1 scan on a 3T Siemens Skyra with a T/R head coil (MPRAGE 0.98 mm², slice thickness = 1 mm, FOV = 256 mm², TR = 1900 ms, TE = 3.44 ms, FA = 10°). Preoperative structural MRI images were co-registered to post-implantation structural MRI images, guided by post-implantation computed tomography (CT) scans (in-plane resolution 0.5 × 0.5 mm, slice thickness 1–3 mm), using custom MATLAB (Mathworks, Natick, MA) scripts and affine registration from FSL FLIRT (v6.0)[72], and all images were processed to 1 mm³ isotropic resolution. Visual comparison with intra-operative photographs was conducted to verify accuracy. Electrode contacts were localized with CT-guided post-implantation structural MRI images, then transferred onto the pre-implantation MRI space specific to each participant. Each participant's structural MRI was then co-registered to the MNI template brain using ANTs (v2.3.1)[73], and resulting contact locations are shown against the template (see Fig. S1A) using the Nilearn toolbox[74].

## Preprocessing

Intracranial electroencephalography (iEEG) recordings were pre-processed using custom scripts relying on functions from the MNE-Python (v1.2.3)[75] and scikit-learn (v1.3.0)[76] toolboxes. The data were trimmed to include only periods in which participants were doing the task, downsampled to a sampling rate of 500 Hz (i.e., 1 sample every 2 ms), and line noise was removed using a 60 Hz notch finite impulse response filter. The data were high pass filtered at 1 Hz, entered into an independent components analysis (ICA[77]) denoising step, and then low-pass filtered at 250 Hz. The ICA-based denoising approach involved computing an ICA on the continuous contact timeseries, extracting the absolute value weight matrix from the ICA decomposition, and comparing the distribution of weights over contacts against an uniform distribution using the Kullback-Leibler divergence ($D_{kl}$[78]); given evidence that systematic sources of noise and volume conduction effects can affect multiple contacts and to isolate spatially specific sources[79–81]. Independent components corresponding to weight distributions reflecting uniform distributions, defined with respect to a $D_{kl}$ threshold, were visually inspected and removed from the data. This was done at two-stages: across all contacts on all electrodes within-subject ($D_{kl}$ threshold = 0.2) to detect global sources of noise, and across all contacts along each electrode ($D_{kl}$ threshold = 0.05) to detect local sources of noise permeating across an electrode. Finally, contacts sharing an electrode were re-referenced along bipolar pairs, identified by adjacent contacts from the same electrode, to further isolate local activity around the contact locations and mitigate the effects of volume conduction[79,82]. Bipolar pairs were excluded if both contacts were in white matter as defined by a segmentation of the template brain using FSL's FAST algorithm[83]. The template-space coordinates between bipolar pairs were interpolated for visualization (see Figs. 1C and S1B). The preprocessed data were epoched at within-trial events: the guess response, the onset of card 1, the offset of card 1, the onset of card 2, and the report response. Epochs for all events were defined by a period spanning −200 ms to +500 ms around the event onset, with the period before event onset defined for baseline correction. The epoched bipolar timeseries were entered into all subsequent neural analyses.

## Behavioral analysis

To survey the degree to which participants paid attention and completed the task, we analyzed behavior using mixed-effects regression models to differentiate between within- and between-subject sources of variance, specifying random intercepts and slopes for all terms, for each participant (shown in Wilkinson-Rogers notation). Here we used a threshold for statistical significance as $\alpha = 0.05$ and interpret effects in which the associated $p$ value is below 0.05 as statistically significant. We justify this threshold to remain congruent with the non-parametric statistics reported below for the neural analyses, in which we define the significance cut-off at the 95th percentile of the permuted null distribution.

First, we tested whether their reported accuracy $R$ different significantly from chance (50% accuracy), with accurate responses in the behavioral report data defined as correctly selecting whether they won or lost on a trial:

$$R - 0.5 = 1 + (1 \mid \text{subject}) \qquad (1)$$

Using a mixed-effects logistic regression model, we tested whether report accuracy tended to decrease over the span of the task by regressing the report accuracy onto the trial number (scaled between 0 and 1) within a block. Here and below proportionality represents the log-odds (i.e., logit) function on the dependent variable:

$$p(R = \text{Correct}) \propto t + (1 + t \mid \text{subject}) \qquad (2)$$

To examine whether participants were influenced on each trial by previous computational quantities, we modeled the guess choice on a trial ($G$; whether the second card would be higher or lower) as a function of the computational variables of interest on the previous trial: expected value (EV), reward prediction error (RePE), expected risk (E.Risk), and risk error (RiPE):

$$p(G_t = \text{Lower}) \propto \text{EV}_{t-1} + \text{RePE}_{t-1} + \text{E.Risk}_{t-1} + \text{RiPE}_{t-1}$$
$$+ (1 + \text{EV}_{t-1} + \text{RePE}_{t-1} + \text{E.Risk}_{t-1} + \text{RiPE}_{t-1} \mid \text{subject}) \quad (3)$$

We assessed whether participants' choices were influenced by previous events (e.g., win-stay/lose-shift across high/low guesses). To do so, we re-parameterized their choice on a given trial in terms of switch (different from the guess on the previous trial) or stay, and regressed switch/stay responses onto the outcome of the previous trial:

$$p(G_t \neq G_{t-1}) \propto \text{OUT}_{t-1} + (1 + \text{OUT}_{t-1} \mid \text{subject}) \quad (4)$$

To further test whether participants understood the independent nature of each trial, we constructed models examining guess choice and left/right key press ($K$) stickiness, extending two trials back:

$$p(G_t = \text{Lower}) \propto G_{t-1} + G_{t-2} + (1 + G_{t-1} + G_{t-2} \mid \text{subject}) \quad (5)$$

$$p(K_t = \text{Right}) \propto K_{t-1} + K_{t-2} + (1 + K_{t-1} + K_{t-2} \mid \text{subject}) \quad (6)$$

## Computational analysis

Before the onset of the first card, the prediction about the number on card 1 (and thus the probability of reward given the guess) does not vary with the initial guess, so is denoted following[12] as a constant $P_0$ and excluded from subsequent analyses. Upon the onset of the first card, participants are able to compute the EV: the expectation of the probability of reward conditioned on the value of card 1 and their guess:

$$\text{EV} = E[P(\text{OUT} \mid \text{Card}_1, \text{Guess})] \quad (7)$$

This term can be computed in a normative manner given the relatively limited set size of 10 total values and the knowledge that cards are drawn without replacement, meaning that the second card cannot be the same as the first. For example, if the participant guessed that the second card will be higher and drew a 9 on the first card, it is intuitive that the probability of getting a reward (i.e., that their guess will be correct) will be low given that eight of the nine remaining cards available to draw as the second card are lower than the first. The computation of this EV term also allows for an error term at this stage, defined as the deviation between the EV (the expectation about the probability of reward given the guess), and the constant $P_0$ (the probability of reward given the guess). In practice this error term is perfectly correlated with the EV term since $P_0$ is a constant, and is not included in subsequent analyses.

Note that prior to the onset of card 1 we can formalize an expected risk term. Before participants see the first card, they can already form an expectation about the risk around the eventual expected value by integrating over the squared difference between all possible EV across the 10 values available for card 1 and the constant $P_0$:

$$\text{E.Risk}_{card1} = E[(\text{EV} - P_0)^2] \quad (8)$$

However, as with $P_0$, E.Risk$_{card1}$ does not vary across trials since it is not conditioned on the value of card 1. Nevertheless, it is useful for illustrating the risk prediction error, a higher-order uncertainty defined as the difference between the square deviation from expected value (i.e.,

risk or unsigned prediction error[10]) and expected risk at the onset of card 1:

$$\text{RiPE}_{card1} = (\text{EV} - P_0)^2 - \text{E.Risk}_{card1} \quad (9)$$

We designed the task to include a brief period of 1.50 s in which the first card is offset from the screen but before the second card is shown (Fig. 1A), with the assumption that expectations about the risk around card 2 would be computed in this period[12]. Here we similarly define E.Risk$_{card2}$ in terms of the expectation about the variance/risk around the expected value, where risk is again defined as the squared deviation of the outcome OUT from expected value (i.e., reward prediction error):

$$\text{E.Risk}_{card2} = E[(\text{OUT} - \text{EV})^2] \quad (10)$$

Finally, when the second card is presented, the participant is able to computationally resolve their previous estimates of the probability of reward since the combination of their guess, the value of card 1, and now the value of card 2 provide all necessary information to know the outcome, and thus the computation of reward prediction error:

$$\text{RePE} = \text{OUT} - \text{EV} \quad (11)$$

The RePE accordingly informs the risk prediction error term at card 2:

$$\text{RiPE}_{card2} = \text{RePE}^2 - \text{E.Risk}_{card2} \quad (12)$$

## Region of interest definition

We specified a set of 5 a priori regions of interest (ROI) spanning frontal and subcortical regions given our strong hypotheses of the role of the prefrontal cortex (PFC), insula, and amygdala in value- and uncertainty-based computations, using the Harvard-Oxford (H-O) probabilistic atlas[84] to delineate contacts respective to each ROI. The 5 a priori ROIs included Frontal Pole, OFC ("Frontal Orbital Cortex" in the H-O atlas), and ventromedial prefrontal cortex (vmPFC, labeled "Subcallosal Cortex" in the H-O atlas). We modified the ROI names for OFC and vmPFC from their original labels in the H-O atlas to better speak to relevant findings from previous literature that have used these labels. We also hypothesized that the anterior area of the insula (Insular Cortex) would be particularly relevant for risk representations given findings from prior work[4] using the same experimental paradigm. Since the H-O anatomical mask for the Insular Cortex comprised voxels spanning both anterior and posterior aspects of the insula, we split the anatomical mask at the mid-point of the anterior-posterior (Y) plane ($y = 4.66$) ensuring that all contacts ascribed to anterior and posterior were respectively located on positive and negative coordinates in template space along the Y plane.

Because the Frontal Pole and OFC ROI delineations from the H-O atlas were spatially extensive, and as a result comprised the largest number of included contacts in our sample (see Supplementary Table 1), we investigated whether computational variables that were able to be decoded from the multivariate activity in these regions were driven by particular nested sub-regions of these ROIs. Our analytic approach and the results of this investigation are presented in the Supplementary Materials. We were also interested in exploring whether reward and risk representations could be held in wide-spread regions across the brain outside of our main hypothesized ROIs. We thus included an additional set of 6 ROIs comprising: Posterior Insula, Putamen, Cingulate gyrus, Hippocampus, Angular Gyrus, and Supramarginal Gyrus. The contact locations within each labeled ROI and their corresponding Brodmann area labels are provided with the source data. ROIs were defined anatomically according to the H-O atlas as implemented in FSL and mask boundaries were defined at a voxel

threshold of 10 ($p = 0.1$). Contacts within the boundaries of the thresholded probabilistic maps of multiple ROIs were assigned according to the max probability. Contacts across bilateral ROIs were aggregated for the results reported in the main text, though see Supplementary Materials for an additional investigation into laterality differences.

## Decoding analysis

**Feature preprocessing.** We set up an encoding model to preprocess each of our features (contacts) for the decoding model. At the within-subject level before concatenation across subjects (see section "Pseudo-population design"), for each feature we defined a general linear model (GLM) to regress out covariates of no interest, as well as computational variables aside from the current variable of interest for decoding. The confound design matrix of the GLM importantly included an intercept modeling the average contact activity across trials for that subject; regressing this design out of each neural feature therefore removes subject-specific variance in the neural data in preparation for our pseudo-population decoding analysis. We included indicator vectors for the onset of the cards, the offset of card 1, and mean-centered parametric vectors for respective (i.e., not currently decoded) computational variables: $EV_{card1}$, $OUT_{card2}$, $RePE_{card2}$, $E.Risk_{card1Off}$, and $RiPE_{card2}$ along with the observed risk O.Risk (i.e., squared RePE), aligned to the trial event denoted in the subscripts. Because of the design of the computational model, we were concerned that any effects purportedly related to RiPE might have been driven by its correlation to O.Risk. O.Risk was consequently included in the design matrix to regress its associated variance out of our neural features when decoding RiPE, after orthogonalizing O.Risk and RiPE such that the variance of the RiPE term was only the component unique to RiPE and not shared with O.Risk.

Because of collinearity between $OUT_{card2}$ and $RePE_{card2}$ (Fig. S6B), we orthogonalized the latter latent variable with respect to the former observed variable such that the residual variance in $RePE_{card2}$ was unrelated to $OUT_{card2}$. Likewise, we orthogonalized $RiPE_{card2}$ with respect to $O.Risk_{card2}$ (Fig. S6C). Note that despite RiPE and O.Risk being computed at both card 1 and card 2 onsets (see "Computational analysis"), because $RiPE_{card1}$ and $O.Risk_{card1}$ are perfectly correlated, when we orthogonalized $RiPE_{card1}$ with respect to $O.Risk_{card1}$ the latter retained all the variance and $RiPE_{card1}$ was zeroed out (i.e., no variance retained; see Fig. S6D). The implication is that our RiPE results are only driven by its variance at card 2 ($RiPE_{card2}$; Fig. S6E) whereas variance in O.Risk is driven by both card1 and card2. Finally, the per-trial report accuracy was also included as a covariate, aligned at the report response event. We regressed each feature vector against this set of covariates (excluding the variable of interest for decoding), and replaced each feature by its respective residual vector.

**Pseudo-population design.** For each ROI and participant, we set up an expanded pseudo-population design to create the feature matrix for decoding. First, we concatenated the epoched potentials across trial events (the epoch period and set of 5 trial events are described in "Preprocessing"), resulting in a dimensionality of 251 time points (at 2 ms resolution) by 5 events per trial in the neural data. For participants with 2 recording sessions, we further concatenated the trials across sessions. This was done separately for each contact within an ROI, resulting in a per-subject design matrix of dimensionality: $nEvents \times nContacts \times nTimepoints$. To prepare the feature matrix and target vector for our decoding analyses, we split each participant's data into cross-validation (CV) train/test folds, and separately concatenated the design matrices across all subjects for each fold. This ensured balance in the per-subject contribution to decoding, across CV folds. To create the final pseudo-population matrix, we concatenated across subjects (for each fold) along the $nEvents$ and $nContact$ dimensions for each ROI; this approach avoids the assumption that each trial occurs at the same time across participants and results in a largely sparse feature matrix (see Fig. S7).

**Decoding model and procedure.** The variable of interest to be decoded was binarized into to +1 and −1 around 0 (i.e., the mean) as the target vector for classification analyses. The target vector and feature matrix were split into 10 folds for cross-validation (CV) along the $nEvents$ dimension, and each fold was normalized (i.e., centered and scaled to unit variance) by computing statistics on the training set and applying to both training and test sets[85]. All feature preprocessing (see "Feature preprocessing") and normalization was conducted prior to the between-subject concatenation described in "Pseudo-population design". We specified logistic regression decoding models with L2 regularization to account for potentially high dimensionality in the feature matrix. The 10-fold CV decoding analysis was run at each time point in the $nTimepoints$ dimension independently and the resulting decoding curve was submitted for statistical analyses (see "Statistical analysis"). We employed the receiver operating characteristic (ROC) area under the curve (AUC) as our classification accuracy metric for increased robustness against cutoff values of the probabilistic predictions of our logistic classifier. ROC AUC is a valid metric of effect size in multivariate logistic regression[86], including robustness to outliers, flexibility to number of classes, and invariance to order-preserving transformations[87,88], though see caveats of interpreting effect sizes in decoding analyses[89]. Decoding curves were low-pass filtered (1st-order Butterworth filter at 0.1 Hz) for visualization; all statistics are conducted on non low-pass filtered data.

**Statistical analysis.** We used a non-parametric maximum cluster statistic approach[90] to determine temporal clusters of significant decoding while keeping FWE < 0.05. Within a ROI and for a given computational variable, we permuted the target vector labels 1000 times and tested the model on the permuted labels to simulate a null distribution of mean CV accuracy, at each timepoint. Each permutation's shuffled mean CV accuracy was thresholded at the 95th percentile of this distribution and the maximum above-threshold AUC of the resulting clusters was extracted; these comprised the max. cluster-statistic null distribution. The procedure was applied to the unshuffled decoding curve and each above-threshold cluster was deemed significant if its cluster statistic ($C_{AUC}$) was above the 95th percentile of the max. cluster-statistic null distribution ($C_{thresh}$), keeping FWE < 0.05. Cluster-wide $p$ values reported in the main text are calculated as $1 - P_i^c/100$ with $P_i^c$ denoting the $i$th percentile of cluster $c$'s above-threshold statistic along the max. cluster-statistic null distribution. Effect sizes for each cluster are also reported using the ROC AUC at the peak time point within the cluster ($AUC_{time}$). Results in exploratory ROIs were FDR-corrected against the full set of 11 ROIs and corrected $p$ values are reported as $q$ values. To evaluate whether significant clusters differed in latency, we conducted non-parametric Mann–Whitney $U$-rank tests to compare the timings respective to each significant cluster, for pairs of temporally adjacent clusters as ordered by the latency of the peak decoding accuracy. The Mann–Whitney $U$-statistic and $p$ value of the test are reported alongside the [min,max] of the time span of significant clusters.

**Confidence intervals.** To provide measurements of uncertainty around the reported decoding accuracy for each ROI and variable, we employed an established bootstrapping procedure[85,91,92] adapted to a cross-validation scheme similar to[93] and summarized here. Within each iteration across our 10 folds, we evaluated the uncertainty around the decoding estimates by resampling the test dataset with replacement ($n = 500$) and computing the out-of-sample accuracy (ROC AUC) on the sampled test set, then averaging across the folds. The 95% CI around the reported out-of-sample decoding accuracy was computed

using the 2.5th and 97.5th percentiles of the distribution over bootstrapped accuracy estimates, respectively.

**Feature importance.** We used a leave-one-feature-out procedure for determining the contribution of each contact to the overall decoding accuracy for a given ROI and variable. The weight of a given feature (i.e., contact) was defined as the change in mean CV ROC AUC between a full model and one in which only that feature was dropped, and resulting features weights were of dimensionality *nContacts* × *nTimepoints* per ROI. To test for differences in the contribution of each OFC contact in decoding distinct variables, we *z*-scored the feature weights and averaged the normalized feature weights over the overlapping period of significant decoding across variables (OUT, RePE, and RiPE) per contact. We then computed Pearson's correlation (*r*) values between the average normalized feature weights respective to each variable in a tested pair (i.e., OUT and RePE; RePE and RiPE) across contacts.

### Decoding generalization

For our across-variable and temporal generalization decoding[18,94] analysis focusing on Anterior Insula and OFC, we applied the same feature preprocessing steps and leveraged the same pseudo-population design reported above. Following the procedure reported in the section "Decoding model and procedure", we ran a 10-fold CV test using L2-regularized logistic regression classifiers and report ROC AUC. Here we trained the classifier on the binarized RePE target vector at each time-point, and without further tuning model weights, tested the model on its ability to generalize on the out-of-sample binarized RiPE target vector across each time-point, with performance measured by the average across-fold ROC AUC. We constrained the time window for generalized decoding analysis to 0.200–0.500 s after the onset of card 2, to encompass the time points in which we were able to decode RePE and RiPE alone (i.e., without generalization) in both ROIs and the a priori hypothesis that RePE (and thus the extent to which it can inform RiPE) can only occur after the onset of card 2 when the outcome is known. To test for robustness, we repeated this procedure without constraining the time window, thus using the entire epoch after the onset of card 2 (see Fig. S4). For the OFC ROI, we also ran this procedure trained on Outcome and tested on RePE given our finding that OFC contacts significantly decode both these variables in overlapping periods after the onset of card 2. We applied the same max. cluster-statistic approach described in the section "Statistical analysis" but extended the cluster based inference to two dimensions, and report results exceeding the 95th and 99th percentiles of the permuted null distribution, both keeping FWE < 0.05 at the cluster level. We report cluster-wide *p* values with respect to the max. cluster-statistic null distribution.

### Reporting summary

Further information on research design is available in the Nature Portfolio Reporting Summary linked to this article.

## Data availability

Data and materials from this study have been deposited in the Open Science Framework (OSF) database. Identifier: https://doi.org/10.17605/OSF.IO/RKG4Q. The following databases were used in the study: Harvard-Oxford probabilistic atlas and MNI152 standard space template from the FSL toolbox (v6.0; https://fsl.fmrib.ox.ac.uk/fsl/fslwiki/Atlases), and the Neubert cingulate and orbitofrontal cortex atlas (http://www.rbmars.dds.nl/CBPatlases.htm). Source data are provided with this paper.

## Code availability

Custom code is available at this Github repository. Identifier: https://doi.org/10.5281/zenodo.10525236.

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

## Acknowledgements

This work was supported by a grant to J.P.O. and M.A.H. from NIMH (R01MH111425). We thank members of the Human Reward and Decision Making lab and the Human Brain Research Lab for discussions and feedback. We thank all participants and their families for participating in this research.

## Author contributions

V.M., J.C., O.F., P.G., M.S., C.K.K., H.K., and H.O. collected the data. J.C. and J.P.O. conceived of the experiment. M.A.H. and J.P.O. acquired funding and supervised research. V.M. and J.P.O. designed the data analysis. V.M. conducted the data analysis. V.M., J.C., and J.P.O. wrote the initial draft of the manuscript. All authors revised the manuscript.

## Competing interests

The authors declare no competing interests.
