## [Peer Review File · Nature Communications]

Temporally organized representations of reward and risk in the human brainREVIEWER COMMENTS

Reviewer #1 (Remarks to the Author):

This study examined the temporal sequence of reward and risk processing via intracranial EEG in 10 epilepsy patients. They present evidence for parallel and sequential processing of different pieces of information related to the experimental task across the brain. This manuscript is generally well and clearly written, the introduced literature seems appropriate, the study design is very well thought through, and the conclusions derived from the findings seem sound. Thus, I am overall fairly enthusiastic about this manuscript. Yet, there are some issues with the manuscript in its current form, mainly with the small sample size and the accompanying uncertainty of results, which I would like to point out to the authors and ask for clarifications and explanations in the manuscript.

Further, I would like to ask the authors to consider making their data and analysis code openly available and include a Data Availability Statement in the manuscript (which should be included even if there were reasons against sharing the data openly; in that case, please specify the reason for not openly sharing your data).

Major issues:

1. The sample in this study is (naturally) small. This cannot be avoided with this type of method and sample. Yet, this can have far-reaching consequences for the robustness and generalizability of the findings, above all in combination with the relatively small number of trials per participant (90 trials for four participants, 180 trials for six participants). Please specify explicitly in the manuscript, which measures you have taken to increase precision/robustness/reliability of your measures/findings and how large the uncertainty around your statistical estimates is due to the small sample size and number of trials. Also, please include a sample size justification in the manuscript. I can guess why the sample is so small, but it would be better to state that explicitly.

2. Adding to the previous comment, different amounts of data from different subjects for different brain areas have factored into the analysis. For example, you highlight the role of the insula's activation in the manuscript, yet there were only data of six and five participants for the anterior and posterior insula, respectively, available for these analyses (according to Tables S5 and S6). Also, there were only six participants with electrodes in OFC (though with more contacts per subject). Yet, these two ROIs receive the most attention in your manuscript and seemingly showed the most associations with reward and risk representations. I am wondering whether this pattern of findings might stem from those two regions being genuinely more involved in these processes or whether the relative scarcity of data in those regions might have given rise to coincidental or biased findings. Please discuss in how far the chosen

analytical approach deals with the relatively small number of individuals and trials per individual in these analyses as well as differing amounts of data across participants and ROIs.

3. Please report all inference statistical values, for example t-, z-, or r-values, wherever possible instead of only reporting p- (or q-) values for each result from line 127 onwards. Please also include estimates of the uncertainty around your statistical estimates (e.g., standard error, confidence intervals) wherever it is possible and you have not done so already.

4. Figures 3A, C, 4, 5A, and B show relatively large fluctuations in the area under the curve and mostly effects seem to be just so above the 95th percentile of the permuted null distribution. What does that mean in terms of effect size? How robust might the findings be? Would you have expected larger effects given previous studies and your own previous experience? Please discuss these matters in your manuscript.

5. I think “2.6 Reward error representation generalizes to risk error in Anterior Insula but not OFC” is a bit difficult to understand for readers lacking background knowledge with this kind of analysis and this part would benefit from a more basic, comprehensible description of the method.

Minor issues:

1. You present mean and standard deviation in line 86 and mean and standard error in line 89. Please be consistent in how to report the variance component so that it is more easily comparable.

2. In frequentist statistics, there is no such thing as a trend towards statistical significance. You interpret a p-value of 0.075 in line 89 as a sign of fatigue, but you do not interpret a p-value of 0.086 in line 92 as behavioral strategy during learning tasks. This is inconsistent. Please define and justify a threshold of statistical significance explicitly in the Methods section and stick with it for interpretation of the existence of effects. That is, do not interpret effects with a p-value above that threshold.

3. In some paragraphs, you seem to be using causal language (at least it seems so to a non-native reader). Starting from “2.3 Distinct modes ...”, you repeatedly write about one brain area “leading” another one. This wording might imply a causal relationship between activity in those areas. However, as I understand your analyses, they are purely correlational and should not be interpreted causally. Even though you show a temporal pattern that might fit causality, without a mechanistic model of that relationship and/or a perturbation of this system you should not conclude causality. Please revise language, which might be interpreted as implying causality where it should not, throughout the manuscript to make sure to avoid misinterpretation.

4. In Figure 3F, there are numerous black dots though the color scale entails only yellow to orange. Please specify, what the black dots mean in the figure caption.

Reviewer #2 (Remarks to the Author):

Key Results and Significance

This is a very interesting research article investigating the detailed neural processing of reward and risks computations at high spatiotemporal resolution. Using intracranial EEG (iEEG) recordings in regions of the brain known to be involved in reward / risk processing, including the anterior and ventromedial prefrontal cortices, amygdala, orbitofrontal cortex and anterior insula, the authors for the first time elucidated the temporal involvement of brain regions in the computation of reward and risk. Building on previous findings using non-invasive studies, they showed that expected value is processed simultaneously in vmPFC, amygdala and hippocampus, while more computational demanding calculations such as reward outcomes is represented via sequential processing that move from posterior to anterior regions of the brain. Additionally, error signals in both reward and risk calculations utilized a mixture of sequential and parallel neural processing. This adds to the current understanding of detailed temporal neural processes involved in decision making behaviors involving reward and consequently risks.

Validity

The design of the study using iEEG recordings overcome traditional limitations of non-invasive recordings. Additionally, the task was designed to be a simple gambling paradigm, yet able to decouple the variables (Expected value, EV; Outcome, OUT; expected risk, E.Risk; reward prediction error, RePE and risk prediction error, RiPE) involved in reward and risks computation using a normative model of reward and risks, and allowed independent sampling of these variables in each trial, preventing inter-trial interactions. The basis of the study relies heavily on the hypothesis that EV, OUT, E.Risk, RePE and RiPE are computed in a sequential manner as information is acquired, and that RiPE is a second-order term, derived as a function of RePE, therefore RePE signal would precede RiPE signal. This hypothesis is scientifically conceivable.

Data and methodology

The authors presented the data in a logical and systematic approach, first explaining the study design (Figure 1) and methods taken to ensure no confounding in data acquisition between trials as well as robustness of results of patient participation (Figure 2). Results of multivariate decoding approach to decipher each of the variable involved in rewards / risks computation were then presented in Figures 3-

6. Figures 3-5 displayed the main results clearly in a manner that is intuitive. They showed parallel decoding for EV in the vmPFC and amygdala, versus sequential processing from posterior to anterior brain regions for outcomes decoding. Based on a hypothesis driven approach, they also presented findings of representation of E.Risk decoding in the OFC. Further results were then presented with display of parallel and sequential computations for RePE and RiPE, as well as identification of unexpected new brain regions involved such as the cingulate gyrus and posterior insula from exploratory analyses.

Analytical approach

The data analysis approach was generally robust and comprehensive. The authors used mixed-effects regression models to differentiate within- and between-subject sources of variance, and ensured no inter-trial learning confounders. General linear model was employed to regress variables for the decoding analysis model to ensure no contamination of result. Within-subject analysis and cross-validation was performed, as well as across-subjects validation and accuracy metric with the receiver operating characteristic AUC (significance defined as above 95th percentile). Multiple layers of tests were used to ensure validity of analysis results.

Suggested improvements

In this study cohort of 10 subjects with 16 recording sessions, there were 6 implantations the right, 3 on the left and only 1 bilaterally. Although briefly mentioned in Section 4.5, it is not clear if the contact localization was performed first in patient's brain space, before registration to a common template, or localized in template space. The contact localization after warping into template space may introduce localization error. Throughout the manuscript there was also no mention of segregation of ROIs based on laterality and it was unclear if there was any significant difference between left and right ROIs in the findings. Such an analysis should be performed, and at least acknowledged in the limitations if the sample is not powered sufficiently to examine the results. In Section 4.8, line 508, there is a minor error in ... " given in 8/10 of the remaining"... this should be 8/9. There are, in addition, numerous references in the Supplementary material and figures that are mislabeled or left black with a [?] which should be corrected.

What a cortical-cortical evoked potential analysis strengthen their cross-correlation approach to connectivity? Was this performed?

Clarity, Context and References

The authors have presented several previous work and current understanding of reward / risk computation and referenced them in the introduction, and discussion and discussed comparisons of the current findings to other authors and groups.

Reviewer #3 (Remarks to the Author):

In their article titled "Temporally Organized Representations of Reward and Risk in the Human Brain," Man et al. investigate the temporal and spatial dynamics of risk and reward representation, as well as their associated prediction error, using intra-EEG. They employed a simple gambling task where participants had to predict whether the second of two sequentially-drawn playing cards would have a higher or lower numerical value than the first card. Correct guesses were rewarded with points, while incorrect guesses resulted in a loss of points. The authors defined five variables of interest: expected value (EV) (given guess and card 1), outcome (OUT) (at card 2), expected risk (E.Risk) at card 2, reward prediction error (RePE) at card 2, and risk prediction error (RiPE) for each card. The results can be summarized as follows:

1. Expected value was observed in the ventromedial prefrontal cortex (vmPFC), amygdala, and hippocampus.
2. Outcome was detected in multiple brain regions, exhibiting a postero-anterior gradient of temporal representation. The authors found directed functional connectivity from the frontal pole to the orbitofrontal cortex (OFC) and from the OFC to the vmPFC.
3. Expected risk was identified in the OFC but not significantly in the anterior insula. It was also found in the posterior insula and intraparietal lobule.
4. Reward prediction error representation was observed in the OFC and anterior insula.
5. Risk prediction error representation was found in the anterior insula and OFC. Notably, RiPE representation in the anterior insula occurred later than RePE. An additional result indicated that RePE representation is relevant for the RiPE of card 2.

Overall, these results provide valuable insights into the temporal dynamics of risk and reward-related neural signals. However, I have a series of general and specific concerns. There are several missing details, and the figures fail to effectively support the article. I will provide more detailed comments below.

1) General concerns

a) Regarding ROIs:

- It would be helpful to specify the number of contacts belonging to each ROI. Including this information will provide transparency and aid readers in understanding the sample size for each region of interest.
- The size of the Frontal pole ROI appears to be much larger than any other region, and the OFC seems relatively posterior according to Figure 1B. Authors should aim to define ROIs with a similar number of recording sites and ensure that the ROIs are comparable in terms of anatomical size or delineation. For

example, it is unlikely that dlPFC and anterior lateral OFC compute the same variables, so it is important to address this discrepancy.

- Consistency in ROI names is essential. The manuscript currently uses various terms for the same region, such as for the OFC (which is actually the posterior OFC) : lateral OFC, and posterior lateral OFC. It is crucial to maintain consistency in ROI names throughout the manuscript to avoid confusion. I recommend using a single, agreed-upon term for each ROI and using it consistently across the entire article.

b) Regarding Timings:

- The timing description for the variables depicted in Figure 1C is currently confusing. The label "before/at card" is imprecise and lacks clarity. It would greatly benefit the reader if the authors unpacked this figure and provided a clear explanation of the timing for each variable. For instance, it should be clarified that EV will be investigated at card 1 onset, E.Risk1 before onset (constant), and RiPE1 at card 1 onset. Similarly, E.Risk2 will be investigated before card 2 onset, while RePE and RiPE will be examined at card 2 onset. By providing this clarification, readers will have a better understanding of the temporal sequence of events.

- Conclusions on timings are valid within time windows of interest (at onset of card 1 or onset of card 2), but not across time windows as variables are decoded only in their specific time windows. In other words, the conclusions drawn about temporal dynamics, apart from outcome decoding and RePE representation informative for RiPE representation, may not be as convincing. To strengthen the results, it would be more persuasive to systematically apply decoding techniques to three time windows for each ROI (guess time, card 1 onset, card 2 onset). This can be visualized in the schematic representation of the results I made (to understand the results). This approach would demonstrate that decoding occurs specifically within the expected time window. Once this specificity is established, the authors can delve deeper into the temporal comparisons within each particular time window. Conducting such analyses would provide reassurance that observed differences are not merely false positives. It is worth noting that the authors have appropriately applied corrections for multiple comparisons.

c) Regarding analyses:

- Why did the authors choose to do multivariate analyses? I understand the rationale for binary variables, but less for continuous variables. What are the results for univariate analyses with continuous variables, are they similar? In any case, justification for the choice of multivariate analyses should be clearly stated somewhere.

- The directed connectivity analyses were conducted solely for outcome, but it would be more consistent to extend these analyses to EV, RePE, and RiPE2 as well. Including directed connectivity analyses for these variables would provide a comprehensive assessment of the functional connectivity across the different ROIs. Ensuring consistency in the choice of variables for directed connectivity analyses would enhance the overall validity and comprehensiveness of the study's conclusions.

d) Regarding computational variables:

- When it comes to the naming of variables, it is essential to ensure clarity and consistency. While it appears that the variables are investigated in a specific order, it can be confusing to determine whether RiPE refers to RiPE1 or RiPE2 when examining the equations. To avoid ambiguity, I would recommend to clearly state somewhere in the manuscript whether RiPE refers to RiPE1 or RiPE2 consistently throughout the text. Alternatively, authors could choose to consistently use either RiPE2 or RiPE1, as well as E.Risk1 or E.Risk2, to maintain clarity and avoid confusion for the readers.
- The inclusion of the observed risk as a computational variable in the design matrix space raises questions since it is not tested in the results. It would be helpful to provide a justification or rationale for including this variable in the design matrix space if it is not examined or reported on in the results section. Providing clarification on the purpose or role of this variable will enhance the understanding of the experimental design and the analyses rationale.

e) Regarding Figures:

- Figures are cited in the following order: 1A, 1C, 2A, 2B, 2C, 2E, 2F, 2D, 3A, 3B, S1, 3D, 3F, S5B, S5A, S1B, 4, S1B bottom, 5A, S2A, 5C, S2A, 3C, 5A, 5B, 5, 6A, S3, S4A, S4B, 6B, 1A, S8A, 1B, S8B, S6B, S7, S3. This is particularly confusing, and some are not referenced in the text, and supplementary figures appear in a random order.
- Consider enhancing the visual elements: the figures could benefit from a more aesthetically pleasing design. It would be helpful to pay attention to factors such as font size, color schemes, and overall layout (alignment) to ensure a visually coherent representation that facilitates understanding of the results.

2) Specific concerns (I numbered the results consistently with the summary provided at the beginning of the review):

a) Result 2-Connectivity: “While we found evidence of differential cross-correlation between high and low outcome trials in prefrontal cortex, we instead observed that areas found to encode outcomes later led areas encoding outcomes earlier in time.” This sentence is not clear, what does it mean? Also, why testing only PFC – OFC and OFC-vmPFC and not all possible combinations? Or why not reporting other possible combinations (even if not significant)? The direction of connectivity seems counter-intuitive. Authors interpret it as a top-down process in the discussion, but this should be more discussed, or more explored in the analyses.

b) Result 3: Regarding expected risk results, I would suggest not to interpret non-significant results (even “suggestions”, which I understand as “trends”).

c) Result 5: It is not clear what “contrasted temporal profile of RiPE against RePE” means.

3) Minor concerns:

- I don't understand the difference between figure 6A and figure S3
- The task should be briefly described in the abstract.

We thank the editor and the three reviewers for their time and effort in giving us important comments for improving the paper. We have conducted additional analyses and revised our manuscript in light of these comments. We believe our efforts to address the concerns that the reviewers raised have significantly improved the quality of our work. Below, we demonstrate the ways that we have: 1) added substantial clarifying text to our analyses and discussion in the manuscript, 2) conducted additional neural analysis to strengthen our argument about the timing of reward and risk decoding, 3) reinforced through additional analyses the robustness of our reported effects, 4) investigated the contributions of different sub-regions in our reported decoding results, and 5) added additional text in the manuscript providing a more comprehensive description of the limitations of our study thereby offering a more nuanced perspective of our contribution to the field. Please find our replies below alongside changes to the manuscript (**in blue bolded font**). To provide context to our changes we have in some cases also copied unchanged text from the manuscript around the changes (**in blue, unbolded font**). The reviewers' original comments are shown in **black bolded italicized font**.

REVIEWER COMMENTS

Reviewer #1 (Remarks to the Author):

This study examined the temporal sequence of reward and risk processing via intracranial EEG in 10 epilepsy patients. They present evidence for parallel and sequential processing of different pieces of information related to the experimental task across the brain. This manuscript is generally well and clearly written, the introduced literature seems appropriate, the study design is very well thought through, and the conclusions derived from the findings seem sound. Thus, I am overall fairly enthusiastic about this manuscript. Yet, there are some issues with the manuscript in its current form, mainly with the small sample size and the accompanying uncertainty of results, which I would like to point out to the authors and ask for clarifications and explanations in the manuscript.

We thank the reviewer for their positive outlook and enthusiasm about our manuscript. Please see below our clarifications and explanations to each specific point and corresponding changes in the manuscript.

Further, I would like to ask the authors to consider making their data and analysis code openly available and include a Data Availability Statement in the manuscript (which should be included even if there were reasons against sharing the data openly; in that case, please specify the reason for not openly sharing your data).

We have made our data and code openly available. We have added a Data Availability Statement as well as Code Availability Statement directly in the manuscript:

5. Data availability

Data and materials from this study are available in the Open Science Framework (OSF) database. Identifier: <https://doi.org/10.17605/OSF.IO/RKG4Q>. Source data are provided with this paper.

6. Code availability

Custom code is available at this Github repository {https://github.com/manvincent/temp_org_iEEG}

Major issues:

1. The sample in this study is (naturally) small. This cannot be avoided with this type of method and sample. Yet, this can have far-reaching consequences for the robustness and generalizability of the findings, above all in combination with the relatively small number of trials per participant (90 trials for four participants, 180 trials for six participants). Please specify explicitly in the manuscript, which measures you have taken to increase precision/robustness/reliability of your measures/findings and how large the uncertainty around your statistical estimates is due to the small sample size and number of trials.

We note the reviewer's concern on the reliability of the findings and the associated uncertainty around our statistical estimates. We have taken several approaches to address this concern. Specifically, we adopted several signal processing techniques to increase the signal-to-noise ratio of our neural recordings. First, to lower the measurement noise at the level of each recording contact, we performed an ICA-based denoising method previously validated for iEEG data (Michelmann et al., 2018), which removes systematic noise from distant contacts and thereby increases the spatial specificity of neural sources. We also adopted a bipolar referencing scheme, at the cost of reducing the number of neural measurements by half (i.e. pairs of adjacent contacts' signals are subtracted resulting in one source). We felt this was justified because bipolar referencing further increases SNR and critically ensures that inferences about source location are valid due to the reduction of volume conduction effects. The increased spatial precision afforded by bipolar referencing was important here given our interest in differential timings for decoding across regions. In other words, we prioritized the quality (SNR and precision of spatial interpretation) of our model features over the quantity of features.

Furthermore, in our statistical analyses we endeavored to increase the computational specificity of our findings in multiple ways. First, we ensured that computational variables were not correlated either by leveraging the sequential structure of a trial to temporally space apart variables at different respective epochs, or by statistically orthogonalizing variables that occur at the same epoch within a trial (e.g. Outcome and RePE). This ensured that findings regarding e.g. RePE were attributable only to the unique variance of that variable, allowing for precise interpretations about the neural representation for a particular variable. Further, we took steps to preprocess our decoding model's neural features again with the aim of ensuring that variance related to computational variables *other* than the one under current investigation did not

contribute to our decoding results. Finally, we used the area under the curve of the receiver operating characteristic as our decoding accuracy because this metric is robust against the choice of threshold in the probabilistic output of our logistic regression model (Ling et al., 2003; Bradley, 1997), as described in 4.10.3 “Decoding model and procedure”. We have previously detailed these approaches in the Methods section; now we further expand on the justification and advantages of these approaches to decrease measurement noise, and increase precision and robustness in the main text.

In Section 2.2 Decoding computational variables with multivariate neural signals

To ensure unique decoding of each variable of interest, we regressed out all other variables and covariates (e.g. per-trial report accuracy) from each neural feature. We then extended our decoding approach across ROIs spanning cortical and subcortical regions, focusing primarily in our key regions of interest but also conducting an exploratory investigation into a larger set of potentially relevant ROIs. **Because our analytic approach compares across ROIs in terms of respective timings, we took care to ensure our neural signals were spatially precise and reflected unique sources of information. To do so, we used a bipolar referencing scheme in which pairs of adjacent contacts were subtracted to reflect a single source (see Figure~\ref{suppl:contacts}) alongside an ICA-based denoising method which removed distal noise. These processing approaches also increased the signal to noise ratio of our neural features, and together facilitated the comparison of decoding results across regions in order to identify temporal patterns of representation at a wide scale across the brain.**

In addition to the clarification in the text on measures we took to increase the precision and robustness of our measurements, to further address this concern we now report confidence intervals (95 CI) around the decoding accuracy for all reported effects. We employed a bootstrapping procedure to compute confidence intervals based on resampling the test set, which has been shown through simulation work (<https://github.com/rasbt/machine-learning-notes/tree/main/evaluation/ci-for-ml>) to better recover ground truth accuracy metrics and produce 95 CI containing the ground truth parameter 95% of the time, compared to other methods (e.g. resampling the training set via the .632 method [Efron, 1983; Efron & Tibshirani, 1997]). We have now added a new section to the Methods documenting precisely our bootstrapping approach to compute confidence intervals.

4.10.5 Confidence intervals

To provide measurements of uncertainty around the reported decoding accuracy for each ROI and variable, we employed an established bootstrapping procedure \cite{glaser2020machine, sanchez2019machine, raschka2018model} adapted to a cross validation scheme similar to \cite{fu2005estimating} and summarized here. Within each iteration across our 10 folds, we evaluated the uncertainty around the decoding estimates by resampling the test dataset with replacement (n=500) and computing the out-of-sample accuracy (ROC AUC) on the sampled test set, then averaging across the folds. The 95 percent CI around the reported out-of-sample decoding accuracy was

computed using the 2.5th and 97.5th percentiles of the distribution over bootstrapped accuracy estimates, respectively. The approach of bootstrapping the test set has been shown through simulation studies to more accurately.

We also now report 95 CI directly in-text throughout our Results section, and added figures panels for each respective variable denoting the confidence interval around our decoding accuracy estimate (e.g. Figure 3A,E):

Figure captions have been updated accordingly. For example starting in Figure 3: (A) **Uncertainty estimates around peak decoding accuracies. To compare across ROIs, differences between decoding accuracy and the median (50th percentile) of the permutation-based null distribution, respective to the decoding analysis of each ROI, is depicted. Points and the violin plots show the distribution of differential bootstrapped test accuracies against chance. Box plots depict the median (black point), interquartile range (thick bar), and 95 CI (thin bar) of the bootstrapped distribution.**

Also, please include a sample size justification in the manuscript. I can guess why the sample is so small, but it would be better to state that explicitly.

Below we respond to both this concern of a small sample size as well as point 2 regarding regions containing different numbers of patients and contacts. As the reviewer points out, they are related issues:

2. Adding to the previous comment, different amounts of data from different subjects for different brain areas have factored into the analysis. For example, you highlight the role of the insula's activation in the manuscript, yet there were only data of six and five participants for the anterior and posterior insula, respectively, available for these analyses (according to Tables S5 and S6). Also, there were only six participants with electrodes in OFC (though with more contacts per subject). Yet, these two ROIs receive the most attention in your manuscript and seemingly showed the most associations with reward and risk representations. I am wondering whether this pattern of findings might stem from those two regions being genuinely more involved in these processes or whether the relative scarcity of data in those regions might have given rise to coincidental or biased findings. Please discuss how the chosen analytical approach deals with the relatively small number of individuals and trials per individual in these analyses as well as differing amounts of data across participants and ROIs.

We appreciate the reviewer's concerns regarding the small sample size. As suggested by the reviewer, we have now added an explicit sample size justification in the main text of the manuscript:

2. Results

To understand how the brain temporally organizes reward and risk representations, we combined iEEG recordings in 10 patients across 16 recording sessions, all of whom completed a task designed to elicit an array of reward and risk computations \cite{preuschoff2006neural}. **All patients were recruited on the basis of clinical considerations only and were implanted with multi-contact subdural grid and/or depth electrodes to enable chronic evaluation for the treatment of refractory epilepsy. Electrode placement was determined by the clinical requirements specific to each patient; as such the number of contacts varied across individuals. We thus pooled all recording contacts across patients using a pseudo-population approach \cite{rutishauser2015representation} to sample from our regions of interest (see individual subject coverage in Figure~\ref{suppl:contacts}). In total 428 contacts were included in the analysis across all ROIs. The number of contacts within each ROI is shown in Supplementary table \ref{tabS1}).**

We have added a paragraph in the Discussion regarding the small sample size of this study. We discuss the rare patient population from which we could record invasive electrophysiology (data collection occurred over a 5 year period averaging 2 patients per year, and all patients who consented to participating in this research project were included). As discussed in the text, our sample size ($n = 10$ across 16 recording sessions), as well as the number of contacts within a ROI (now presented in supplementary table 1 - see response to point 2 below), is consistent with related recent work on value and uncertainty conducted by independent research groups. For example, a recent study on the exploration under uncertainty recorded from 6 patients with 13 and 12 contacts in their vmPFC and dmPFC ROIs, respectively (Domenech et al., 2020, Science). Two recent papers investigating reward-related computations in OFC (Saez et al., 2018, Current Biology; Marciano et al., 2023, Cell Reports) also included 10 patients in their study, with 134 contacts examined in their OFC ROI.

As described in the text, we feel that our findings contribute to the theoretical understanding of how the brain computes a set of basic variables important for diverse cognitive phenomena, such as reinforcement learning and decisions under uncertainty. We believe that a low sample size as a result of the rarity of intracranial recordings (especially to record while patients perform specifically designed tasks that elicit these computational variables of interest) is balanced by the insights afforded by neural data of high temporal and spatial precision (and as discussed in the response to point 1 above, we took measures to ensure the spatial precision of our recordings). Nonetheless, we now more explicitly discuss the limitations of our small sample size. Specifically, we point out critical methodological limitations (e.g., unequal number of patients and contacts between ROIs, see our response continued further below) as well as limitations in generalization.

In 3. Discussion:

While invasive electrophysiology confers the advantage of precise timing and high spatial specificity, critical here for isolating temporal effects across wide-spread regions of interest, the limitations of this neural modality needs to be carefully considered. Like other work involving populations from which data collection is challenging, our current study is limited by a relatively small sample size and accompanying concerns about generalization. We carefully considered the design of the study to ensure that the task was simple and straightforward enough for patients to understand and complete. For example, our task was paced so that each trial event could be read and understood by the patients, and we collected and controlled statistically behavioural measures of attention on each trial. We also took efforts during data collection to ensure patients were fully engaged with the task and only administered the task in periods distant from seizures. Finally, our behavioural analyses validated that patients understood the structure of the task (e.g. that trials were independent which was important for our decoding scheme), and behavioural metrics of task performance (e.g. report accuracy) were similar to that reported in previous literature on healthy controls \cite{preuschhoff2006neural}. Further supporting the inference that the neural circuits implicated in our results generalize to the broader population, it is also worth noting that our findings corroborate reported findings in the literature as discussed above.

Regarding different numbers of patients contributing to the decoding analysis across different ROIs:

Our approach of pooling data from different participants to form a pseudo-population to neural activity across regions follows a technique common in invasive electrophysiology studies (Rutishauser et al., 2015, Nat Neuro; Meyers, 2013, Front Neuroinform). This is necessitated by both the small sample size typical in these studies, as well as the unequal amount of contacts between patients for a given region (due to clinically-based implantation decisions). We took several measures in the creation of our pseudo-population design for decoding analysis to mitigate between-subject variance. Specifically, for a given ROI, our feature design matrix was constructed by taking subject-specific feature matrices of $n_{\text{Trials}} \times n_{\text{Contacts}}$ dimension, and concatenating along both dimensions (see Supplementary Figure 7C) for each patient that had recordings in that ROI. Critically, *prior* to concatenating each subject's feature matrices, we regress out of each contact a confound matrix modeling the average contact activity across trials for that subject (analogous to modeling the effect of subject in a fixed-effects approach). Only once this subject-specific variance is regressed out of the neural signal are subjects compiled into a group-level pseudo-population design. We have expanded on this step and its importance in the manuscript:

4.10.1 Feature preprocessing

We set up an encoding model to preprocess each of our features (contacts) for the decoding model. At the within-subject level before concatenation across subjects (see

\ref{decode_pseudopop}), for each feature we defined a general linear model (GLM) to regress out covariates of no interest, as well as computational variables aside from the current variable of interest for decoding. The confound design matrix of the GLM **importantly included an intercept modeling the average contact activity across trials for that subject; regressing this design out of each neural feature therefore removes subject-specific variance in the neural data in preparation for our pseudo-population decoding analysis.**

As with other studies employing this pseudo-population approach, we therefore make the assumption that neural activity actually recorded separately from different patients can be treated as if they were recorded simultaneously from a 'representative' brain (i.e. fixed-effects). Our implementation of the pseudo-population approach, by carefully regressing out confounding variance from the neural data, reduces the impact of between-subject variance on our decoding results. However, we acknowledge this is not equivalent to having the same amount of data across ROIs. As the reviewer points out, we also have different numbers of contacts and trials between ROIs.

On this front we made analytic decisions to protect against potential false positives from ROIs with a greater number of contacts. Specifically, our results are based on a cross-validation scheme in which the decoding score (ROC AUC) is based on the out-of-sample test set. This protects against overfitting as a result of an increased number of features (i.e., explanatory variables in the logistic regression classification model) by virtue of balancing the bias-variance trade-off (Geman et al. 1992). Further, we apply regularization to our logistic regression model, another approach widely adopted to prevent overfitting by shrinking features with low explanatory power and thereby reducing generalization error (Goodfellow et al., 2017). We adopted stringent statistical techniques to control the family-wise error that can potentially arise from conducting multiple tests along time-points within a time window of interest, for a given variable (e.g. in the epoch after the card 2 onset for outcome), by using a non-parametric maximum cluster statistic approach (Maris and Oostenveld, 2007) common in electrophysiological work (e.g. Lopez-Persem et al., 2020 used a similar approach to correct decoding across multiple time points).

Rather the concern regarding ROIs with different numbers of contacts (features) is in the possibility of false negatives when investigating ROIs with relatively fewer measurements. For example, work on the impact of searchlight size in fMRI multivariate analyses has demonstrated that a smaller searchlight radius (analogous to having fewer contacts within a ROI here in that both result in fewer features for the decoding model) can result in lower out-of-sample decoding accuracies, as can searchlight shapes not corresponding to the true underlying spatial distribution of a representation (Etzel et al, 2013). To avoid potential biases we thus used anatomical defined ROI masks from an established atlas (Harvard-Oxford, Desikan et al., 2006) in conjunction with data-driven model regularization as discussed above. Notably, our approach of aggregating contacts across multiple subjects to sample from brain regions with different numbers of recording sites has been similarly adopted by recent iEEG work on value representations (Lopez-Persem et al., 2020, Nat Neuro). In their study, ROIs defined by the AAL atlas ranged from 9 [e.g. Frontal Mid post/sup] to 148 [Temporal Mid ant] recording sites, and

the authors similarly refer to their approach as 'pseudo whole-brain' in recognition of the unequal sampling of measurements across ROIs. Following their lead, we now also explicitly discuss the methodological limitations of this approach and provide greater nuance to our interpretations in light of this concern. It is worth noting that issue of potential false negatives arising from under-sampling a ROI is an inherent limitation of intracranial electrophysiology; regions not sampled at all (e.g. due to clinical irrelevance) may hold reward and risk representations that are not possible to investigate and report. We also raise this point in the addition to the discussion section:

In 3. Discussion:

The approach of pooling across our patient sample to create a pseudo-population of recording sites follows from other works in invasive electrophysiology (Rutishauser 2015, Lopez 2020) and allowed us to sample from diverse regions across the brain. As with other studies employing this pseudo-population approach, an assumption underlying the interpretation of our findings is that neural signals actually recorded from separate patients can be treated as if they were recorded from a representative 'pseudo-population' brain. Further, it results in different ROIs containing different numbers of measurements and thus differing amounts of explanatory power between decoding models across regions (see also Lopez 2020). Specifically, regions that are under-sampled given a fewer number of recording sites may yield non-significant decoding accuracy due to decreased statistical power, especially if the true underlying neural representation is spatially widespread but weakly informative when only surveying each recording site individually (Eitzel 2013). Indeed we chose to aggregate bilateral ROIs with this consideration of increasing our sampling of each ROI, though we conducted additional analyses probing laterality differences in our reported decoding results across ROIs (see Supplementary section \ref{laterality}). Of course, since electrode placement is guided only by clinical considerations, failure to observe an effect can potentially arise in our study simply due to not having electrodes in a region relevant for a particular computation, an inherent limitation to invasive electrophysiology. Therefore, non-invasive imaging techniques that provide greater spatial coverage such as fMRI is an important complement to intracranial electrophysiology. Our findings should be contextualized in light of these considerations, and future work endeavoring to pool across larger samples, for example by acquiring data on the same task across multiple sites, will be important to mitigate these limitations. Larger samples will also be conducive to expanding our findings on temporal organization, for example by validating that a particular representation is specific to the time window we specified for that variable in our analyses, revealing whether the encoding of a variable can extend over the course of multiple periods within a trial, and elucidating phasic versus tonic properties of distinct representations.

Furthermore, for greater transparency and to assist the reader in understanding the context of our sample, we now include a supplementary table documenting, for each ROI, the number of

patients (out of 10) from whom we have recordings in that ROI and the number of contacts within the ROI. (Note that for the OFC we have recordings from all 10 patients).

ROI	# Participants	# Contacts	# L	# R
Ant. Insula	6	14	4	10
Post. Insula	5	10	6	4
OFC	10	72	30	42
vmPFC	5	12	6	6
Frontal Pole	8	158	58	100
Putamen	4	8	0	8
Hippocampus	10	62	20	42
Amygdala	8	32	12	20
Cingulate Gyr	6	16	2	14
Supramarg. Gyr	6	30	2	28
Angular Gyr	5	14	6	8

Supplementary Table 1: Sample of patients and contacts per ROI

For full transparency we now include contact coordinate tables for all ROIs analyzed and thus all contacts included in this paper - previously we only included these coordinates for Frontal Pole, OFC, vmPFC, and anterior and posterior insula. To reduce clutter we include all tables in the Source Data provided with the paper rather than in the Supplementary Materials.

3. Please report all inference statistical values, for example t -, z -, or r -values, wherever possible instead of only reporting p - (or q -) values for each result from line 127 onwards. Please also include estimates of the uncertainty around your statistical estimates (e.g., standard error, confidence intervals) wherever it is possible and you have not done so already.

Our reporting of statistical values has been amended to be consistent throughout the text and comprehensive. For descriptive statistics we report mean and standard error of the mean, correcting the typo (see our response to minor issue #1 below). For inferential statistics we now report 95 CI directly in-text throughout our Results section. Specifically, for our behavioral results we report statistics in the following form: (β coefficient, **95 CI** [lower upper] around coefficient, z - or $t(df)$ -statistic depending on a logistic vs linear model, **p -value**). For example in section 2.1:

We found evidence that participants indeed attended to the information provided by the presented cards as demonstrated by strong reporting accuracy significantly above chance in our sample (**mean = 91.50 %**, **s.e.m. = 2.84 %**, **$\beta = 0.415$** , **95 CI [0.358, 0.471]**, **$t(9) = 14.398$** , **$p = 1.184e-7$** ; **Figure \ref{fig2}A**). There was no effect of trial count on reporting accuracy (**$\beta_{t} = -1.272$** , **95 CI [-2.672 0.127]**, **$z = -1.782$** , **$p = 0.075$** ; **Figure \ref{fig2}B**) and the accuracy remained high even in the late trials (mean = 88.4 %, s.e.m. = 3.96 %). In all subsequent neural analyses we controlled for trials with inaccurate reports across the entire span of the task. Further, participants did not show any evidence that they adopted behavioral

strategies commonly seen in learning tasks such as win-stay/lose-shift ($\beta_{\text{OUT}} = -0.103$, 95 CI [-0.220 0.014], $z = -1.719$, $p = 0.086$; Figure \ref{fig2}C), nor did we find evidence that participants adopted other behavioral heuristics such as sticking to the same guess ($\beta_{\text{t-1}} = -0.092$, 95 CI [-0.346 0.162], $z = 0.709$, $p = 0.478$; $\beta_{\text{t-2}} = 0.005$, 95 CI [-0.235 0.245], $z = 0.039$, $p = 0.969$; Figure \ref{fig2}E) or left/right response key ($\beta_{\text{t-1}} = 0.181$, 95 CI [-0.488 0.851], $z = 0.531$, $p = 0.595$; $\beta_{\text{t-2}} = 0.166$, 95 CI [-0.270 0.602], $z = 0.746$, $p = 0.456$; Figure \ref{fig2}F).

For our neural results we now report statistics in the following form: (C_{AUC} cluster metric [above-threshold AUC integrated across time points within cluster], C_{thresh} the cluster metric of the respective max-cluster-statistic null distribution at the 95th percentile critical threshold, **p-value** of the cluster, AUC_{time} the decoding metric at the peak time point (sec) within the cluster, **95 CI [lower upper]** around AUC_{time}). This notation is described in the methods:

In 4.10.4 Statistical analysis:

Within a ROI and for a given computational model, we permuted the target vector labels 1000 times and tested the model on the permuted labels to simulate a null distribution of mean CV accuracy, at each timepoint. Each permutation's shuffled mean CV accuracy was thresholded at the 95th percentile of this distribution and the maximum above-threshold AUC of the resulting clusters was extracted; these comprised the max. cluster-statistic null distribution. **The procedure was applied to the un-shuffled decoding curve and each above-threshold cluster was deemed significant if its cluster statistic (C_{AUC}) was above the 95th percentile of the max. cluster-statistic null distribution (C_{thresh}), keeping FWE < 0.05.** Cluster-wide p-values reported in the main text are calculated as $(1 - P_{\text{c}}^i / 100)$ with P_{c}^i denoting the i^{th} percentile of cluster c 's above-threshold statistic along the max. cluster-statistic null distribution. **Effect sizes for each cluster are also reported using the ROC AUC at the peak time point (sec) within the cluster (AUC_{time}).**

All effects reported throughout the results have been amended to include this additional information. For example in section 2.3:

Consistent with a perspective highlighting differences in what the brain needs to do to compute each respective variable, we found different temporal configurations in the decoding patterns for EV and outcome across regions. We were able to decode EV in our hypothesized regions of vmPFC ($C_{\text{AUC}} = 0.725$, $C_{\text{thresh}} = 0.543$, $p = 0.032$, $\text{AUC}_{\text{0.096}} = 0.576$, 95 CI [0.517 0.605]) and amygdala ($C_{\text{AUC}} = 0.471$, $C_{\text{thresh}} = 0.234$, $p = 0.022$, $\text{AUC}_{\text{0.070}} = 0.564$, 95 CI [0.525 0.592]) (Figure \ref{fig3}A) in temporally overlapping windows (vmPFC [0.040 - 0.142 sec]; amygdala [0.044 - 0.102 sec]) after the onset of card 1 (Figure \ref{fig3}B). In an exploratory analysis, we were also able to decode EV in the hippocampus with two temporal clusters (**early**: $C_{\text{AUC}} = 0.382$, $C_{\text{thresh}} = 0.239$, $p = 0.025$; $q = 0.117$, $\text{AUC}_{\text{0.100}} = 0.559$, 95 CI [0.528 0.592]; **late**: $C_{\text{AUC}} = 0.428$, $C_{\text{thresh}} = 0.239$, $p = 0.023$; $q = 0.117$; Figure~S\ref{suppl:expectations}), with an earlier

cluster overlapping in decoding timing [0.082 - 0.138 sec] when compared to vmPFC and amygdala.

4. Figures 3A, C, 4, 5A, and B show relatively large fluctuations in the area under the curve and mostly effects seem to be just so above the 95th percentile of the permuted null distribution. What does that mean in terms of effect size? How robust might the findings be?

We appreciate the reviewer's point about the robustness and size of our reported effects. First, we emphasize again here that we took a stringent statistical approach for determining significance, in which temporal clusters were evaluated against the maximum cluster statistic from a permuted null distribution and thus accounted for family-wise error (Maris and Oostenveld, 2007). Additionally, because our chance level decoding accuracies were determined by a non-parametric permutation-based approach, our statistical analyses make no assumptions about the properties (e.g. normality) of the null distribution nor the chance decoding level being 0.5 for a binary classification problem; that the median of our permuted null distribution is close to 0.5 for all ROIs and variables validates that our permutation shuffle was effective and that there was no spurious contamination between train and test splits of the data.

Regarding the reviewer's concern about the robustness of our effects, we have now included (in response to point 1 by the reviewer) additional bootstrap-based analyses to measure the confidence intervals round the decoding accuracies, at the relevant time point within statistically significant periods that characterizes the temporal profile for a given variable across relevant ROIs. The lower bound of the 95% CI for all our reported effects lies outside of the chance decoding level. We have now included estimates of uncertainty in-text for all results (see response to point 3 above).

Regarding the effect size of our findings, while decoding accuracy scores are often used to reflect effect size, it is important to note that these metrics provide relative measures of effect size rather than standardized metrics, such as Cohen's d (Hebart and Baker 2018). Indeed, reported decoding accuracy can be affected by treatments to data prior to classification analysis, such as averaging data by experimental conditions (Mumford et al., 2012), with greater apparent classification accuracy when data are averaged (Hebart and Baker, 2018) compared to decoding at the level of single trials as was done in our study. Despite this caveat, we chose to employ the ROC AUC as the scoring metric for our logistic regression decoding model given its advantages as a metric of effect size in multivariate logistic regression (Hosmer et al., 2013), including robustness to outliers, flexibility to number of classes, and invariance to order-preserving transformations (Smithson, 2023). We now elaborate on the utility of the ROC AUC metric as a metric of effect size in the Methods:

In 4.10.3. Decoding model and procedure:

...We specified logistic regression decoding models with L2 regularization to account for potentially high dimensionality in the feature matrix. The 10-fold CV decoding analysis was run

at each time point in the n Timepoints dimension independently and the resulting decoding curve was submitted for statistical analyses (see \ref{decode_stats}). **We employed the receiver operating characteristic (ROC) area under the curve (AUC) as our classification accuracy metric for increased robustness against cutoff values of the probabilistic predictions of our logistic classifier. ROC AUC is a valid metric of effect size in multivariate logistic regression \cite{hosmer2013applied}, including robustness to outliers, flexibility to number of classes, and invariance to order-preserving transformations \cite{smithson2023receiver, kraemer2014effect}, though see caveats of interpreting effect sizes in decoding analyses \cite{hebart2018deconstructing}.**

Would you have expected larger effects given previous studies and your own previous experience? Please discuss these matters in your manuscript.

Nevertheless as the reviewer points out it can be useful context to consider how our decoding scores compare to those in the existing literature. Given the aforementioned caveats of comparing across (unstandardized) decoding accuracies, we constrained our survey to work from our lab and independent research groups who conducted decoding analysis on similar value-based variables with intracranial electrophysiological data. We found similar levels of out-of-sample decoding accuracy scores. For example, recent work from our lab was able to differentially decode expected value from the single-unit firing activity of vmPFC and amygdala neurons in a Pavlovian conditioning context (Aquino et al., 2023, biorxiv) with average decoding scores below 0.43 (in their case chance was 0.33; Figure 4). Recent work from an independent group (Lopez-Persem et al., 2020, Nat Neuro) examined value representations (liking ratings) in intracranial EEG signal across ROIs and applied temporally resolved decoding analysis similar to us (i.e., using a logistic regression model for binary classification). They found decoding accuracy scores below 0.54 in vmPFC and lateral OFC, and below 0.52 in hippocampus (extended Fig 6). We have now integrated discussion of these comparisons in the manuscript.

In 3. Discussion:

... Spatially, our findings of EV decoding were consistent with previous work that reported representations of EV in vmPFC \cite{domenech2020neural}, amygdala \cite{holland2004amygdala, baxter2002amygdala, aquino2020value}, and hippocampus \cite{rigoli2019role, johnson2007integrating}. Further, our findings are convergent with recent work that similarly decoded value-based variables from intracranial electrophysiology data despite different experimental contexts. **For example, other work from our group has shown that EV can be significantly decoded in the vmPFC and amygdala in a Pavlovian conditioning task, with similar effect sizes (as determined by relative decoding accuracy, \cite{aquino2023encoding}). Work from independent groups have also reported similar significant decoding accuracy levels (~0.54 out-of-sample accuracy for binary classification) when decoding value (i.e. likeability) ratings from vmPFC and OFC \cite{lopez2020four}**. Here we extend this previous work further not only by showing that EV is coded in the distributed, multivariate activity within each relevant region, but also showing that the code is held in a parallel temporal configuration.

5. I think “2.6 Reward error representation generalizes to risk error in Anterior Insula but not OFC” is a bit difficult to understand for readers lacking background knowledge with this kind of analysis and this part would benefit from a more basic, comprehensible description of the method.

We agree with the reviewer that the original subheading for 2.6 can be convoluted. We have replaced it with **“Anterior insula representations of reward error can predict risk error”** which we feel more directly and simply depicts the finding from our decoding generalization analysis.

Minor issues:

1. You present mean and standard deviation in line 86 and mean and standard error in line 89. Please be consistent in how to report the variance component so that it is more easily comparable.

We apologize for the typo – the variance component depicted in line 86 (now line 123) reflects the standard error of the mean, consistent with that reported in line 89 (now line 126). We have now amended the section:

We found evidence that participants indeed attended to the information provided by the presented cards as demonstrated by strong reporting accuracy significantly above chance in our sample (mean = 91.50 %, **s.e.m. = 2.84%**, $\beta = 0.415$, 95 CI [0.358, 0.471], $t(9) = 14.398$, $p = 1.184e-7$; Figure 2A).

2. In frequentist statistics, there is no such thing as a trend towards statistical significance. You interpret a p-value of 0.075 in line 89 as a sign of fatigue, but you do not interpret a p-value of 0.086 in line 92 as behavioral strategy during learning tasks. This is inconsistent. Please define and justify a threshold of statistical significance explicitly in the Methods section and stick with it for interpretation of the existence of effects. That is, do not interpret effects with a p-value above that threshold.

We thank the reviewer for pointing out this inconsistency in inference. We agree that in frequentist statistics, p-values greater than the critical value (here $\alpha = 0.05$) should not be interpreted as ‘trending’. We have clarified our definition of a threshold for statistical significance in the Methods:

To survey the degree to which participants paid attention and completed the task, we analyzed behavior using mixed-effects regression models to differentiate between within- and between-subject sources of variance, specifying random intercepts and slopes for all terms, for each participant (shown in Wilkinson-Rogers notation). **Here we used a threshold for statistical significance as $\alpha = 0.05$ and interpret effects in which the associated p-value is below 0.05 as statistically significant. We justify this threshold to remain congruent with the non-parametric statistical analyses reported below for the neural**

analyses, in which we define the significance cut-off at the 95th percentile of the permuted null distribution.

And we have amended the interpretation of the non-significant result in line 89 (now line 125), and removed the figure panel showing this non-significant trend from Figure 2. It now reads:

There was no effect of trial count on reporting accuracy ($\beta_{t} = -1.272$, 95 CI [-2.672 0.127], $z = -1.782$, $p = 0.075$) and the accuracy remained high even in the late trials (mean = 88.4 %, s.e.m. = 3.96 %).

3. In some paragraphs, you seem to be using causal language (at least it seems so to a non-native reader). Starting from “2.3 Distinct modes ...”, you repeatedly write about one brain area “leading” another one. This wording might imply a causal relationship between activity in those areas. However, as I understand your analyses, they are purely correlational and should not be interpreted causally. Even though you show a temporal pattern that might fit causality, without a mechanistic model of that relationship and/or a perturbation of this system you should not conclude causality. Please revise language, which might be interpreted as implying causality where it should not, throughout the manuscript to make sure to avoid misinterpretation.

We thank the reviewer for their attention to detail regarding the nuance of causal language. We agree that the original version suggested causal relationships between regions despite the decoding analysis not directly testing for causal relations. As such, we have amended the language to more accurately reflect that our results speak to temporal ordering across regions in their timing for outcome decoding, but not to one region ‘leading’ another. For example in section 2.3:

We further tested whether there was a significant difference in the timing of decoding between pairs of ROIs, ordered by the relative latency of the peak of their respective decoding accuracy curves. For regions that exhibited two significant temporal clusters, we tested for differences in decoding latencies by using the most temporally proximal cluster with respect to pairwise ROIs. **We found significant differences in timing between all pairs of temporally (and thus spatially) proximal ROIs, with outcome decoding at staggered latencies across regions: angular gyrus [0.008 - 0.064 sec; $y = -56$], hippocampus [0.086 - 0.150 sec; $y = -17$] ($U = 957$, $p = 2.44e-18$), amygdala [0.188 - 0.236 sec; $y = -3$] ($U = 825$, $p = 5.73e-17$), and vmPFC [0.194 - 0.299 sec; $y = 15$] ($U = 1108$, $p = 8.33e-7$).**

We have also adjusted the tone of our results on the cross-correlation analyses. We present all cross-correlation analyses and results in the supplementary section, and discuss the cross-correlation effects in direct statistical terms rather than imposing a causal lens on the findings:

Across all computational variables and their respective pairs of ROIs, we did not find significant interactions between lag and variable level in the cross-correlation measure

(see Supplementary tables \ref{tabS3} to \ref{tabS7}). When we examined differences in cross-correlation between high and low levels of the computational variable separately at positive and negative lags, we found that there were different correlations between early signal in the frontal pole and later signal in OFC across observed outcomes ($t(334) = -2.226, p = 0.027$). Similarly, early activity in OFC contacts predicted later vmPFC activity in an outcome-dependent manner ($t(24) = 2.165, p = 0.041$; Figure~S\ref{suppl:xcorr}C). However as the interaction tests respective to each of these simple effects were not significant, we refrain from making more interpretations about directed connectivity or causal relationships between regions.

We now include in the Discussion section a paragraph on limitations to inferring causality given the nature of our observations and correlational analyses. Indeed, as suggested by reviewer 2, we propose that an important future line of work is to perturb the signal in one region and measure downstream effects in another. Within the domain of human invasive electrophysiology, we discuss how cortico-cortical evoked potential manipulations within the experimental design can strengthen inference about causality.

In 3. Discussion:

Indeed our finding that computations are structured in time across areas raised the question of whether the functional connectivity between different regions encoded computational information. Across all variables, we did not find significant differences in the cross-correlation between regions between high and low levels of that variable (see Supplementary section \ref{xcorr}). However, our approach is only one of several ways to address the question of directed functional connectivity relevant to processing these computations. Future work in human electrophysiology is well-positioned to elucidate the mechanisms driving directed connectivity. For example, cortico-cortical evoked potential manipulations \cite{matsumoto2004functional, keller2014mapping} directly perturb neural activity and therefore might provide a casual picture of how computational information propagates across the brain.

4. In Figure 3F, there are numerous black dots though the color scale entails only yellow to orange. Please specify what the black dots mean in the figure caption.

The black dots originally depicted in Figure 3F depict contacts for which the feature importance weight was below zero. In other words, these contacts did not contribute to the decoding accuracy reported for a given ROI; dropping the features did not result in a hit in decoding accuracy. Feature importance weights can be negative in our case since the feature importance, like the reported decoding accuracy, is computed based on the out-of-sample decoding accuracy which incorporates in the accuracy metric the balance between bias and variance and protects against overfitting. Importantly, in our overall decoding analysis we apply regularization to the logistic regression model, which shrinks the influence of non-contributing features (i.e. neural contacts) in a data-driven way and allows us to down-weight features with low explanatory power in an unbiased manner. In the revise manuscript this panel is no longer

included in figure 3 given the additional supplementary section on the contribution of sub-regions in our large ROIs, including OFC and Frontal Pole, to their respective decoding effects (see our response to reviewer #3 below). Instead, we show a feature importance map for outcome decoding in Frontal Pole (the largest ROI) where we clarify the colour scheme:

In Supplementary Figure S9: Sub-regional contributions to originally reported decoding results in OFC and Frontal Pole:

...(E) Feature importance map of Frontal Pole shows a spatially distributed outcome representation across sub-regions. Each coloured dot represents a bipolar source which contributed to outcome decoding in Frontal Pole, with colour depicting normalized feature weight. Black dots depict contacts with negative feature weights prior to normalization.

Reviewer #2 (Remarks to the Author):

Key Results and Significance

This is a very interesting research article investigating the detailed neural processing of reward and risks computations at high spatiotemporal resolution. Using intracranial EEG (iEEG) recordings in regions of the brain known to be involved in reward / risk processing, including the anterior and ventromedial prefrontal cortices, amygdala, orbitofrontal cortex and anterior insula, the authors for the first time elucidated the temporal involvement of brain regions in the computation of reward and risk. Building on previous findings using non-invasive studies, they showed that expected value is processed simultaneously in vmPFC, amygdala and hippocampus, while more computational demanding calculations such as reward outcomes is represented via sequential processing that move from posterior to anterior regions of the brain. Additionally, error signals in both reward and risk calculations utilized a mixture of sequential and parallel neural processing. This adds to the current understanding of detailed temporal neural processes involved in decision making behaviors involving reward and consequently risks.

We thank the reviewer for their interest in our work, and for their time in carefully considering the details of the research approach and contribution to the current understanding of decisions involving reward and risk.

Validity

The design of the study using iEEG recordings overcome traditional limitations of non-invasive recordings. Additionally, the task was designed to be a simple gambling paradigm, yet able to decouple the variables (Expected value, EV; Outcome, OUT; expected risk, E.Risk; reward prediction error, RePE and risk prediction error, RiPE) involved in reward and risks computation using a normative model of reward and risks, and allowed independent sampling of these variables in each trial, preventing inter-trial interactions. The basis of the study relies heavily on the hypothesis that EV, OUT, E.Risk,

RePE and RiPE are computed in a sequential manner as information is acquired, and that RiPE is a second-order term, derived as a function of RePE, therefore RePE signal would precede RiPE signal. This hypothesis is scientifically conceivable.

Data and methodology

The authors presented the data in a logical and systematic approach, first explaining the study design (Figure 1) and methods taken to ensure no confounding in data acquisition between trials as well as robustness of results of patient participation (Figure 2). Results of multivariate decoding approach to decipher each of the variable involved in rewards / risks computation were then presented in Figures 3-6. Figures 3-5 displayed the main results clearly in a manner that is intuitive. They showed parallel decoding for EV in the vmPFC and amygdala, versus sequential processing from posterior to anterior brain regions for outcomes decoding. Based on a hypothesis driven approach, they also presented findings of representation of E.Risk decoding in the OFC. Further results were then presented with display of parallel and sequential computations for RePE and RiPE, as well as identification of unexpected new brain regions involved such as the cingulate gyrus and posterior insula from exploratory analyses.

Analytical approach

The data analysis approach was generally robust and comprehensive. The authors used mixed-effects regression models to differentiate within- and between-subject sources of variance, and ensured no inter-trial learning confounders. General linear model was employed to regress variables for the decoding analysis model to ensure no contamination of results. Within-subject analysis and cross-validation was performed, as well as across-subjects validation and accuracy metric with the receiver operating characteristic AUC (significance defined as above 95th percentile). Multiple layers of tests were used to ensure validity of analysis results.

We are glad that the reviewer found our efforts to communicate our analytic approach clear and our presentation of our findings intuitive. To further improve transparency to the reader and demonstrate the robustness of our results, we have incorporated additional analyses and now report estimates of uncertainty around our statistical measures throughout the text in response to comments by reviewer 1. We have also taken efforts to further improve the communication of our findings in our figures, such as re-organizing the order of figure panels to flow more smoothly with references from the main text, in response to comments by reviewer 3. We believe these changes facilitate understanding of our findings even more to readers.

We thank the reviewer for their suggestions and have responded to each comment below.

Suggested improvements

In this study cohort of 10 subjects with 16 recording sessions, there were 6 implantations the right, 3 on the left and only 1 bilaterally. Although briefly mentioned in Section 4.5, it is not clear if the contact localization was performed first in patient's brain space, before

registration to a common template, or localized in template space. The contact localization after warping into template space may introduce localization error.

We confirm that localization was first performed in each patient's specific brain space before registration to a common template (with corresponding transformation of contact coordinates to the template space). Contacts were localized with post-implantation structural MRI images and CT scans and registered to the high-resolution pre-implantation structural MRI space specific to each patient, before a final transformation to template space. We have clarified this detail in the text describing localization:

In 4.5 Imaging and contact localisation

... Preoperative structural MRI images were co-registered to post-implantation structural MRI images, guided by post-implantation computed tomography (CT) scans (in-plane resolution 0.5 x 0.5 mm, slice thickness 1-3 mm), using custom MATLAB (Mathworks, Natick, MA) scripts and affine registration (FSL's FLIRT, Jenkinson et al., 2012), and all images were processed to 1mm³ isotropic resolution. Visual comparison with intra-operative photographs was conducted to verify accuracy. **Electrode contacts were localised with CT-guided post-implantation structural MRI images, then transferred onto the pre-implantation MRI space specific to each participant.** Each participant's structural MRI was then co-registered to the MNI template brain using ANTs software \cite{avants2009advanced}, and resulting contact locations are shown against the template (see Figure~S\ref{suppl:contacts}A).

Throughout the manuscript there was also no mention of segregation of ROIs based on laterality and it was unclear if there was any significant difference between left and right ROIs in the findings. Such an analysis should be performed, and at least acknowledged in the limitations if the sample is not powered sufficiently to examine the results.

We thank the reviewer for their comment on laterality differences in our ROIs. Indeed, as the reviewer pointed out, for the purpose of increasing statistical power we originally had combined between left and right analogs for a given region. Nevertheless, the reviewer's comments intrigued us as to whether the effects we reported were primarily driven by a particular hemisphere. Although the sample of contacts was not sufficient to statistically compare between left versus right differences in a given ROI, we took an alternative analytic approach to address this question descriptively. The logic is that if a reported effect is driven by primarily one hemisphere, then decoding a particular variable from only contacts in that hemisphere would be expected to yield similar effects. For this follow up analysis, we excluded the vmPFC (subcallosal) and cingulate gyrus ROIs because they were situated at the midline, and the putamen and supramarginal gyrus because all contacts were already lateralized. While there were 2/30 contacts in the left supramarginal gyrus, because of our bipolar referencing scheme this meant we would only have one neural feature for decoding from left supramarginal gyrus preventing a comparable multivariate analysis. The remaining ROIs were separated into left and right sub-ROIs. We have also included in the supplementary section a breakdown of the sample size of each ROI by the number of patients, total contacts, and contacts in each hemisphere.

ROI	# Participants	# Contacts	# L	# R
Ant. Insula	6	14	4	10
Post. Insula	5	10	6	4
OFC	10	72	30	42
vmPFC	5	12	6	6
Frontal Pole	8	158	58	100
Putamen	4	8	0	8
Hippocampus	10	62	20	42
Amygdala	8	32	12	20
Cingulate Gyr	6	16	2	14
Supramarg. Gyr	6	30	2	28
Angular Gyr	5	14	6	8

Supplementary Table 1: Sample of patients and contacts per ROI

Because the purpose of this analysis was to dissect the contribution of sub-regions to our reported findings, we examined the decoding accuracy of the (lateralized) sub-sample of contacts in a given ROI for relevant variables during the period previously reported to be significant. For example, separately for the left and right Frontal Pole, we examined the average decoding accuracy during the window of 0.236 - 0.345 sec after the onset of card 2 since this is when we originally were able to decode outcome using the bilateral Frontal Pole. To the extent that one hemisphere is primarily driving the reported effect, we expected to see differential average decoding accuracy during this period. Following our response to reviewer #1 above regarding reporting uncertainty around estimates, we bootstrapped the test set to compute 95% CI for each sub-region and variable.

The supplementary section now includes an additional section describing our approach to examining lateralization effects.

8.2 Lateralization analysis

For our primary decoding analysis we pooled across left and right hemisphere analogs of a given ROI (e.g. L and R amygdala) to increase the statistical power of our analyses, particularly in ROIs for which we had relatively fewer numbers of contacts (See Supplementary table \ref{tabS1}). Here we interrogate further the relative contribution of left versus right hemisphere contracts for a given ROI in our reported decoding results. Given the limited number of contacts that remain in a ROI when considering only one hemisphere, we provide a descriptive rather than inferential analysis of whether our reported effects were driven primarily by one hemisphere or relied on information provided bilaterally. The logic of this follow-up analysis was that if a reported effect is driven by primarily one hemisphere, then decoding a particular variable from only contacts in that hemisphere would be expected to yield similar effects. If on the

other hand a reported effect relied on bilateral information, decoding that variable only from one hemisphere would result in a relatively lower accuracy score.

For this follow up analysis, we excluded the vmPFC (subcallosal cortex) and cingulate gyrus ROIs because they were situated at the midline, and the putamen and supramarginal gyrus because contacts were already lateralized to the right hemisphere (See Supplementary table \ref{tabS1}). While there were 2/30 contacts in the left supramarginal gyrus, because of our bipolar referencing scheme this meant we would only have one neural feature for decoding from left supramarginal gyrus preventing a comparable multivariate analysis. The remaining ROIs were separated into left and right sub-ROIs. Because the purpose of this analysis was to dissect the contribution of sub-regions to our reported findings, we examined the decoding accuracy of the (lateralized) sub-sample of contacts in a given ROI for relevant variables during the period previously reported to be significant. To the extent that one hemisphere is primarily driving the reported effect, we expected to see differential average decoding accuracy between left and right ROIs (compared to the bilateral effect reported in the main text) during this period. While we did not conduct statistical tests (e.g. t-tests or non-parametric equivalents such) given the aforementioned limitation of low sample sizes upon dividing bilateral ROIs into left and right regions, we conducted a bootstrap analysis to provide 95% confidence interval estimates of uncertainty using the approach described in \ref{decode_CI}.

Using this descriptive approach, we found that the majority of the effects we previously reported were driven bilaterally for each corresponding ROI (Figure S~\ref{suppl:laterality}), with a few potential exceptions. Amygdala contribution to outcome decoding was predominantly driven by the L amygdala (95 CI [-0.127 -0.005], Δ ROC AUC relative to bilateral amygdala), and RiPE decoding in anterior insula showed sensitivity, albeit weaker, to L anterior insula (95 CI [-0.095 0.005], Δ ROC AUC relative to bilateral anterior insula).

With a corresponding supplemental figure depicting results of our laterality analysis:

Differential contributions of left and right hemisphere ROIs to decoding: A) outcome, B) expected value, C) expected risk, D) reward prediction error, E) risk prediction error. Δ ROC AUC represents the difference between the decoding accuracy (ROC AUC) of left and right subsets of each ROI and the originally reported bilateral ROI (centered at 0; dashed red line), averaged around the significant period respective to each variable and ROI reported in the main findings. 95 CI of the bootstrapped test accuracy (shown in the violin and scatter distributions) are depicted with vertical lines, boxes depict the interquartile range, and points represent the median.

We refer to this additional supplementary section in section 4.9 Region of interest definition:

... Contacts across bilateral ROIs were aggregated for the results reported in the main text, though see Supplementary section \ref{laterality} for an additional investigation into laterality differences.

And in section 3: Discussion:

...Specifically, regions that are under-sampled given a fewer number of recording sites may yield non-significant decoding accuracy due to decreased statistical power, especially if the true underlying neural representation is spatially widespread but weakly informative when only surveying each recording site individually \cite{etzel2013searchlight}. **Indeed we chose to**

aggregate bilateral ROIs with this consideration of increasing our sampling of each ROI, though we conducted additional analyses probing laterality differences in our reported decoding results across ROIs (see Supplementary section \ref{laterality})...

In Section 4.8, line 508, there is a minor error in ... “ given in 8/10 of the remaining”... this should be 8/9.

We thank the reviewer for noticing this error, and have corrected the text.

There are, in addition, numerous references in the Supplementary material and figures that are mislabeled or left black with a [?] which should be corrected.

Similarly we have fixed the broken references to supplementary material and figures.

What a cortical-cortical evoked potential analysis strengthen their cross-correlation approach to connectivity? Was this performed?

We thank the reviewer for this suggestion and have reviewed further the advantages of a cortico-cortical evoked potential (CCEP) analysis for understanding the effective connectivity between regions in intracranial electrophysiology studies. CCEP refers to a procedure in which a current is injected between at a cortical site, triggering a local electrical response around the stimulation target as well as remote regions; the measure response at distal sites (the “evoked potential”) is in proportion to the strength of effective connectivity between the two regions (Matsumoto et al., 2004; Keller et al. 2014). As such, CCEP analyses have high potential to strengthen inferences about connectivity above and beyond cross-correlation approaches to connectivity. For example, the direction of connectivity is not inferred by time-lagged correlations between regions, but rather directly manipulated and assessed in vivo. Further, relative to non-invasive methods for measuring connectivity, as an intracranial electrophysiology technique CCEP continues to have high spatio-temporal resolution and precise localization of downstream stimulated regions. Indeed, CCEP is a promising approach to address the concerns raised by reviewer 1 regarding our use of causal language in text, where the reviewer commented that casual relationships cannot be interpreted without, for example, perturbations of the system. (We have amended the casual tone throughout the text in response to this comment, please see above). CCEP could be used to introduce such perturbations in order to understand the casual and directed relationship between different ROIs.

Unfortunately, CCEP manipulations were not included in the data acquisition protocol for our sample and thus additional inferences about connectivity afforded by this method cannot be made in the context of the present work. Typically, CCEP manipulations are conducted after enough seizures have been observed for clinical purposes (e.g. identifying seizure foci; Keller et al., 2014) whereas the administration of experimental tasks including ours occurs during the monitoring period. Nonetheless, we have added a paragraph in the Discussion section in which we include this interesting suggestion by the reviewer, discuss in more detail the paucity of

cross-correlation effects in our analysis, and propose the specific ways in which CCEP manipulations can improve inference about connectivity for future studies:

In 3. Discussion:

Indeed our finding that computations are structured in time across areas raised the question of whether the functional connectivity between different regions encoded computational information. Across all variables, we did not find significant differences in the cross-correlation between regions between high and low levels of that variable (see Supplementary section \ref{xcorr}). However, our approach is only one of several ways to address the question of directed functional connectivity relevant to processing these computations. Future work in human electrophysiology is well-positioned to elucidate the mechanisms driving directed connectivity. For example, cortico-cortical evoked potential manipulations \cite{matsumoto2004functional, keller2014mapping} directly perturb neural activity and therefore might provide a casual picture of how computational information propagates across the brain.

Clarity, Context and References

The authors have presented several previous work and current understanding of reward / risk computation and referenced them in the introduction, and discussion and discussed comparisons of the current findings to other authors and groups.

Reviewer #3 (Remarks to the Author):

In their article titled "Temporally Organized Representations of Reward and Risk in the Human Brain," Man et al. investigate the temporal and spatial dynamics of risk and reward representation, as well as their associated prediction error, using intra-EEG. They employed a simple gambling task where participants had to predict whether the second of two sequentially-drawn playing cards would have a higher or lower numerical value than the first card. Correct guesses were rewarded with points, while incorrect guesses resulted in a loss of points. The authors defined five variables of interest: expected value (EV) (given guess and card 1), outcome (OUT) (at card 2), expected risk (E.Risk) at card 2, reward prediction error (RePE) at card 2, and risk prediction error (RiPE) for each card. The results can be summarized as follows:

1. Expected value was observed in the ventromedial prefrontal cortex (vmPFC), amygdala, and hippocampus.

2. Outcome was detected in multiple brain regions, exhibiting a postero-anterior gradient of temporal representation. The authors found directed functional connectivity from the frontal pole to the orbitofrontal cortex (OFC) and from the OFC to the vmPFC.

3. Expected risk was identified in the OFC but not significantly in the anterior insula. It was also found in the posterior insula and intraparietal lobule.

4. Reward prediction error representation was observed in the OFC and anterior insula.

5. Risk prediction error representation was found in the anterior insula and OFC. Notably, RiPE representation in the anterior insula occurred later than RePE. An additional result indicated that RePE representation is relevant for the RiPE of card 2.

Overall, these results provide valuable insights into the temporal dynamics of risk and reward-related neural signals. However, I have a series of general and specific concerns. There are several missing details, and the figures fail to effectively support the article. I will provide more detailed comments below.

We thank the reviewer for their careful read of the paper reflected by their precise summary of our key findings. We also thank the reviewer for their constructive comments. We have taken efforts to address the reviewer's comments point by point, and we feel that the corresponding revisions improve the quality of the manuscript.

1) General concerns

a) Regarding ROIs:

• It would be helpful to specify the number of contacts belonging to each ROI. Including this information will provide transparency and aid readers in understanding the sample size for each region of interest.

We agree with the reviewer, and similar comments by other reviewers indicate that this information is critical to providing appropriate context to our sample. We have now added a supplementary table which depicts the number of contacts and number of patients contributing to the decoding analysis of each ROI:

ROI	# Participants	# Contacts	# L	# R
Ant. Insula	6	14	4	10
Post. Insula	5	10	6	4
OFC	10	72	30	42
vmPFC	5	12	6	6
Frontal Pole	8	158	58	100
Putamen	4	8	0	8
Hippocampus	10	62	20	42
Amygdala	8	32	12	20
Cingulate Gyr	6	16	2	14
Supramarg. Gyr	6	30	2	28
Angular Gyr	5	14	6	8

Supplementary Table 1: Sample of patients and contacts per ROI

• The size of the Frontal pole ROI appears to be much larger than any other region, and the OFC seems relatively posterior according to Figure 1B. Authors should aim to define ROIs with a similar number of recording sites and ensure that the ROIs are comparable in terms of anatomical size or delineation. For example, it is unlikely that dlPFC and anterior lateral OFC compute the same variables, so it is important to address this discrepancy.

We thank the reviewer for raising this concern; a similar point was raised by reviewer #1 above. Please see our response to reviewer #1's point 2 above.

Further, we were intrigued by the point raised here by reviewer #3 as to whether sub-regions of a coarse (albeit anatomically defined) prefrontal cortex ROI (e.g. dlPFC) and OFC ROI (e.g. the anterior lateral division) might have differential contributions to representing our variables of interest. Motivated by the reviewer's question, we have now conducted additional analyses probing whether and which spatially distinct subregions of our large ROIs (Frontal Pole and OFC) were driving the main decoding results we originally reported.

We now include a new section in the supplementary section documenting this additional investigation. In section 8.1 Prefrontal sub-regional analysis:

We originally defined anatomical ROIs according to the Harvard-Oxford (H-O) probabilistic atlas (see Section \ref{ROI}) which included two spatially extensive ROIs (OFC and Frontal Pole). Given our a priori interest in reward and risk representations in prefrontal cortex, the large size of these ROIs, and work documenting functional heterogeneity of these regions, particularly with respect to reward signals \cite{wallis2010heterogeneous,kringelbach2004functional}, we sought to further investigate which sub-region(s) of each of these two large ROIs drove the originally reported decoding effects respective to that ROI. For example, as we were originally able to decode Outcome, E.Risk, RiPE, and RePE in OFC, we probed here which sub-regions of OFC were driving the decoding of those variables. Following our original approach to defining ROIs, to remain free of bias in our sub-regional ROI definition we relied on an independently derived atlas produced by a connectivity-based parcellation \cite{neubert2015connectivity} of the OFC (Figure S~\ref{suppl:subROI}A) and Frontal Pole (Figure S~\ref{suppl:subROI}B). Within each ROI, we report the number of ROI contacts that spatially comprise each sub-division as determined by the Neubert atlas (Supplementary table \ref{tabS2}).

ROI	Sub-region	# Subs	# Contacts
OFC	Area 11	6	11
	Area 13	9	37
	Area 25	3	3
	14m	1	1
	47m	1	1
	47o	9	19
Frontal Pole	Area 11	4	5
	Area 13	3	4
	FPm	2	3
	FPl	8	53
	8m	3	12
	9m	5	9
	11m	5	11
	47m	8	36
	47o	6	13
Undefined	5	12	

Supplementary Table 2: Sub-regional division of OFC and Frontal Pole. # Subs denotes the sample size of subjects going into each sub-region. # Contacts is the number of recording sites per sub-region. Certain sub-regional labels are delineated within both OFC and Frontal Pole ROIs (e.g. Area 11). In this case contacts are labeled hierarchically (H-O ROI / Neubert sub-ROI; e.g. OFC / Area 11 versus Frontal Pole / Area 11). Importantly, each contact is only included in one unique sub-division. In other words, there are no overlapping contacts across decoding models. The “undefined” label refers to contacts originally within the Frontal Pole ROI (H-O atlas) but are not within any sub-ROI label of the Neubert atlas; they were not included in analyses.

We now also include a supplementary figure depicting the contact assignments to sub-regions of OFC and Frontal Pole:

Supplementary Figure 8: Contacts locations in sub-regions of (A) OFC, and (B) Frontal Pole, distinguished by labels from the Neubert atlas \cite{neubert2015connectivity}. Different colors identify distinct subregions, and contours show boundaries of the H-O atlas ROI.

Continuing in Section 8.1: Prefrontal sub-regional analysis, we document our analytic approach and results:

We took two analytic approaches to probe the sub-regional contributions to our reported decoding effects in OFC and Frontal Pole. First, we conducted a ‘model lesion’ analysis in which we dropped the features of the decoding model corresponding to the contacts within a given sub-region (e.g. Area 13 from the Neubert atlas) and measured the change in decoding accuracy compared to a ‘full’ decoding model containing all contacts in the original large ROI (e.g. OFC from the H-O atlas). The rationale was that the extent to which that sub-region contributed to our originally reported decoding effects would be reflected by the drop in model performance between the nested and full model. In contrast, sub-regions that were not relevant for the decoding of a particular computational variable in that large ROI originally would exhibit relatively less change in model performance upon their exclusion. Our second approach, independent from the first and from the original decoding analysis using the larger H-O ROI, was to re-run a

decoding analysis using only the contacts from each sub-region, asking which sub-region(s) of a larger ROI could decode each computational variable, and when in the epoch they did so.

In OFC we were able to decode the reward variables Outcome and RePE in sub-regions 47o ($C_{AUC} = 0.386$, $C_{thresh} = 0.224$, $p = 0.009$, $AUC_{0.263} = 0.557$, time [0.257-0.311 sec]) and Area 11 ($C_{AUC} = 1.189$, $C_{thresh} = 0.336$, $p < 0.001$, $AUC_{0.319} = 0.648$, time [0.287-0.335 sec]), respectively, two sub-regions of lateral OFC (Figure S~\ref{suppl:subROI}A). In contrast, activity in central Area 13 could significantly decode both risk-related variables: E.Risk ($C_{AUC} = 0.432$, $C_{thresh} = 0.250$, $p = 0.013$, $AUC_{0.307} = 0.563$, time [0.281-0.327 sec]), and RiPE ($C_{AUC} = 0.598$, $C_{thresh} = 0.162$, $p = 0.002$, $AUC_{0.222} = 0.556$, time = [0.196-0.283 sec]; Figure S~\ref{suppl:subROI_res}A). Notably, we found that the temporal window within which we were able to decode each computational variable from each corresponding OFC sub-region aligned with the temporal windows originally reported using the larger OFC ROI, which in conjunction with convergent results from our 'model lesion' approach (Figure S~\ref{suppl:subROI_res}B), substantiates the interpretation that these respective sub-regions supported our originally reported OFC decoding effects.

Consistent with the implication of 47o within our original OFC ROI in outcome decoding reported above, we found that more anterior aspects of 47o within our original Frontal Pole ROI was also able to decode outcome ($C_{AUC} = 0.330$, $C_{thresh} = 0.248$, $p = 0.027$, $AUC_{0.273} = 0.573$) within a similar period [0.250-0.285 sec]. Further, outcome decoding within Frontal Pole was also driven by sub-region 9m, in which outcome could be decoded in early ($C_{AUC} = 0.491$, $C_{thresh} = 0.398$, $p = 0.032$, $AUC_{0.281} = 0.579$, time [0.248-0.299]) and late ($C_{AUC} = 0.499$, $C_{thresh} = 0.398$, $p = 0.032$, $AUC_{0.409} = 0.578$, time [0.397-0.457]) temporal clusters. Our Frontal Pole model lesion analysis conditioned on the timing of our original results did not show specificity to 47o or 9m (Figure~S\ref{suppl:subROI_res}D); instead, multiple sub-regions contributed to our original effect despite most not individually able to decode outcome. Indeed, this is corroborated by the spatial extent of relevant contacts for outcome decoding across our original Frontal Pole ROI (Figure~S\ref{suppl:subROI_res}E), and together supports the idea that outcome is held in a multivariate, distributed representation across Frontal Pole.

With a corresponding supplemental figure depicting results of our sub-regional analyses:

Supplementary Figure 9: Sub-regional contributions to originally reported decoding results in OFC and Frontal Pole. (A) Specific sub-regions of OFC contribute to the decoding of distinct computational variables (black: Outcome; blue: RePE; yellow; E.Risk, red: RiPE). **(B)** Δ ROC AUC represents the difference between the decoding accuracy (ROC AUC) of each OFC sub-region and the larger OFC ROI, averaged around the significant period respective to each variable reported in the main findings. 95 CI of the bootstrapped test accuracy (shown in the violin and scatter distributions) are depicted with vertical lines, boxes depict the interquartile range, and points represent the median. **(C-D)** Same as A and B, for Frontal Pole sub-regions. **(E)** Feature importance map of Frontal Pole shows a spatially distributed outcome representation across sub-regions. Each coloured dot represents a bipolar source which contributed to outcome decoding in Frontal Pole, with colour depicting normalized feature weight. Black dots depict contacts with negative feature weights prior to normalization.

We now point to this additional section in both the Methods and Results of the main text. In the methods, we discuss:

In methods section 4.9 Region of interest definition:

...Because the Frontal Pole and OFC ROI delineations from the H-O atlas were spatially extensive, and as a result comprised the largest number of included contacts in our sample (See Supplementary table \ref{tabS1}), we investigated whether computational variables that were able to be decoded from the multivariate activity in these regions were driven by particular nested sub-regions of these ROIs. Our analytic approach and the results of this investigation are presented in Supplementary section \ref{subregion}.

Relevant sections of the Results now refer to this additional analysis as well:

In section 2.3. Distinct modes of temporal organization between value-based computations:

... We found evidence that outcome encoding was spatially distributed in Frontal Pole (see Supplementary section \ref{subregion} and Figure~S\ref{suppl:subROI_res}D-E)...

...In further analyses, we clarify that the late-epoch outcome decoding in Frontal Pole is driven by sub-region 9m, a relatively rostradorsal aspect of the ROI (Figures~S\ref{suppl:subROI}B and S\ref{suppl:subROI_res}C).

In section 2.5. Error computations across domains share a mixture of temporal configurations

...

Our analyses of error computations across reward and risk domains revealed a further temporal configuration, in which we were able to decode multiple computational variables in overlapping periods within OFC. In this region we were able to decode all computational variables designed experimentally to arise in the period after the onset of card 2: outcome (Figure~\ref{fig3}D), RePE (Figure~\ref{fig5}C), and RiPE (Figure~\ref{fig5}F), with overlapping temporal extents [0.273-0.289 sec]. **However, in follow-up analyses investigating which spatial divisions within OFC contributed to the respective decoding of each of these variables, we found across variables that distinct sub-regions of OFC drove the reported decoding effects (Supplementary section \ref{subregion}; Figure S~\ref{suppl:subROI_res}), suggestive of intra-regional parallel processing of multiple computational variables during the same period, across sub-regions.**

In section 2.6. Anterior insula representations of reward error can predict risk error

... Indeed, when we investigated sub-regional contributions to computational representations in OFC (see Supplementary section \ref{subregion}), we found that distinct sub-regions of 47o, Area 11, and Area 13 (Figure~S\ref{suppl:subROI}A)

contributed to the parallel decoding of outcome, RePE, and RiPE, respectively (Figure~S\ref{suppl:subROI_res}A).

And in Section 3. Discussion:

... We demonstrate the importance of elucidating the timing of different types of computational representations by presenting evidence that the distributed code in anterior insula relevant for reward error precedes and directly contributes to decoding risk error. **In contrast, representations related to outcome, reward prediction error, and risk error are spatially distinct, but temporally parallel, across sub-regions of OFC.** Together, our findings highlight distinctive patterns of timing by which the brain represents reward and risk variables...

• Consistency in ROI names is essential. The manuscript currently uses various terms for the same region, such as for the OFC (which is actually the posterior OFC) : lateral OFC, and posterior lateral OFC. It is crucial to maintain consistency in ROI names throughout the manuscript to avoid confusion. I recommend using a single, agreed-upon term for each ROI and using it consistently across the entire article.

We thank the reviewer for pointing out potential confusion among readers due to inconsistent ROI naming. We now use a consistent set of ROI names throughout the paper, which is based on the labels from the Harvard-Oxford (H-O) from which we defined our anatomical ROIs. We made slight modifications to these default labels, to better speak to the relevant literature on value and uncertainty. Specifically, we refer to the 'Subcallosal Cortex' from the H-O atlas as 'ventromedial prefrontal cortex (vmPFC)', and to 'Frontal Orbital Cortex' as 'orbitofrontal cortex (OFC)'. We have amended the corresponding section in the Methods to clarify this:

4.9 Region of interest definition

We specified a set of 5 \textit{a priori} regions of interest (ROI) spanning frontal and subcortical regions given our strong hypotheses of the role of the prefrontal cortex (PFC), insula, and amygdala in value and uncertainty based computations, using the Harvard-Oxford (H-O) probabilistic atlas to delineate contacts respective to each ROI. **The 5 \textit{a priori} ROIs included Frontal Pole, OFC ("Frontal Orbital Cortex" in the H-O atlas), and ventromedial prefrontal cortex (vmPFC, labeled 'Subcallosal Cortex' in the H-O atlas). We modified the ROI names for OFC and vmPFC from their original labels in the H-O atlas to better speak to relevant findings from previous literature that have used these labels.**

Given the existing literature on which we build in the current study and findings implicating the anterior insula specifically in these risk-related variables (Preuschoff et al., 2008, J Neuro), we divided the anterior and posterior sections of the integrated 'Insular Cortex' from the Harvard-Oxford atlas. Continuing on in section 4.9 Region of interest definition:

We also hypothesized that the anterior area of the insula (Insular Cortex) would be particularly relevant for risk representations given findings from prior work \cite{preuschoff2008human}

using the same experimental paradigm. Since the H-O anatomical mask for the Insular Cortex comprised voxels spanning both anterior and posterior aspects of the insula, we split the anatomical mask at the mid-point of the anterior-posterior (Y) plane of the mask ($y = 4.66$) ensuring that all contacts ascribed to anterior and posterior were respectively located on positive and negative coordinates in template space along the Y plane.

We agree with the reviewer that the large spatial extent of our Frontal Pole and OFC ROIs means that our “OFC” ROI refers to posterior aspects of the orbitofrontal region whereas the Frontal Pole subsumes multiple divisions of anterior prefrontal cortex, including the anterior OFC. We have removed the description of this allocation of anterior and posterior OFC between the H-O ROIs in this methods section because, as the reviewer points out, it can be confusing to ascribe multiple labels to the same ROI. Instead, and as suggested by the reviewer from their previous point, we have now included an additional analysis surveying the contribution of sub-divisions of the ‘Frontal Pole’ and ‘OFC’ H-O-based ROIs towards their respective effects. In our response to the previous point by the reviewer we detail our analytic approach to examining these subdivisions of the Frontal Pole and OFC. We have included this description in the supplementary section, where we also describe the subregional ROI labels derived from the corresponding connectivity-based parcellation study (Neubert et al., 2015, PNAS).

b) Regarding Timings:

• The timing description for the variables depicted in Figure 1C is currently confusing. The label "before/at card" is imprecise and lacks clarity. It would greatly benefit the reader if the authors unpacked this figure and provided a clear explanation of the timing for each variable. For instance, it should be clarified that EV will be investigated at card 1 onset, E.Risk1 before onset (constant), and RiPE1 at card 1 onset. Similarly, E.Risk2 will be investigated before card 2 onset, while RePE and RiPE will be examined at card 2 onset. By providing this clarification, readers will have a better understanding of the temporal sequence of events.

We thank the reviewer for their suggestion on improving the visual communication of the model variables in Figure 1C. Following the recommendations of the reviewer in making the unfolding of computations over the course of a trial more clear (i.e. the timing of variables with regard to trial events) as well as continuing to depict predicted variance of the computational terms (e.g. as a function of the card 1 drawn and outcome), we have rehauled the figure panel. We believe it now better displays which variable is investigated at each trial event given the computational theory and thus conveys the temporal sequence:

In the caption for Figure 1: ...**(C) Variation in computational variables as a function of the drawn number of card 1, conditioned on an initial guess that card 2 will be lower as illustrated in the example trial depicted in A. Each column anchors at a trial event for the events analyzed in the study (the period before Card 1 is not included in the neural analysis but depicted here for completeness). Model predictions of the computational variables are shown at the respective trial event predicted by the model. Reward-related variables including expected value (EV; green) and reward prediction error (RePE; blue) are shown in the top row. Risk-related variables including expected risk (E.Risk; yellow) and risk prediction error (RiPE; red) are in the bottom row. Variables after the onset of card 2 are conditioned on the experienced outcome (win or loss).**

• Conclusions on timings are valid within time windows of interest (at onset of card 1 or onset of card 2), but not across time windows as variables are decoded only in their specific time windows. In other words, the conclusions drawn about temporal dynamics, apart from outcome decoding and RePE representation informative for RiPE representation, may not be as convincing. To strengthen the results, it would be more persuasive to systematically apply decoding techniques to three time windows for each ROI (guess time, card 1 onset, card 2 onset). This can be visualized in the schematic representation of the results I made (to understand the results). This approach would demonstrate that decoding occurs specifically within the expected time window. Once this specificity is established, the authors can delve deeper into the temporal comparisons within each particular time window. Conducting such analyses would provide reassurance that observed differences are not merely false positives. It is worth noting that the authors have appropriately applied corrections for multiple comparisons.

This is an interesting and thought-provoking point raised by the reviewer. Our original motivation for directing our decoding analyses to particular epochs respective to each computational variable was driven by our *a priori* hypotheses about when computations would arise given the design of the task, in conjunction with the computational model. For example, we held prior

hypotheses that expected value would be computed in the epoch following the presentation of card 1 because the information conferred by card 1 allowed the computational of the conditional probability (conditioned on the initial guess and now the value of card 1) of eventually receiving a reward on that trial, expecting over all possible remaining values of card 2. Expected risk was hypothesized to be represented later than expected value, given previous work documenting the relative delay in risk signals (Li et al., 2016; Fiorillo et al., 2003); as such we directed our decoding analysis of expected risk to be in the subsequent period between card 1 and card 2. We now clarify our specific predictions about time windows respective to each computational variable with the amended Figure 1C, as suggested by the reviewer's previous comment.

Given that we held these strong prior predictions, we had originally decided to conduct directed decoding analyses for each variable at specific epochs. It's worth noting that this approach of tethering computational variables to hypothesized epochs conferred a methodological benefit of temporally separating, and thus de-correlating, certain variables such as EV and Outcome (as illustrated in Figure S6), allowing us to have greater specificity interpreting our decoding results. We agree with the reviewer's perspective that our original approach allows us to make conclusions about timing within epochs of interest, but does not speak to the epoch specificity of a given variable across the trial more generally. We have now constrained and qualified the scope of temporal organization in the manuscript to reflect this nuance. For example in section 2.2: Decoding computational variables with multivariate neural signals

...

Our design matrix for single-trial decoding allowed us to assess the decodability of each computational variable at its respective point of occurrence within a trial. **While this allowed us to leverage the sequential structure of the task to temporally separate variables, it constrains our investigation of temporal organization to within the specified time windows respective to each variable...**

And in section 3: Discussion

...

Critically, we leveraged an experimental design in conjunction with a normative model of reward and risk which separated, in time over the span of multiple sequential trial events, when the information relevant for each computational process was available. We could therefore examine specific periods of neural data with respect to each computational variable in the span of a single trial. **It is worth noting, however, that our approach of examining specific, hypothesized periods for a given variable constrains our conclusions about the temporal organization of that variable across regions to within our specified time windows, and does not preclude the possibility of a variable being represented at other periods in a trial.** Nevertheless, this approach allowed us to look at the unique decodability of a set of computational variables which are not often de-coupled in other experimental contexts...

We did not originally conduct our decoding analysis over all possible epochs for each computational variable and ROI. Such an exploratory approach would have drastically increased the number of statistical comparisons: 11 ROIs x 4 epochs x 5 variables would have resulted in 220 decoding analyses, each entailing decoding across time points. This amounts to

a 4-fold increase in the number of statistical tests compared to what we had originally conducted in the paper (165 additional tests). As the reviewer points out, we took great care originally to correct both within epoch, using the maximum cluster statistic approach (Maris et al., 2007) to keep family-wise error < 0.05 , and across ROIs to keep the false discovery rate below 0.05 (note that correcting across epochs in our original analysis does not apply as each variable was specifically tested at only 1 epoch). Given the much greater number of tests required to pursue the question of the epoch specificity of each computational variable and ROI, and consequently the more stringent correction for multiple comparisons required to keep the false discovery rate at our tolerance of 0.05, we feel that this investigation is beyond the scope of the current paper as we did not have enough data for the statistical power required to address it (please see our response to reviewer #1's point 1 regarding this issue of sample size as well as the corresponding new changes in our manuscript explicitly addressing this limitation, above). Further making explicit this particular limitation of our current manuscript:

In 3. Discussion:

... Our findings should be contextualized in light of these considerations, and future work endeavoring to pool across larger samples, for example by acquiring data on the same task across multiple sites, will be important to mitigate these limitations. **Larger samples will also be conducive to expanding our findings on temporal organization, for example by validating that a particular representation is specific to the time window we specified for that variable in our analyses, revealing whether the encoding of a variable can extend over the course of multiple periods within a trial, and elucidating phasic versus tonic properties of distinct representations.**

We nonetheless appreciate this issue raised by the reviewer, as well as the reviewer's efforts to convey their point clearly. As such, during the revision process we conducted the analysis proposed by the reviewer in which we ran statistical analyses on each of the four epochs within a trial (guess, card 1, card 1 off, card 2) for each computational variable (EV, Outcome, Erisk, RePE, RiPE) and each of our 11 ROIs, keeping in mind the caveats with such an analysis discussed above. Across these tests, we did not find any additional temporal clusters of significant decoding that survived multiple comparisons correction, adopting the same procedure for multiple comparisons correction we had originally employed in the manuscript for fair comparison (i.e., correcting to keep the false discovery rate below 0.05).

c) Regarding analyses:

• Why did the authors choose to do multivariate analyses? I understand the rationale for binary variables, but less for continuous variables. What are the results for univariate analyses with continuous variables, are they similar? In any case, justification for the choice of multivariate analyses should be clearly stated somewhere.

We thank the reviewer for raising this interesting and important discussion point. Ultimately multivariate and univariate methods approach the question of how the brain implements these computations (or other kinds of information) from different perspectives. We were interested in

whether a particular region held a representation of the computational variable of interest, and ultimately to elucidate the (temporal) properties of these representations. Our motivation for using a multivariate approach is that it would be sensitive to representations coded in the distributed neural activity within a region, across our recording sites. In contrast, univariate approaches assess whether activity in an individual contact correlates with the variable, separately for each contact or after averaging across contacts to characterize a ROI. This latter univariate approach necessarily assumes that the information related to a computation is encoded in a single dimension in neural space. On the other hand a multivariate approach, while retaining sensitivity to unidimensional neural codes (Davis et al., 2014), is further sensitive to representations encoded in a multidimensional neural space. Multivariate analyses are therefore able to uncover representations of complex variables likely to comprise multiple features and be encoded in the brain along multiple dimensions (Haynes and Rees, 2006). We argue that this is particularly relevant for the types of computational variables investigated in our work (see also Kahnt, 2018). For example, RePE is composed of multiple input computations (EV and Outcome) which are encoded in distinct dimensions of neural activity. We now provide an expanded justification of the multivariate approach:

In Section 2.2. Decoding computational variables with multivariate neural signals

We aimed to investigate the coordination of reward and risk representations across multiple brain regions. Using a multivariate decoding approach, we examined the timing of each computational variable's representation within a region of interest (ROI). **We took a multivariate analytic approach given our objective of describing neural representations of complex computations variables. We reasoned that the information pertaining to these reward and risk computations were likely to involve multiple dimensions of neural representation given the multifaceted nature of the computational information** \cite{kahnt2018decade}, as such, a multivariate approach examining how information is coded across spatially distributed sites within a ROI would provide greater sensitivity \cite{haynes2006decoding, davis2014differences} to uncover neural representations and thus their temporal properties.

Further, we were interested in whether there were similar temporal patterns in the representations, across regions, of different computational variables. For example, in section 2.5 we report similarities in the temporal profiles for RiPE and RePE. We thus strove to ensure that the analytic approach was matched across variables. Because Outcome is a binary variable in our task (i.e. rewards have no magnitude), we binarized other computational variables to match the multivariate analysis to outcome. Specifically, this allowed us to use a constant decoding model (regularized logistic regression) and accuracy metric (ROC AUC) across all variables. We especially felt the binarization of our variables was justified in that not all variables were truly continuous; for example, given the task structure and the computational model, EV has 10 unique values (it is conditioned on the value of the first card of a set of 10); similarly E.Risk2 comprises 5 unique values as it takes as input the square of the EV. For these reasons we binarized the target variables for decoding and evaluated whether neural representations could differentiate between high and low levels of the variable.

• The directed connectivity analyses were conducted solely for outcome, but it would be more consistent to extend these analyses to EV, RePE, and RiPE2 as well. Including directed connectivity analyses for these variables would provide a comprehensive assessment of the functional connectivity across the different ROIs. Ensuring consistency in the choice of variables for directed connectivity analyses would enhance the overall validity and comprehensiveness of the study's conclusions.

We thank the reviewer for this comment and now present results for the cross-correlation analysis for all variables. We agree with the reviewer that including the results of this analysis for all variables is a more comprehensive assessment. We now present these results in the cross-correlation section, which is fully reported in the supplementary section, and no longer in the main results.

The motivation for moving the cross-correlation analyses and results to the supplementary section is driven by a view towards ensuring all results reported in the main text are robust; we felt that our cross-correlation analyses were not sufficiently strong to include in the main text. In our originally reported cross-correlation effects we found different correlations between outcomes (i.e. reward vs no-reward) for early signal in the frontal pole and later signal in OFC ($t(334) = -2.226$, $p = 0.027$), and for early signal in OFC and later signal in vmPFC ($t(24) = 2.165$, $p = 0.041$). This analysis is akin to testing the simple effects of the difference in cross-correlation between outcome levels, at positive- and negative-lagged domains of the cross-correlogram. However, a statistically more rigorous approach would be to test the two-way interaction between lag domain (positive vs negative) and level of the variable (high vs low). We have now done so and the interaction terms corresponding to these effects are not significant.

We now explain this reasoning in more depth. In supplementary section 8.3 Cross-correlation analysis:

...We summed the average z-values separately for high and low trials of the variable, and for negative and positive lag regions of the resulting cross-correlogram. **For statistical analyses, we used a two-way ANOVA to test for the interaction between levels of the computational variable of interest (high / low) and lag regions in the space of the cross-correlogram (positive / negative). We also conducted t-tests at each lag region to examine differences in the average cross-correlation between high and low levels of the computational variable of interest.**

...**Across all computational variables and their respective pairs of ROIs, we did not find significant interactions between lag and variable level in the cross-correlation measure (see Supplementary tables \ref{tabS3} to \ref{tabS7}). When we examined differences in cross-correlation between high and low levels of the computational variable separately at positive and negative lags, we found that there were different correlations between early signal in the frontal pole and later signal in OFC across observed outcomes ($t(334) = -2.226$, $p = 0.027$). Similarly, early activity in OFC contacts predicted later vmPFC activity**

in an outcome-dependent manner ($t(24) = 2.165$, $p = 0.041$; Figure~S\ref{suppl:xcorr}C). However as the interaction tests respective to each of these simple effects were not significant, we refrain from making more interpretations about directed connectivity or causal relationships between regions.

For this reason we take caution in making inferences from the simple effects, but for completeness as suggested by the reviewer we include all cross-correlation results for all variables in the supplement:

Seed ROI	Target ROI	Interaction	Neg. Lag	Pos. Lag
vmPFC	HPC	F(1,4) = 2.850, p = 0.167	t(19) = -0.381, p = 0.707	t(19) = 0.965, p = 0.347
Amygdala	vmPFC	F(1,2) = 0.251, p = 0.666	t(11) = 0.160, p = 0.876	t(11) = -0.824, p = 0.427

Supplementary Table 3: Cross-correlation between temporally adjacent ROIs for expected value

Seed ROI	Target ROI	Interaction	Neg. Lag	Pos. Lag
OFC	Frontal Pole	F(1,7) = 0.688, p = 0.434	t(334) = -0.190, p = 0.849	t(334) = -2.226, p = 0.027
vmPFC	OFC	F(1,4) = 4.493, p = 0.101	t(24) = 1.306, p = 0.204	t(24) = 2.165, p = 0.041
Amygdala	vmPFC	F(1,2) = 2.407, p = 0.261	t(11) = 0.019, p = 0.985	t(11) = -1.231, p = 0.244
HPC	Amygdala	F(1,7) = 2.686, p = 0.145	t(35) = -0.757, p = 0.454	t(35) = 0.483, p = 0.632
Angular Gyr	HPC	F(1,4) = 0.389, p = 0.567	t(24) = 0.582, p = 0.566	t(24) = 0.470, p = 0.642

Supplementary Table 4: Cross-correlation between temporally adjacent ROIs for outcome. Simple effects in which $p < 0.05$ are bolded, but their respective interaction terms are not statistically significant.

Seed ROI	Target ROI	Interaction	Neg. Lag	Pos. Lag
vmPFC	HPC	F(1,4) = 2.850, p = 0.167	t(19) = -0.381, p = 0.707	t(19) = 0.965, p = 0.347
Amygdala	vmPFC	F(1,2) = 0.251, p = 0.666	t(11) = 0.160, p = 0.876	t(11) = -0.824, p = 0.427

Supplementary Table 5: Cross-correlation between temporally adjacent ROIs for expected risk

Seed ROI	Target ROI	Interaction	Neg. Lag	Pos. Lag
Cingulate Gyr	OFC	F(1,5) = 0.106, p = 0.758	t(19) = -0.198, p = 0.845	t(19) = 1.408, p = 0.175
Cingulate Gyr	Ant. Insula	F(1,2) = 0.737, p = 0.481	t(2) = -3.339, p = 0.079	t(2) = -0.613, p = 0.602
Cingulate Gyr	Post. Insula	F(1,2) = 0.015, p = 0.914	t(2) = 0.534, p = 0.647	t(2) = 0.930, p = 0.450

Supplementary Table 6: Cross-correlation between temporally adjacent ROIs for reward prediction error

Seed ROI	Target ROI	Interaction	Neg. Lag	Pos. Lag
OFC	Ant. Insula	F(1,5) = 2.550, p = 0.171	t(25) = -0.445, p = 0.660	t(25) = 0.263, p = 0.795
OFC	Putamen	F(1,3) = 1.634, p = 0.291	t(13) = -0.841, p = 0.415	t(13) = 0.193, p = 0.85

Supplementary Table 7: Cross-correlation between temporally adjacent ROIs for risk prediction error

We have similarly expanded the corresponding figure to include all variables:

Supplementary Figure 11: Cross-correlation analysis between pairs of ROIs respective to the decoding results for each variable: A) EV, B) E.Risk, C) Outcome, D) RePE, E) RiPE. Top rows in each panel depict the shuffle-corrected cross-correlogram with traces and shaded areas depicting the mean and sem normalized cross-correlation over across-ROI contact pairs. The bottom rows depict a summary of the cross-correlogram calculated by integrating over the area under each mean trace, separately for low and high trials of the respective variable and for negative and positive lags.

d) Regarding computational variables:

• When it comes to the naming of variables, it is essential to ensure clarity and consistency. While it appears that the variables are investigated in a specific order, it can be confusing to determine whether RiPE refers to RiPE1 or RiPE2 when examining the equations. To avoid ambiguity, I would recommend to clearly state somewhere in the manuscript whether RiPE refers to RiPE1 or RiPE2 consistently throughout the text. Alternatively, authors could choose to consistently use either RiPE2 or RiPE1, as well as E.Risk1 or E.Risk2, to maintain clarity and avoid confusion for the readers.

We thank the reviewer for pointing out this potential source of confusion for the readers, and have taken the reviewer's advice to clearly state in the manuscript that indeed E.Risk and RiPE refer to E.Risk2 and RiPE2, respectively. This is done at the start of the decoding results section:

In 2.2 Decoding computational variables with multivariate neural signals

... Note that while the computational model describes an expected risk term twice (before and after card 1 is shown; see \ref{comp_model}), because expected risk before card 1 did not vary across trials it was not included in the decoding analysis. Consequently all references to E.Risk in the results describe E.Risk after the onset of card 1 and before the onset of card 2 (i.e., $E.Risk_{\{card1Off\}}$). Similarly risk prediction error (RiPE) in our decoding results refer to the RiPE computed after the onset of card 2 only ($RiPE_{\{card2\}}$) due to our process of ensuring computational specificity by orthogonalizing variables occurring at the same trial event (see \ref{feat_preproc}).

We now also clarify the point made in section 2.2. regarding the effect of orthogonalization on isolating RiPE to only the period after card 2 is shown. Specifically, RiPE1 is defined as the difference between O.Risk1 and E.Risk1; however, because E.Risk1 is constant, RiPE1 is consequently perfectly correlated with O.Risk1. As such, when we applied statistical orthogonalization for correlated variables occurring at the same trial event (Outcome and RePE; O.Risk and RiPE) such that the observed terms (Outcome and O.Risk) retained shared variance components, this meant that O.Risk1 remained identical and RiPE1 had no residual variance. The implication is that our RiPE results are only driven by its variance at card 2 whereas O.Risk varied at both card 1 and card 2. For clarity, we show here the effects of orthogonalization on RiPE1 (top panel) and RiPE2 (bottom panel). In the top panel RiPE1 orthogonalized with

respect to O.Risk1 becomes a constant across trials (red line). RiPE1 and Orisk1 are perfectly overlapping (purple bars). in the bottom panel RiPE2 similarly orthogonalized still varied across trials (red line) and thus drove our reported decoding results on RiPE.

We have included this depiction of the effect of orthogonalization on RiPE1 and RiPE2 in Supplementary Figure S6. We now also describe this in the Method section:

4.10.1 Feature preprocessing

Because of colinearity between Out_{card2} and $RePE_{card2}$ (Figure~S\ref{suppl:multicolin}B), we orthogonalized the latter latent variable with respect to the former observed variable such that the residual variance in $RePE_{card2}$ was unrelated to OUT_{card2} . Likewise, we orthogonalized $RiPE_{card2}$ with respect to $O.Risk_{card2}$ (Figure~S\ref{suppl:multicolin}C). Note that despite $RiPE$ and $O.Risk$ being computed at both card 1 and card 2 onsets (see \ref{comp_model}), because $RiPE_{card1}$ and $O.Risk_{card1}$ are perfectly correlated, when we orthogonalized $RiPE_{card1}$ with respect to $O.Risk_{card1}$ the latter retained all the variance and $RiPE_{card1}$ was zeroed out (i.e., no variance retained; see Figure~S\ref{suppl:multicolin}D). The implication is that our RiPE results are only driven by its variance at card 2 ($RiPE_{card2}$; Figure~S\ref{suppl:multicolin}E) whereas variance in $O.Risk$ is driven by both card1 and card2. Finally, the per-trial report accuracy was also included as a covariate, aligned at the report response event. We regressed each feature vector against this set of covariates (excluding the variable of interest), and replaced each feature by its respective residual vector.

• The inclusion of the observed risk as a computational variable in the design matrix space raises questions since it is not tested in the results. It would be helpful to provide a justification or rationale for including this variable in the design matrix space if it is not examined or reported on in the results section. Providing clarification on the purpose or role of this variable will enhance the understanding of the experimental design and the analyses rationale.

Following from our response above, we constructed a design matrix containing all covariates other than the computational variable under consideration for decoding. Because of the design of the computational model, we were concerned that any effects purportedly related to RiPE might have been driven by its correlation to O.Risk. Indeed this consideration is often held in regards to the correlation between reward outcome and RePE (Hare et al. 2008, J Neurosci); for the same reason we similarly orthogonalize RePE with respect to outcome since they are not temporally separated (both occurring at the onset of card 2). We therefore included O.Risk in the design matrix to regress its associated variance out of our neural features when decoding RiPE, after orthogonalizing O.Risk and RiPE such that the variance of the RiPE term was only the component unique to RiPE and not shared with O.Risk. As shown in the response to the previous comment, our statistical treatment of variables resulted in O.Risk retaining variance at both card1 and card2 events. As such, decoding results for O.Risk lacked temporal specificity and was not included. We have added to the text an explicit rationale for including O.Risk in the design matrix for regressing out covariates:

In 4.1.10 Feature preprocessing

We set up an encoding model to preprocess each of our features (contacts) for the decoding model. At the within-subject level before concatenation across subjects (see \ref{decode_pseudopop}), for each feature we defined a general linear model (GLM) to regress out covariates of no interest, as well as computational variables aside from the current variable of interest for decoding. The confound design matrix of the GLM importantly included an intercept modeling the average contact activity across trials for that subject; regressing this design out of each neural feature therefore removes subject-specific variance in the neural data in preparation for our pseudo-population decoding analysis. We included indicator vectors for the onset of the cards, the offset of card 1, and mean-centered parametric vectors for respective (i.e., not currently decoded) computational variables: EV_{card1} , OUT_{card2} , $RePE_{\text{card2}}$, $E.Risk_{\text{card1Off}}$, and $RiPE_{\text{card2}}$ along with the observed risk $O.Risk$ (i.e. squared $RePE$), aligned to the trial event denoted in the subscripts. **Because of the design of the computational model, we were concerned that any effects purportedly related to RiPE might have been driven by its correlation to O.Risk. O.Risk was consequently included in the design matrix to regress its associated variance out of our neural features when decoding RiPE, after orthogonalizing O.Risk and RiPE such that the variance of the RiPE term was only the component unique to RiPE and not shared with O.Risk.**

e) Regarding Figures:

• **Figures are cited in the following order: 1A, 1C, 2A, 2B, 2C, 2E, 2F, 2D, 3A, 3B, S1, 3D, 3F, S5B, S5A, S1B, 4, S1B bottom, 5A, S2A, 5C, S2A, 3C, 5A, 5B, 5, 6A, S3, S4A, S4B, 6B, 1A, S8A, 1B, S8B, S6B, S7, S3. This is particularly confusing, and some are not referenced in the text, and supplementary figures appear in a random order.**

• Consider enhancing the visual elements: the figures could benefit from a more aesthetically pleasing design. It would be helpful to pay attention to factors such as font size, color schemes, and overall layout (alignment) to ensure a visually coherent representation that facilitates understanding of the results.

We thank the reviewer for highlighting sources of confusion regarding references to figure panels, and for their suggestions on how to improve the aesthetic design of the figures to better communicate our findings. We have made several changes to our figures in light of these suggestions.

First, figure panels are reordered to respect the order in which they are cited in the text. Similarly, we organized supplementary figures so that they are no longer in a random order but also organized in a reasonable manner given their citation order in the main text or associated supplementary section. We have also made sure all figure panels (including those in the supplementary) are referenced in the text. For example in figure 2, now the panels depicting behavioral effects have been reorganized to follow the order in which effects are presented in results section 2.1. Other figures have been similarly corrected to follow this logic.

Next, we have taken steps now to ensure aesthetic consistency throughout the figures to better communicate our findings to readers. Following editorial requests, we have replaced bar graphs (Figure 2A,B) with box-and-whisker plots to depict measures of centrality and dispersion, as well as ensure that data points are depicted. We have updated the figure captions to explicitly communicate plot features. For example in Figure 2: Behavior reflects understanding of task structure.

(A) Overall average report accuracy across the task. Boxes depict the interquartile range with the median shown with the red line. Whiskers show the range of report accuracy across participants (blue dots)...

Other figures also follow the details in the editorial requests, such as displaying measures of centrality (e.g. means), and dispersion (e.g. variances, confidence intervals). We have now also ensured that panels use the same font across panels and figures, that the font size is legible, and that panels are visually aligned for coherency. We use a consistent color set to depict each variable through all figures in the paper and supplement, following this legend: (EV: green, Outcome: black, E.Risk: yellow, RePE: blue, RiPE: red) – for example, the generalization results between RePE and RiPE is depicted with a purple panel to evoke each variable's respective color (blue and red, respectively).

2) Specific concerns (I numbered the results consistently with the summary provided at the beginning of the review):

a) Result 2-Connectivity: “While we found evidence of differential cross-correlation between high and low outcome trials in prefrontal cortex, we instead observed that areas

found to encode outcomes later led areas encoding outcomes earlier in time.” This sentence is not clear, what does it mean?

We thank the reviewer for pointing out the in clarity of our initial description of the connectivity results and agree. We have rewritten this section to enhance clarity. Now we more directly describe the pattern of cross-correlation between adjacent ROIs (i.e. early signal in region A predicts later signal B) rather than using the language of one region “leading” another:

In Supplementary section 8.3 Cross-correlation analysis:

...When we examined differences in cross-correlation between high and low levels of the computational variable separately at positive and negative lags, we found that there were different correlations between early signal in the frontal pole and later signal in OFC across observed outcomes ($t(334) = -2.226$, $p = 0.027$). Similarly, early activity in OFC contacts predicted later vmPFC activity in an outcome-dependent manner ($t(24) = 2.165$, $p = 0.041$; Figure~S\ref{suppl:xcorr}C). However as the interaction tests respective to each of these simple effects were not significant, we refrain from making more interpretations about directed connectivity or causal relationships between regions.

Also, why testing only PFC – OFC and OFC-vmPFC and not all possible combinations? Or why not reporting other possible combinations (even if not significant)? The direction of connectivity seems counter-intuitive. Authors interpret it as a top-down process in the discussion, but this should be more discussed, or more explored in the analyses.

We thank the reviewer for this suggestion, and we agree that reporting all effects for outcome-conditioned connectivity would increase the transparency of these results. We now include a table reporting all effects. Please see our response to the reviewer’s earlier comment regarding connectivity. We constrained our cross-correlation analyses to specific pairings for ROIs for each computational variable based on the temporal results we found in our main decoding analysis. This is elaborated in supplementary section 8.3. Cross-correlation analysis:

For each computational variable of interest, we defined pairs of ROIs to submit to the cross-correlation analysis based on regions in which we could decode each respective variable in our main results. In other words, we directed and constrained our cross-correlation analysis as a follow-up to our decoding findings, with the hypothesis that the temporal profile of decoding across regions, for a given variable, would be related to the cross-correlation of signals across those regions. Specifically, we hypothesized that areas earlier in the temporal profile of a variable (as informed by the results of our decoding analyses) would predict delayed signals in areas later in the timing profile of that variable. For EV, Outcome, and E.Risk, we aligned the ROIs based on their peak decoding accuracy and tested for differences in the cross-correlation between adjacent regions (e.g. for outcome representations: Angular Gyrus and Hippocampus, Hippocampus and Amygdala, etc) as a function of trials sorted along high or low levels of the variable. For RePE and RiPE, we defined pairs of ROIs for

cross-correlation analysis based on the temporal profile of decoding common to these two variables, in which RePE and RiPE are first decoded in one ROI, followed by parallel decoding across a set of regions. For these latter two variables we thus computed cross-correlations between a seed ROI in which we found early decoding (RePE: Cingulate gyrus; RiPE: OFC), and each target ROI in which we could decode the variable relatively later.

Indeed we agree that the direction of cross-correlation reported originally for outcome decoding in Frontal Pole - OFC and OFC - vmPFC is counter-intuitive. As discussed above, we now take great care in not over-interpreting these results, especially given our view that they do not hold up to a more rigorous statistical test in that the interaction is not significant. As such we have removed this result from the main text and it is presented in the supplementary section, along with additional results reporting other combinations of regions and across all other computational variables, for completeness. Instead, we now elaborate in the discussion suggestions for how future research might more strongly examine directed connections between regions, for example via experimentally manipulated perturbations to the signal in one region:

In 3. Discussion:

Indeed our finding that computations are structured in time across areas raised the question of whether the functional connectivity between different regions encoded computational information. Across all variables, we did not find significant differences in the cross-correlation between regions between high and low levels of that variable (see Supplementary section \ref{xcorr}). However, our approach is only one of several ways to address the question of directed functional connectivity relevant to processing these computations. Future work in human electrophysiology is well-positioned to elucidate the mechanisms driving directed connectivity. For example, cortico-cortical evoked potential manipulations \cite{matsumoto2004functional, keller2014mapping} directly perturb neural activity and therefore might provide a casual picture of how computational information propagates across the brain.

b) Result 3: Regarding expected risk results, I would suggest not to interpret non-significant results (even “suggestions”, which I understand as “trends”).

We have removed the interpretation of a ‘suggested’ sequential processing of expected risk based on uncorrected significant results in the posterior insula and supramarginal gyrus. We have also removed any interpretation of sub-threshold decoding accuracy in the OFC. Now we plainly present the results of our expected risk decoding analyses without striving to interpret temporal organization for the representation of this variable.

In our exploratory analysis we were able to decode expected risk in posterior insula ([0.102 - 0.154 sec], $\$C_{\{AUC\}}\$ = 0.456$, $\$C_{\{thresh\}}\$ = 0.302$, $p = 0.022$; $q = 0.103$, $\$AUC_{\{0.122\}}\$ = 0.574$, 95 CI [0.526 0.611]) and in the supramarginal gyrus of the intraparietal lobule ([0.449 -

0.499 sec], $C_{AUC} = 0.362$, $C_{thresh} = 0.234$, $p = 0.016$, $q = 0.103$, $AUC_{0.457} = 0.564$, 95 CI [0.526 0.602]; see Figure~S\ref{suppl:expectations}B).

Instead, we add in the Discussion a comment on our limitation in surveying temporal organization given the lack of E.Risk representation across multiple ROIs.

In section 3. Discussion:

...Further highlighting the importance of resolving computational timing at a fast timescale, we were able to decode expected risk decoded later in the trial prior to the onset of the second card consistent with the description of expected reward and risk in our normative model. Our results were also congruent with previous reports in human \cite{li2016neural} and animal \cite{fiorillo2003discrete, tobler2005adaptive, schultz2010dopamine} electrophysiology studies on the relative timings between expected reward and risk encoding, in which expected reward is encoded shortly after cue presentation whereas expected risk encoding is evident in activity starting before the period in which the outcome is known. **However, the temporal ordering of how multiple regions represent expected risk remains equivocal from our results, due to the lack of representation across multiple ROIs. While we found robust decoding of expected risk in OFC, contrary to our hypotheses we did not find that signal from the anterior insula could decode this variable....**

c) Result 5: It is not clear what “contrasted temporal profile of RiPE against RePE” means.

We agree that this wording is confusing and does not communicate the idea effectively. What we mean is that we could decode RePE earlier in the period after the onset of card 2 than RiPE in the same period, consistent with the prediction from the computational model. We have clarified this in the text:

In section 2.5: Error computations across domains share a mixture of temporal configurations:

...Moreover, we hypothesized that the timing of decoding for risk error should occur later than that for RePE, given that in the computational formulation the risk error (RiPE) term takes as input the RePE (see \ref{comp_model}). Consistent with the prediction, we were able to decode RePE earlier in the epoch after card 2 compared to RiPE using activity from the anterior insula.

3) Minor concerns:

• ***I don't understand the difference between figure 6A and figure S3***

We apologize for the confusion brought on by an error in creating the figure x-axis and y-axis ticks in the supplementary. The supplementary figure shows that the generalization results shown in the main text in Figure 6A was robust against using a constrained time window of 0.200 - 0.500 sec after the onset of card 2, and continued to remain significant when we

examined the entire epoch. As we describe in the text, this constrained window was originally chosen to capture the window of RePE and RiPE decoding, respectively, in the anterior insula.

In section 2.6 Anterior insula representations of reward error can predict risk error

Consistent with the idea that the brain needs to first compute the reward prediction error before using this information to compute the risk error, we hypothesized that activity in the time window of RePE representation would be able to predict activity related to RiPE at a relatively later period in the same epoch. We constrained our decoding analysis here to the period in the trial relevant to RePE and RiPE representation (0.200 - 0.500 sec after the onset of card 2) and tested for generalization both across variables (i.e. trained on reward error and tested on risk error) and time. Indeed, we found evidence of significant cross-validated generalized decoding... We continued to find evidence of significant generalization when tested against a more stringent statistical threshold (trained on RePE: [0.200 - 0.283 sec]; test: [0.439 - 0.489 sec]; $AUC = 6.554$, $C_{\text{thresh}} = 0.460$, $p < 0.001$, orange cluster in Figure \ref{fig6}A), and when we did not constrain the time window for analysis (see Figure~\ref{suppl:antIns_gen}).

We have now corrected the supplementary figure:

• *The task should be briefly described in the abstract.*

We thank the reviewer for this suggestion. We have added a short description of the task in the abstract, which has been shortened to meet editorial formatting instructions:

The value and uncertainty associated with choice alternatives constitute critical features relevant for decisions. However, the manner in which reward and risk representations are temporally organized in the brain remains elusive. **Here we leverage the spatiotemporal precision of intracranial electroencephalography, along with a simple card game designed to elicit the unfolding computation of a set of reward and risk variables, to**

uncover this temporal organization. Reward outcome representations across wide-spread regions follow a sequential order along the anteroposterior axis of the brain. In contrast, expected value can be decoded from multiple regions at the same time, and error signals in both reward and risk domains reflect a mixture of sequential and parallel encoding. We further highlight the role of the anterior insula in generalizing between reward prediction error and risk prediction error codes. Together our results emphasize the importance of neural dynamics for understanding value-based decisions under uncertainty.

References

Aquino, Tomas G., et al. "Encoding of predictive associations in human prefrontal and medial temporal neurons during Pavlovian conditioning." *bioRxiv* (2023): 2023-02.

Bradley, Andrew P. "The use of the area under the ROC curve in the evaluation of machine learning algorithms." *Pattern recognition* 30.7 (1997): 1145-1159.

Davis, Tyler, et al. "What do differences between multi-voxel and univariate analysis mean? How subject-, voxel-, and trial-level variance impact fMRI analysis." *Neuroimage* 97 (2014): 271-283.

Desikan, Rahul S., et al. "An automated labeling system for subdividing the human cerebral cortex on MRI scans into gyral based regions of interest." *Neuroimage* 31.3 (2006): 968-980.

Domenech, Philippe, Sylvain Rheims, and Etienne Koechlin. "Neural mechanisms resolving exploitation-exploration dilemmas in the medial prefrontal cortex." *Science* 369.6507 (2020): eabb0184.

Efron, Bradley. "Estimating the error rate of a prediction rule: improvement on cross-validation." *Journal of the American statistical association* 78.382 (1983): 316-331.

Efron, Bradley, and Robert Tibshirani. "Improvements on cross-validation: the 632+ bootstrap method." *Journal of the American Statistical Association* 92.438 (1997): 548-560.

Etzel, Joset A., Jeffrey M. Zacks, and Todd S. Braver. "Searchlight analysis: promise, pitfalls, and potential." *Neuroimage* 78 (2013): 261-269.

Fiorillo, Christopher D., Philippe N. Tobler, and Wolfram Schultz. "Discrete coding of reward probability and uncertainty by dopamine neurons." *Science* 299.5614 (2003): 1898-1902.

Geman, Stuart, Elie Bienenstock, and René Doursat. "Neural networks and the bias/variance dilemma." *Neural computation* 4.1 (1992): 1-58.

Goodfellow, Ian, Yoshua Bengio, and Aaron Courville. *Deep learning*. MIT press, 2016.

Hare, Todd A., et al. "Dissociating the role of the orbitofrontal cortex and the striatum in the computation of goal values and prediction errors." *Journal of neuroscience* 28.22 (2008): 5623-5630.

Haynes, John-Dylan, and Geraint Rees. "Decoding mental states from brain activity in humans." *Nature reviews neuroscience* 7.7 (2006): 523-534.

Hebart, Martin N., and Chris I. Baker. "Deconstructing multivariate decoding for the study of brain function." *Neuroimage* 180 (2018): 4-18.

Hosmer Jr, David W., Stanley Lemeshow, and Rodney X. Sturdivant. *Applied logistic regression*. Vol. 398. John Wiley & Sons, 2013.

Kahnt, Thorsten. "A decade of decoding reward-related fMRI signals and where we go from here." *Neuroimage* 180 (2018): 324-333.

Keller, Corey J., et al. "Mapping human brain networks with cortico-cortical evoked potentials." *Philosophical Transactions of the Royal Society B: Biological Sciences* 369.1653 (2014): 20130528.

Li, Yansong, et al. "The neural dynamics of reward value and risk coding in the human orbitofrontal cortex." *Brain* 139.4 (2016): 1295-1309.

Ling, Charles X., Jin Huang, and Harry Zhang. "AUC: a statistically consistent and more discriminating measure than accuracy." *Ijcai*. Vol. 3. 2003.

Lopez-Persem, Alizée, et al. "Four core properties of the human brain valuation system demonstrated in intracranial signals." *Nature Neuroscience* 23.5 (2020): 664-675.

Marciano, Deborah, et al. "Electrophysiological signatures of inequity-dependent reward encoding in the human OFC." *Cell Reports* 42.8 (2023).

Maris, Eric, and Robert Oostenveld. "Nonparametric statistical testing of EEG-and MEG-data." *Journal of neuroscience methods* 164.1 (2007): 177-190.

Matsumoto, Riki, et al. "Functional connectivity in the human language system: a cortico-cortical evoked potential study." *Brain* 127.10 (2004): 2316-2330.

Meyers, Ethan M. "The neural decoding toolbox." *Frontiers in neuroinformatics* 7 (2013): 8.

Michelmann, Sebastian, et al. "Data-driven re-referencing of intracranial EEG based on independent component analysis (ICA)." *Journal of neuroscience methods* 307 (2018): 125-137.

Mumford, Jeanette A., et al. "Deconvolving BOLD activation in event-related designs for multivoxel pattern classification analyses." *Neuroimage* 59.3 (2012): 2636-2643.

Neubert, Franz-Xaver, et al. "Connectivity reveals relationship of brain areas for reward-guided learning and decision making in human and monkey frontal cortex." *Proceedings of the national academy of sciences* 112.20 (2015): E2695-E2704.

Preuschoff, Kerstin, Steven R. Quartz, and Peter Bossaerts. "Human insula activation reflects risk prediction errors as well as risk." *Journal of Neuroscience* 28.11 (2008): 2745-2752.

Rutishauser, Ueli, et al. "Representation of retrieval confidence by single neurons in the human medial temporal lobe." *Nature neuroscience* 18.7 (2015): 1041-1050.

Saez, Ignacio, et al. "Encoding of multiple reward-related computations in transient and sustained high-frequency activity in human OFC." *Current Biology* 28.18 (2018): 2889-2899.

Smithson, Michael. "The receiver operating characteristic area under the curve (or mean rdit) as an effect size." *Psychological Methods* (2023).

REVIEWERS' COMMENTS

Reviewer #1 (Remarks to the Author):

The effort that the authors put into revising their manuscript is commendable. They have addressed each of my concerns comprehensively.

There is just one very minor issue that was introduced in the revision:

The last sentence in "4.10.5 Confidence Intervals" seems to be missing some words: "The approach of bootstrapping the test set has been shown through simulation studies to more accurately."

Besides this, I have no further reservations regarding this paper and would like to thank the authors for addressing the previously raised issues so thoroughly.

Reviewer #2 (Remarks to the Author):

I find the authors have substantially addressed my comments and I recommend publication.

Reviewer #3 (Remarks to the Author):

The authors have addressed my comments with great rigor and seriousness. Their answers are complete, and the manuscript has been corrected in line with their responses and the reviewers' requests. I have no further comments and congratulate the authors on their well-done job.

Reviewer #1 (Remarks to the Author):

The effort that the authors put into revising their manuscript is commendable. They have addressed each of my concerns comprehensively.

There is just one very minor issue that was introduced in the revision:

The last sentence in "4.10.5 Confidence Intervals" seems to be missing some words: "The approach of bootstrapping the test set has been shown through simulation studies to more accurately."

Besides this, I have no further reservations regarding this paper and would like to thank the authors for addressing the previously raised issues so thoroughly.

Thank you very much. The incomplete last sentence in 4.10.5 Confidence Intervals has been removed for clarity.

Reviewer #2 (Remarks to the Author):

I find the authors have substantially addressed my comments and I recommend publication.

Thank you.

Reviewer #3 (Remarks to the Author):

The authors have addressed my comments with great rigor and seriousness. Their answers are complete, and the manuscript has been corrected in line with their responses and the reviewers' requests. I have no further comments and congratulate the authors on their well-done job.

Thank you.